# Breaking Barriers: Combinatorial Algorithms for Non-Monotone Submodular Maximization with Sublinear Adaptivity and $1/e$ Approximation

**Yixin Chen** [1]   **Wenjing Chen** [1]   **Alan Kuhnle** [1]

## Abstract

With the rapid growth of data in modern applications, parallel algorithms for maximizing non-monotone submodular functions have gained significant attention. In the parallel computation setting, the state-of-the-art approximation ratio of $1/e$ is achieved by a continuous algorithm (Ene & Nguyen, 2020) with adaptivity $\mathcal{O}\left(\log(n)\right)$. In this work, we focus on size constraints and present the first combinatorial algorithm matching this bound – a randomized parallel approach achieving $1/e - \varepsilon$ approximation ratio. This result bridges the gap between continuous and combinatorial approaches for this problem. As a byproduct, we also develop a simpler $(1/4 - \varepsilon)$-approximation algorithm with high probability ($\geq 1 - 1/n$). Both algorithms achieve $\mathcal{O}\left(\log(n)\log(k)\right)$ adaptivity and $\mathcal{O}\left(n\log(n)\log(k)\right)$ query complexity. Empirical results show our algorithms achieve competitive objective values, with the $(1/4 - \varepsilon)$-approximation algorithm particularly efficient in queries.

## 1. Introduction

**Submodular Optimization.** Submodular optimization is a powerful framework for solving combinatorial optimization problems that exhibit diminishing returns (Nemhauser et al., 1978; Feige & Goemans, 1995; Cornuejols et al., 1977). In a monotone setting, adding more elements to a solution always increases its utility, but at a decreasing rate. On the other hand, non-monotone objectives may have elements that, when added, can reduce the utility of a solution. This versatility makes submodular optimization widely applicable across various domains. For instance, in data summarization (Mirzasoleiman et al., 2018; Tschi-

atschek et al., 2014) and feature selection (Bilmes, 2022), it helps identify the most informative subsets efficiently. In social network analysis (Kempe et al., 2003), it aids in influence maximization by selecting a subset of individuals to maximize information spread. Additionally, in machine learning (Bairi et al., 2015; Elenberg et al., 2017; Prajapat et al., 2024), submodular functions are used for diverse tasks like active learning, sensor placement, and diverse set selection in recommendation systems. These applications demonstrate the flexibility and effectiveness of submodular optimization in addressing real-world problems with both monotone and non-monotone objectives.

**Problem Definition and Greedy Algorithms.** In this work, we consider the size-constrained maximization of a submodular function: given a submodular function $f$ on ground set of size $n$, and given an integer $k$, find $\arg\max_{S \subseteq \mathcal{U}, |S| \leq k} f(S)$. If additionally $f$ is assumed to be monotone, we refer to this problem as SM-MON; otherwise, we call the problem SM-GEN. For SM-MON, a standard greedy algorithm gives the optimal[1] approximation ratio of $1 - 1/e \approx 0.63$ (Nemhauser et al., 1978) in at most $kn$ queries[2] to $f$. In contrast, standard greedy can't achieve any constant approximation factor in the non-monotone case; however, an elegant, randomized variant of greedy, the RAN-DOMGREEDY of Buchbinder et al. (2014) obtains the same greedy ratio (in expectation) of $1 - 1/e$ for monotone objectives, and $1/e \approx 0.367$ for the general, non-monotone case, also in at most $kn$ queries. However, as modern instance sizes in applications have become very large, $kn$ queries is too many. In the worst case, $k = \Omega(n)$, and the time complexity of these greedy algorithms becomes quadratic. To improve efficiency, Buchbinder et al. (2017) leverage sampling technique to get FASTRANDOMGREEDY, reducing the query complexity to $\mathcal{O}\left(n\right)$ when $k \geq 8\varepsilon^{-2}\log(2\varepsilon^{-1})$.

**Parallelizable Algorithms.** In addition to reducing the number of queries, recently, much work has focused on developing parallelizable algorithms for submodular opti-

---

[1]Department of Computer Science & Engineering, Texas A&M University, College Station, Texas, USA. Correspondence to: Alan Kuhnle <kuhnle@tamu.edu>.

*Proceedings of the 42$^{nd}$ International Conference on Machine Learning*, Vancouver, Canada. PMLR 267, 2025. Copyright 2025 by the author(s).

---

[1]Optimal in polynomially many queries to $f$ in the value query model (Nemhauser & Wolsey, 1978).

[2]Typically, queries to $f$ dominate other parts of the computation, so in this field time complexity of an algorithm is usually given as oracle complexity to $f$.

*Table 1.* Theoretical comparison of greedy algorithms and parallel algorithms with sublinear adaptivity.

| Algorithm | Ratio | Queries | Adaptivity |
|---|---|---|---|
| FASTRANDOMGREEDY (Buchbinder et al., 2017) | $\mathbf{1/e - \varepsilon}$ | $\mathcal{O}(n)$ | $k$ |
| ANM (Fahrbach et al., 2023) | $0.039 - \varepsilon$ | $\mathcal{O}(n\log(k))$ | $\mathcal{O}(\log(n))$ |
| Ene & Nguyen (2020) | $\mathbf{1/e - \varepsilon}$ | $\mathcal{O}(k^2 n\log^2(n))$ ‡ | $\mathcal{O}(\log(n))$ |
| PARKNAPSACK(Amanatidis et al., 2021) | $0.172 - \varepsilon$ | $\mathcal{O}(kn\log(n)\log(k))\,\|\,\mathcal{O}(n\log(n)\log^2(k))$ | $\mathcal{O}(\log(n))\,\|\,\mathcal{O}(\log(n)\log(k))$ |
| AST (Chen & Kuhnle, 2024) | $1/6 - \varepsilon$ | $\mathcal{O}(n\log(k))$ | $\mathcal{O}(\log(n))$ |
| ATG (Chen & Kuhnle, 2024) | $0.193 - \varepsilon$ | $\mathcal{O}(n\log(k))$ | $\mathcal{O}(\log(n)\log(k))$ |
| PARSKP (Cui et al., 2023) | $0.125 - \varepsilon$ | $\mathcal{O}(kn\log^2(n))\,\|\,\mathcal{O}(n\log^2(n)\log(k))$ | $\mathcal{O}(\log(n))\,\|\,\mathcal{O}(\log(n)\log(k))$ |
| PARSSP (Cui et al., 2023) | $0.25 - \varepsilon$ | $\mathcal{O}(kn\log^2(n))\,\|\,\mathcal{O}(n\log^2(n)\log(k))$ | $\mathcal{O}(\log^2(n))\,\|\,\mathcal{O}(\log^2(n)\log(k))$ |
| PARALLELINTERLACEGREEDY (Alg. 6) | $0.25 - \varepsilon$† | $\mathcal{O}(n\log(n)\log(k))$ | $\mathcal{O}(\log(n)\log(k))$ |
| PARALLELINTERPOLATEDGREEDY (Alg. 11) | $\mathbf{1/e - \varepsilon}$ | $\mathcal{O}(n\log(n)\log(k))$ | $\mathcal{O}(\log(n)\log(k))$ |

‡ The parallel algorithm in Ene & Nguyen (2020) queries to the continuous oracle.
† The approximation ratio is achieved with high probability (at least $1 - 1/n$).

mization. One measure of parallelizability is the *adaptive complexity* of an algorithm. That is, the queries to $f$ are divided into adaptive rounds, where within each round the set queried may only depend on the results of queries in previous rounds; the queries within each round may be arbitrarily parallelized. Thus, the lower the adaptive complexity, the more parallelizable an algorithm is. Although the initial algorithms with sublinear adaptivity were impractical, for the monotone case, these works culminated in two practical algorithms: FAST (Breuer et al., 2020) and LS+PGB (Chen et al., 2021), both of which achieve nearly the optimal ratio in nearly linear time and nearly optimal adaptive rounds. For the nonmonotone case, the best approximation ratio achieved in sublinear adaptive rounds is $1/e$ (Ene & Nguyen, 2020). However, this algorithm queries to a continuous oracle, which needs to be estimated through a substantial number of queries to the original set function oracle. Although practical, sublinearly adaptive algorithms have also been developed, the best ratio achieved in nearly linear time is nearly $1/4$ (Cui et al., 2023), significantly worse than the state-of-the-art[3], this $1/4$ ratio also stands as the best even for superlinear time parallel algorithms. Further references to parallel algorithms and their theoretical guarantees are provided in Table 1.

**Greedy Variants for Parallelization.** To enhance the approximation ratios for combinatorial sublinear adaptive algorithms, it is crucial to develop practical parallelizable algorithms that serve as universal frameworks for such parallel approaches. Among existing methods, INTERLACEGREEDY (Kuhnle, 2019), with an approximation ratio of $1/4$, and INTERPOLATEDGREEDY (Chen & Kuhnle, 2023), with an expected approximation ratio of $1/e$, have emerged as promising candidates due to their unique and deterministic interlacing greedy procedures.

INTERLACEGREEDY (Alg. 1) operates by first guessing whether the maximum singleton $a_0 = \arg\max_{x \in \mathcal{U}} f(x)$ is

---

[3]The best known ratio in polynomial time was very recently improved from close to $1/e$ to $0.401$ (Buchbinder & Feldman, 2024)

---

**Algorithm 1:** INTERLACEGREEDY$(f, k)$: The INTERLACEGREEDY Algorithm (Kuhnle, 2019)

**Input:** evaluation oracle $f : 2^{\mathcal{U}} \to \mathbb{R}_{\geq 0}$, constraint $k$
**Initialize:** $a_0 \leftarrow \arg\max_{x \in \mathcal{U}} f(x)$, $A \leftarrow B \leftarrow \emptyset$,
$\quad\quad\quad D \leftarrow E \leftarrow \{a_0\}$, add $2k$ dummy elements
$\quad\quad\quad$ to the ground set

**1 for** $i \leftarrow 0$ *to* $k - 1$ **do**
**2** $\quad A \leftarrow A + \arg\max_{x \in \mathcal{U} \setminus (A \cup B)} \Delta(x \mid A)$
**3** $\quad B \leftarrow B + \arg\max_{x \in \mathcal{U} \setminus (A \cup B)} \Delta(x \mid B)$
**4 for** $i \leftarrow 1$ *to* $k - 1$ **do**
**5** $\quad D \leftarrow D + \arg\max_{x \in \mathcal{U} \setminus (D \cup E)} \Delta(x \mid D)$
**6** $\quad E \leftarrow E + \arg\max_{x \in \mathcal{U} \setminus (D \cup E)} \Delta(x \mid E)$
**7 return** $C \leftarrow \arg\max\{f(A), f(B), f(D), f(E)\}$

---

contained within the optimal solution $O$. Based on this hypothesis, the algorithm initializes two solution pools differently: if $a_0$ is not in $O$, the pools begin empty ($A = B = \emptyset$); otherwise, they are initialized with $a_0$ ($D = E = \{a_0\}$). The algorithm then proceeds by alternately selecting elements for each pool in a greedy fashion, ultimately returning the best solution found. We provide its theoretical guarantees below.

**Theorem 1.1.** *Let* $f : 2^{\mathcal{U}} \to \mathbb{R}_{\geq 0}$ *be submodular, let* $k \in \mathcal{U}$, *let* $O = \arg\max_{|S| \leq k} f(S)$, *and let* $C = $ INTERLACEGREEDY$(f, k)$. *Then*

$$f(C) \geq f(O)/4,$$

*and* INTERLACEGREEDY *makes* $\mathcal{O}(kn)$ *queries to* $f$.

The key to this guarantee lies in the alternating selection strategy between the two pools. For any disjoint pools $\{S, T\}$, it holds that

$$f(S) + f(T) \geq f(O \cup S) + f(O \cup T) \geq f(O),$$

where the first inequality follows from the alternating selection strategy, and the second inequality follows from submodularity and monotonicity.

Building on this, INTERPOLATEDGREEDY (Chen & Kuhnle, 2023) generalizes the approach by maintaining $\ell$ sets, each containing $k/\ell$ elements at each iteration (see pseudocode and theoretical guarantees in Appendix C). However, in their original formulations, both algorithms require an initial step to guess whether the first element added to each solution belongs to the optimal set $O$. This results in repeated for loop with different initial values in INTERLACEGREEDY and a success probability of $(\ell+1)^{-\ell}$.

## 1.1. Contributions

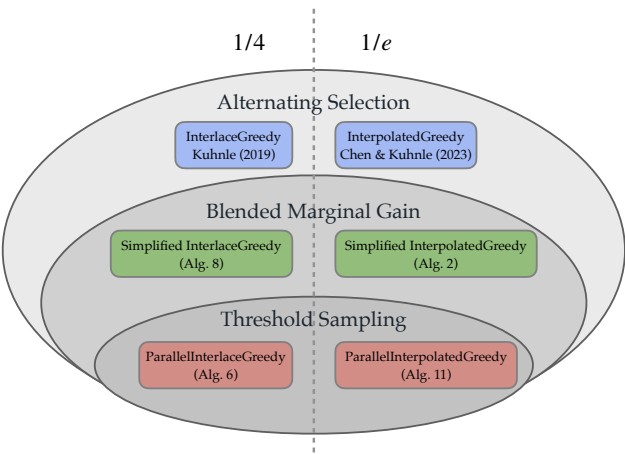

*Figure 1.* This figure illustrates strategies employed by each algorithm. The three leftmost algorithms achieve an asymptotic approximation ratio of $1/4$, while the three rightmost algorithms attain an asymptotic ratio of $1/e$.

**Technical Contributions: Blended Marginal Gains Strategy.** Our first contribution is a novel method of analysis for interlaced greedy-based algorithms, such as INTERPOLATEDGREEDY (Chen & Kuhnle, 2023). The core innovation introduces multiple upper bounds on the marginal gain of the same element regarding the solution, offering greater flexibility in the analysis. This approach yields three notable advances: First, we present Alg. 2 (Section 3) with $\mathcal{O}(kn)$ query complexity, achieving an expected approximation ratios of $1/e - \varepsilon$, eliminating the probabilistic guessing step that limited prior method to $(1+\ell)^{-\ell}$ success probability. Second, as a byproduct, we also provide a simplified version of INTERLACEGREEDY (Kuhnle, 2019) that preserves its theoretical guarantees in Appendix D. Most importantly, these simplified variants establish the theoretical foundation for parallel algorithms. By removing branching dependencies and probabilistic guesswork, our framework enables the first efficient parallelization of these interlaced greedy approaches while preserving their approximation guarantees.

**Parallel Algorithms with Logarithmic Adaptivity and**

**Nearly-linear Query Complexity Using a Unified Subroutine.** In Section 4, we present two sublinear adaptive algorithms, PARALLELINTERLACEGREEDY (PIG) and PARALLELINTERPOLATEDGREEDY (PITG) that share a unified subroutine, PIG. The core innovation of PIG lies in its novel threshold sampling procedure, which simultaneously preserves the crucial alternating selection property of interlaced greedy methods while enabling efficient parallel implementation.

Like prior parallel algorithms, PIG maintains descending thresholds for each solution, adding elements whose average marginal gain exceeds the current threshold. However, PIG introduces two critical modifications to maintain the interlaced greedy structure: 1) strict synchronization of batch sizes across all $\ell$ parallel solutions, and 2) coordinated element selection to maintain sufficient marginal gain for each solution.

This design achieves three fundamental properties. First, it preserves the essential alternating selection property of the interlaced greedy methods. Second, through threshold sampling, it geometrically reduces the size of candidate sets - crucial for achieving sublinear adaptivity. Third, its efficient batch selection ensures each added batch provides sufficient marginal contribution to the solution. Together, these properties allow PIG to match the approximation guarantees of the vanilla interlaced greedy method while achieving parallel efficiency.

Leveraging this unified framework, a single call to PIG achieves an approximation ratio of $(1/4 - \varepsilon)$ with high probability. Repeated calls to PIG yields PITG, further enhance the approximation ratio to an expected $(1/e - \varepsilon)$. Both algorithms achieve $\mathcal{O}(\log(n)\log(k))$ adaptivity and $\mathcal{O}(n\log(n)\log(k))$ query complexity, making them efficient for large-scale applications.

**Empirical Evaluation.** Finally, we evaluate the performance of our parallel algorithms in Section 5 and Appendix H across two applications and four datasets. The results demonstrate that our algorithms achieve competitive objective values. Notably, PARALLELINTERLACEGREEDY outperforms other algorithms in terms of query efficiency, highlighting its practical advantages.

## 2. Preliminary

**Notation.** We denote the marginal gain of adding $A$ to $B$ by $\Delta(A \mid B) = f(A \cup B) - f(B)$. For every set $S \subseteq U$ and an element $x \in \mathcal{U}$, we denote $S \cup \{x\}$ by $S + x$ and $S \setminus \{x\}$ by $S - x$.

**Submodularity.** A set function $f : 2^{\mathcal{U}} \to \mathbb{R}_{\geq 0}$ is submodular, if $\Delta(x \mid S) \geq \Delta(x \mid T)$ for all $S \subseteq T \subseteq \mathcal{U}$ and $x \in \mathcal{U} \setminus T$, or equivalently, for all $A, B \subseteq \mathcal{U}$, it holds

that $f(A) + f(B) \geq f(A \cup B) + f(A \cap B)$. With a size constraint $k$, let $O = \arg\max_{S \subseteq \mathcal{U}, |S| \leq k} f(S)$.

In Appendix B, we provide several key propositions derived from submodularity that streamline the analysis.

**Organization.** In Section 3, the blending technique is introduced and applied to INTERLACEGREEDY and INTERPOLATEDGREEDY, with detailed analysis provided in Appendix D and E. Section 4 discusses their fast versions with pseudocodes and comprehensive analysis in Appendix F. Subsequently, Section 4 delves into our sublinear adaptive algorithms, with further technical details and proofs available in Appendix G. Finally, we provide the empirical evaluation in Section 5, with its detailed setups and additional results in Appendix H.

# 3. A Parallel-Friendly Greedy Variant via Blending Technique

In this section, we present a simplified and practical variant of INTERPOLATEDGREEDY (Chen & Kuhnle, 2023) that retains its theoretical guarantees while improving its success probability from $(\ell+1)^{-\ell}$ to 1. This simplification serves as a key step toward developing efficient parallel submodular maximization algorithms (Section 4). As a byproduct, we also show that INTERLACEGREEDY (Kuhnle, 2019) can be simplified (Appendix D) and parallelized (Section 4).

## 3.1. An Overview of Prior Work and Its Limitation

INTERLACEGREEDY (Alg. 1) employs a two-branch initialization strategy conditioned on differently based on the guessing of whether the maximum singleton $a_0 = \arg\max_{x \in \mathcal{U}} f(x)$ is contained within the optimal solution $O$. The algorithm initializes two solution pools either as empty sets if $a_0 \notin O$, or both containing $a_0$ otherwise. It then alternates between two greedy procedures, growing solutions pools one by one until they reach size $k$. The algorithm returns the best solution, achieving a $1/4 - \varepsilon$ approximation. This structure is necessary to handle the uncertainty about whether the maximum singleton $a_0$ coincides with $o_{\max} = \arg\max_{o \in O} f(o)$.

Later on, INTERPOLATEDGREEDY (Alg. 7, Appendix C) generalizes this idea by interlacing $\ell$ greedy procedures, each adding $k/\ell$ elements to intermediate solutions. Critically, the algorithm must account for uncertainty in the position of $o_{\max}$. Since $o_{\max}$ could be one of the top-$\ell$ elements in marginal gain (or none of them), the algorithm must consider $\ell+1$ possible cases. To handle this, it maintains $\ell+1$ *solution families* (a set of multiple potential solutions), each corresponding to a distinct guess about where $o_{\max}$ appears. The algorithm then returns a random solution from the solution families, achieving a $1/e - \varepsilon$ approximation with success probability $(\ell+1)^{-\ell}$. To guarantee

success probability 1, all possible solution branches must be checked, increasing the query complexity by a prohibitive $\mathcal{O}\left((\ell+1)^{\ell}\right)$ factor. While asymptotically efficient ($\mathcal{O}(nk)$) queries), INTERPOLATEDGREEDY suffers from either low success probability or impractical computational overhead, making it unsuitable for parallelization.

## 3.2. Motivation for Simplification

The primary challenge in parallelizing INTERPOLATEDGREEDY stems from its inherent branching structure - specifically, the need to make $\ell+1$ guesses, each resulting in an independent construction path for solution pools. This complex architecture naturally raises a fundamental question: *Can we develop an alternative analysis framework that eliminates the need to explicitly guess the position of $o_{\max}$?*

To address this, we introduce a novel blended marginal gains strategy. Our key insight demonstrates that by carefully tracking combined marginal gains across iterations, we can effectively work with just a single solution family of pairwise disjoint solutions while preserving all theoretical guarantees.

This simplification not only boosts the success probability to 1 but also enables a more efficient parallel implementation. We present the pseudocode of the simplified INTERPOLATEDGREEDY in Alg. 2, with theoretical guarantees in Theorem 3.1.

---

**Algorithm 2:** A simplified INTERPOLATEDGREEDY with a randomized $1/e$ approximation ratio and $\mathcal{O}(nk\ell)$ query complexity.

**Input:** evaluation oracle $f : 2^{\mathcal{U}} \to \mathbb{R}_{\geq 0}$, constraint $k$, size of solution $\ell$, error $\varepsilon$
**Initialize:** $G \leftarrow \emptyset$, $V \leftarrow \mathcal{U}$, $m \leftarrow \lfloor \frac{k}{\ell} \rfloor$, add $2k$ dummy elements to the ground set.

1 **for** $i \leftarrow 1$ *to* $\ell$ **do**
2    $A_l \leftarrow G, \forall l \in [\ell]$
3    **for** $j \leftarrow 1$ *to* $m$ **do**
4      **for** $l \leftarrow 1$ *to* $\ell$ **do**
5        $a \leftarrow \arg\max_{x \in V} \Delta(x \mid A_l)$
6        $A_l \leftarrow A_l + a, V \leftarrow V - a$
7    $G \leftarrow$ a random set in $\{A_l\}_{l \in [\ell]}$
8 **return** $G$

---

**Theorem 3.1.** *With input instance $(f, k, \ell, \varepsilon)$ such that $\ell = \mathcal{O}(\varepsilon^{-1}) \geq \frac{2}{e\varepsilon}$ and $k \geq \frac{2(e\ell-2)}{e\varepsilon - \frac{2}{\ell}}$, Alg. 2 returns a set $G$ with $\mathcal{O}(kn/\varepsilon)$ queries such that $\mathbb{E}[f(G)] \geq (1/e - \varepsilon) f(O)$.*

### 3.3. Technical Overview of Blended Marginal Gains Strategy for INTERPOLATEDGREEDY

In INTERPOLATEDGREEDY, each iteration maintains $\ell + 1$ solution families, each containing $\ell$ nearly pairwise disjoint sets. Among these solution families, only one is *right* which satisfies the following key inequality:

$$\Delta\left(O \mid A_{u,i}\right) \leq \ell \Delta\left(A_{u,i} \mid G\right), \forall i \in [\ell], \qquad (1)$$

where $G$ is the intermediate solution at the start of this iteration, and $A_{u,i}$ is the $i$-th solution in the $u$-th solution families at the end of this iteration.

The right solution families guarantees that elements added in the first round do not belong to $O \setminus A_{u,i}$ for any $i \in [\ell]$. This property enables a partition of the optimal solution $O$ into $k/\ell$ subsets of size $\ell$, where each subset's marginal gain is dominated by a corresponding element in $A_{u,i}$. Consequently, $\Delta\left(O \mid A_{u,i}\right)$ depends solely on $\Delta\left(A_{u,i} \mid G\right)$.

In what follows, we introduce a novel blended marginal gains approach to analyze INTERPOLATEDGREEDY using only a single interlaced greedy step (Alg. 2). This approach leverages a mixture of marginal gains across solutions to derive tighter bounds for each $\Delta\left(O \mid A_l\right)$. The analysis proceeds in four steps:

**Step 1: Partitioning the Optimal Solution $O$.** Our analysis begins by establishing a correspondence between the algorithm's solutions and partitions of the optimal set $O$. Claim 3.1 provides the foundation for this pairing:

**Claim 3.1.** At an iteration $i$ of the outer for loop in Alg. 2, let $G_{i-1}$ be $G$ at the start of this iteration, and $A_l$ be the set at the end of this iteration, for each $l \in [\ell]$. The set $O \setminus G_{i-1}$ can then be split into $\ell$ pairwise disjoint sets $\{O_1, \ldots, O_\ell\}$ such that $|O_l| \leq \frac{k}{\ell}$ and $(O \setminus G_{i-1}) \cap (A_l \setminus G_{i-1}) \subseteq O_l$, for all $l \in [\ell]$.

This partition enables us to decompose the marginal gains of $O$ with respect to each solution $A_l$. Specifically, we express the total marginal gain as:

$$\sum_{l \in [\ell]} \Delta\left(O \mid A_l\right) \leq \sum_{l \in [\ell]} \sum_{i \in [\ell]} \Delta\left(O_i \mid A_l\right) \quad \text{(Proposition B.1)}$$

$$= \sum_{1 \leq l_1 < l_2 \leq \ell} \left(\Delta\left(O_{l_1} \mid A_{l_2}\right) + \Delta\left(O_{l_2} \mid A_{l_1}\right)\right)$$

$$+ \sum_{l \in [\ell]} \Delta\left(O_l \mid A_l\right). \qquad (2)$$

The decomposition consists of two types of terms: 1) self-interaction term $\Delta\left(O_l \mid A_l\right)$ for each $l \in [\ell]$, and 2) cross-interaction terms $\Delta\left(O_{l_1} \mid A_{l_2}\right) + \Delta\left(O_{l_2} \mid A_{l_1}\right)$ for each $1 \leq l_1 < l_2 \leq \ell$. Below we establish upper bounds for each term type, with detailed analysis to follow.

**Lemma 3.2.** Fix on $G_{i-1}$ for an iteration $i$ of the outer for loop in Alg. 2. Following the definition in Claim 3.1, it holds that

1) $\Delta\left(A_l \mid G_{i-1}\right) \geq \Delta\left(O_l \mid A_l\right), \forall 1 \leq l \leq \ell$,

2) $\left(1 + \dfrac{1}{m}\right)\left(\Delta\left(A_{l_1} \mid G_{i-1}\right) + \Delta\left(A_{l_2} \mid G_{i-1}\right)\right)$
$\geq \Delta\left(O_{l_2} \mid A_{l_1}\right) + \Delta\left(O_{l_1} \mid A_{l_2}\right), \forall 1 \leq l_1 < l_2 \leq \ell.$

**Step 2: Self-Interaction Term Bounding.** The partition from Claim 3.1 immediately yields our first bound. For any $l \in [\ell]$, elements in $O_l \setminus A_l$ were available but not selected by the greedy procedure, directly implying the first required bound.

**Step 3: Cross-Interaction Term Bounding.** The primary technical challenge lies in effectively bounding the cross-interaction terms $\Delta\left(A_{l_1} \mid G_{i-1}\right) + \Delta\left(A_{l_2} \mid G_{i-1}\right)$. Rather than relying on original greedy selection bounds, we develop a more sophisticated approach through our blended marginal gains technique, formalized in the following proposition:

**Proposition 3.3** (Blended Marginal Gains)*. For any submodular function $f : 2^{\mathcal{U}} \to \mathbb{R}_{\geq 0}$ and $S, T, O \in \mathcal{U}$, Let $S_i$ be a prefix of $S$ with size $i$ such that $S_i \subseteq O$. Similarly, define $T_j$. It satisfies that,*

$$\Delta\left(O \mid T\right) \leq \Delta\left(S_i \mid T\right) + \Delta\left(O \setminus S_i \mid T\right), \qquad (3)$$
$$\Delta\left(O \mid S\right) \leq \Delta\left(T_j \mid S\right) + \Delta\left(O \setminus T_j \mid S\right), \qquad (4)$$

The key insight involves strategically partitioning $O$ into two components: 1) a subset keeps the original greedy selection bound, and 2) a residual subset where we apply submodularity only.

Applying this proposition to each solution pair $(A_{l_1}, A_{l_2})$ with carefully chosen prefixes yields the second inequality in Lemma 3.2.

**Step 4: Final Composition.** By applying Inequality (2) and Lemma 3.2, we derive a result analogous to Inequality (1) achieved by INTERPOLATEDGREEDY,

$$\sum_{l \in [\ell]} \Delta\left(O \mid A_l\right) \leq \ell \left(1 + \frac{1}{m}\right) \sum_{l \in [\ell]} \Delta\left(A_l \mid G_{i-1}\right). \qquad (5)$$

This forms the key property necessary to establish the $1/e - \varepsilon$ approximation ratio while requiring only a single pool of solutions. The detailed analysis of the approximation ratio is provided in Appendix E.

## 4. Sublinear Adaptive Algorithms

In this section, we present the main subroutine for our parallel algorithms, PARALLELINTERLACEGREEDY (PIG, Alg. 6). A single execution of PIG achieves an approximation ratio of $1/4 - \varepsilon$ with high probability, while repeatedly running PIG, as in PARALLELINTERPOLATEDGREEDY (PITG, Alg. 11 in Appendix G.3), guarantees a randomized approximation ratio of $1/e - \varepsilon$. Below, we outline the theoretical guarantees, with the detailed analysis provided in Appendix G.

**Theorem 4.1.** *With input* $(f, k, 2, \frac{\varepsilon M}{k}, \varepsilon)$, *where* $M = \max_{x \in \mathcal{U}} f(x)$, PIG *(Alg. 6) returns* $\{A'_1, A'_2\}$ *with* $\mathcal{O}\left(\varepsilon^{-4} \log(n) \log(k)\right)$ *adaptive rounds and* $\mathcal{O}\left(\varepsilon^{-5} n \log(n) \log(k)\right)$ *queries with a probability of* $1 - 1/n$. *It satisfies that* $\max\{f(A'_1), f(A'_2)\} \geq (1/4 - \varepsilon) f(O)$.

**Theorem 4.2.** *With input* $(f, k, \varepsilon)$ *such that* $\ell = \mathcal{O}\left(\varepsilon^{-1}\right) \geq \frac{4}{e\varepsilon}$ *and* $k \geq \frac{(2-\varepsilon)^2 \ell}{e \varepsilon \ell - 4}$, PITG *(Alg. 11) returns* $G$ *such that* $\mathbb{E}[f(G)] \geq (1/e - \varepsilon) f(O)$ *with* $\mathcal{O}\left(\varepsilon^{-5} \log(n) \log(k)\right)$ *adaptive rounds and* $\mathcal{O}\left(\varepsilon^{-6} n \log(n) \log(k)\right)$ *queries with a probability of* $1 - \mathcal{O}\left(1/(\varepsilon n)\right)$.

The remainder of this section is organized as follows. In Section 4.1, we present the key subroutines that form the building blocks of PIG. Section 4.2 then provides a high-level overview of the algorithm, and introduces several critical properties that must be preserved throughout the process.

## 4.1. Subroutines for PIG

**DISTRIBUTE** (Alg. 3) constructs pairwise disjoint subsets $\{\mathcal{V}_l : l \in [\ell]\}$ from candidate pools $\{V_l : l \in [\ell]\}$, preparing for the subsequent threshold sampling phase. This crucial preprocessing step ensures that all elements added to the solution maintain disjointness across different solution sets. We provide its theoretical guarantee in Lemma 4.3.

**Lemma 4.3.** *With input* $\{V_l\}_{l \in [\ell]}$, *where* $|V_l| \geq 2\ell$ *for each* $l \in [\ell]$, DISTRIBUTE *returns* $\ell$ *pairwise disjoint sets* $\{\mathcal{V}_l\}_{l \in [\ell]}$ *s.t.* $\mathcal{V}_l \subseteq V_l$ *and* $|\mathcal{V}_j| \geq \frac{|V_j|}{2\ell}$.

---

**Algorithm 3:** Return $\ell$ pairwise disjoint subsets where $|\mathcal{V}_j| \geq \frac{|V_j|}{2\ell}$ for any $j \in [\ell]$ if $|V_j| \geq 2\ell$

1 **Procedure** DISTRIBUTE $(\{V_l : l \in [\ell]\})$:
    **Input:** $V_1, V_2, \ldots, V_\ell \subseteq \mathcal{U}$
    **Initialize:** $\mathcal{V}_1, \mathcal{V}_2, \ldots, \mathcal{V}_\ell \leftarrow \emptyset, I \leftarrow [\ell]$
2     **for** $i \leftarrow 1$ *to* $\ell$ **do**
3         $j \leftarrow \arg\min_{j \in I} |V_j|$
4         $\mathcal{V}_j \leftarrow$ randomly select $\left\lfloor \frac{|V_j|}{\ell} \right\rfloor$ elements in
        $V_j \setminus \left( \bigcup_{l \in [\ell]} \mathcal{V}_j \right)$
5         $I \leftarrow I - j$
6     **return** $\{\mathcal{V}_l : l \in [\ell]\}$

---

**PREFIX-SELECTION** (Alg. 4) serves as the fundamental building block for threshold sampling procedure. It achieves two critical objectives simultaneously: 1) it identifies a prefix $T_{i^*}$ that provides sufficient marginal gain to the solution, and 2) with probability at least $1/2$, it guarantees that a constant fraction of candidate elements have low marginal gain $(< \tau)$ relative to the augmented solution.

Our implementation of PREFIX-SELECTION follows Lines

---

**Algorithm 4:** Select a prefix of $\mathcal{V}$ s.t. its average marginal gain is greater than $(1 - \varepsilon)\tau$, and with a probability of $1/2$, more than an $\varepsilon/2$-fraction of $\mathcal{V}$ has a marginal gain less than $\tau$ relative to the prefix.

1 **Procedure** PREFIX-SELECTION $(f, \mathcal{V}, s, \tau, \varepsilon)$:
    **Input:** evaluation oracle $f : 2^{\mathcal{U}} \to \mathbb{R}_{\geq 0}$, maximum
        size $s$, threshold $\tau$, error $\varepsilon$, candidate pool $\mathcal{V}$
        where $\Delta(x \mid \emptyset) \geq \tau$ for any $x \in \mathcal{V}$
    **Initialize:** $B[1 : s] \leftarrow [\textbf{none}, \ldots, \textbf{none}]$
2     $\mathcal{V} \leftarrow \{v_1, v_2, \ldots\} \leftarrow$ **random-permutation**$(\mathcal{V})$
3     **for** $i \leftarrow 1$ *to* $s$ *in parallel* **do**
4         $T_{i-1} \leftarrow \{v_1, \ldots, v_{i-1}\}$
5         **if** $\Delta(v_i \mid T_{i-1}) \geq \tau$ **then** $B[i] \leftarrow$ **true**
6         **else if** $\Delta(v_i \mid T_{i-1}) < 0$ **then** $B[i] \leftarrow$ **false**
7     $i^* \leftarrow \max\{i : \#\textbf{true}\text{s in } B[1 : i] \geq (1 - \varepsilon)i\}$
8     **return** $i^*, B$

---

8-15 of THRESHSEQ (Chen & Kuhnle, 2024), and consequently inherits similar theoretical guarantees (Lemma 4 and 5) as follows.

**Lemma 4.4.** *In* PREFIX-SELECTION, *given* $\mathcal{V}$ *after* **random-permutation** *in Line* 2, *let* $D_i = \{x \in \mathcal{V} : \Delta(x \mid T_i) < \tau\}$. *It holds that* $|D_0| = 0, |D_{|\mathcal{V}|}| = |\mathcal{V}|$, *and* $|D_{i-1}| \leq |D_i|$.

**Lemma 4.5.** *In* PREFIX-SELECTION, *following the definition of* $D_i$ *in Lemma 4.4, let* $t = \min\{i : |D_i| \geq \varepsilon|\mathcal{V}|/2\}$. *It holds that* $\mathbb{P}[i^* < \min\{s, t\}] \leq 1/2$.

---

**Algorithm 5:** Update candidate set $V$ with threshold value $\tau$

1 **Procedure** UPDATE $(f, V, \tau, \varepsilon)$:
    **Input:** evaluation oracle $f : 2^{\mathcal{U}} \to \mathbb{R}_{\geq 0}$, candidate
        set $V$, threshold value $\tau$, error $\varepsilon$
2     **for** $j \leftarrow 1$ *to* $\ell$ *in parallel* **do**
3         $V \leftarrow \{x \in V : \Delta(x \mid \emptyset) \geq \tau\}$
4         **while** $|V| = 0$ **do**
5              $\tau \leftarrow (1 - \varepsilon)\tau$
6              $V \leftarrow \{x \in \mathcal{U} : \Delta(x \mid \emptyset) \geq \tau\}$
7     **return** $V, \tau$

---

**UPDATE** maintains the candidate set $V$ for a solution using threshold $\tau$. If $V$ becomes empty, it decrease the threshold and regenerate $V$ until it is not empty. This is a common component of threshold sampling algorithms.

## 4.2. Algorithm Overview

PIG synthesizes threshold sampling techniques with interlaced solution construction to achieve both parallel efficiency and strong approximation guarantees. It begins by initializing $\ell$ empty solutions $\{A_j : j \in [\ell]\}$, with corre-

sponding threshold values set to the maximum singleton marginal gain $M = \max_{x \in \mathcal{U}} \Delta(x \mid \emptyset)$.

At each iteration, PIG dynamically switches between two distinct operational modes based on candidate set sizes. When any candidate set $V_j$ (maintained by UPDATE, Alg. 5) contains fewer than $2\ell$ elements, the algorithm enters an alternating addition phase where single elements are sequentially distributed across solutions. This preserves the alternating selection property crucial for maintaining approximation guarantees while ensuring progress when candidate pools are limited.

For cases where all candidate sets contain sufficient number of elements ($|V_j| \geq 2\ell$ for each $j \in [\ell]$), PIG employs an efficient batch processing approach. First, DISTRIBUTE constructs pairwise disjoint candidate subsets, enabling parallel processing while maintaining solution quality. PREFIX-SELECTION then simultaneously looks for blocks with high-marginal-gain elements. At last, the block size and the elements added to each solution are carefully chosen to maintain both solution quality parity and the equivalent alternating selection effect.

This dual-mode architecture combines the strengths of threshold sampling and interlaced greedy methods. The threshold sampling components (UPDATE and PREFIX-SELECTION) ensure efficient element filtering and geometric threshold reduction, while the interlaced construction maintains the crucial alternating marginal gain properties. This allows PIG to achieve sublinear adaptivity without compromising its approximation guarantee.

Below, we further analyze three fundamental properties that the algorithm must maintain throughout execution.

### 4.2.1. MAINTAINING ALTERNATING ADDITIONS DURING PARALLEL ALGORITHMS

This property is crucial to interlaced greedy variants introduced in prior sections. Below, we demonstrate that PIG preserves this property.

During an iteration of the while loop in Alg. 6, after updating the candidate sets in Lines 3-5, two scenarios arise. In the first scenario, there exists a candidate set satisfies $|V_j| < 2\ell$, Lines 7-14 are executed, and elements are appended to solutions one at a time in turn. In this case, the alternating property is maintained immediately.

In the second scenario, Lines 16-24 are executed. Here, a block of elements with average marginal gain approximately exceeding $\tau_j$ is added to each solution $A_j$. These blocks $S_j$ are of the same size $i^*$ (Line 21) selected from $\mathcal{V}_j$, and guaranteed to be pairwise disjoint by Lemma 4.3 (for DISTRIBUTE, Alg. 3). Crucially, threshold values $\tau_j$ remain unchanged during this step. While a small fraction

of elements in the blocks may have marginal gains below $\tau_j$, the process retains the alternating property at a structural level: the uniform block sizes, and disjoint selection mimic the alternating addition of elements, even when processing multiple elements in parallel.

### 4.2.2. ENSURING SUBLINEAR ADAPTIVITY THROUGH THRESHOLD SAMPLING

The core mechanism for achieving sublinear adaptivity lies in iteratively reducing the pool of high-quality candidate elements (those with marginal gains above the threshold) by a constant factor within a constant number of adaptive rounds. This progressive reduction ensures efficient convergence.

At every iteration of the while loop in Alg. 6, after updating the candidate sets, if there exists $V_j$ such that $|V_j| < 2\ell$, the following occurs after the for loop (Lines 7-13): If the threshold $\tau_j$ remains unchanged, one element from $V_j$ is added to the solution. If $\tau_j$ is reduced, $V_j$ is repopulated with high-quality elements. This implies that a $1/(2\ell)$-fraction of $V_j$ is filtered out after per iteration, or even further, it becomes empty and the threshold value is updated.

In the second case, where Lines 16-24 are executed, the algorithm employs PREFIX-SELECTION (Alg. 4) in Line 18, inspired by THRESHSEQ (Chen & Kuhnle, 2024). Then, the smallest prefix size $i^*$ is selected in Line 19. For the solution where its corresponding call to PREFIX-SELECTION returns $i^*$, the entire prefix with size $i^*$ is added to it. This ensures that a constant fraction of elements in $\mathcal{V}_j$ can be filtered out by Lemma 4.5 with probability at least $1/2$. Moreover, Lemma 4.3 guarantees that $|\mathcal{V}_j| \geq \frac{1}{\ell}|V_j|$ for each candidate set. As a result, with constant probability, at least one candidate set will filter out a constant fraction of the elements.

### 4.2.3. ENSURING MOST ADDED ELEMENTS SIGNIFICANTLY CONTRIBUTE TO THE SOLUTIONS

In THRESHSEQ, the selection of a *good prefix* inherently ensures this property immediately. However, when interlacing $\ell$ threshold sampling processes, prefix sizes selected in Line 18 by each solution may vary. To preserve the alternating addition property introduced in Section 4.2.1, subsets of equal size are selected instead of variable-length good prefixes. This raises the question: *How can a good subset be derived from a good prefix?* The solution lies in Line 21 of Alg. 6.

For any $j \in I$, if $i_j^* = i^*$, $S_j$ is directly the good prefix $\mathcal{V}_j[1 : i_j^*]$. Otherwise, if $i_j^* \geq i^*$, $i^*$ elements are selected from $\mathcal{V}_j[1 : i_j^*]$ in three sequential passes until the size limit is reached:

**First pass**: Iterate through the prefix, selecting those with marginal gains strictly greater than $\tau_j$ (marked as true in

---

**Algorithm 6:** A highly parallelized algorithm with $\mathcal{O}\left(\ell^2\varepsilon^{-2}\log(n)\log\left(\frac{M}{\tau_{\min}}\right)\right)$ adaptivity and $\mathcal{O}\left(\ell^3\varepsilon^{-2}n\log(n)\log\left(\frac{M}{\tau_{\min}}\right)\right)$ query complexity.

1 **Procedure** PARALLELINTERLACEGREEDY $(f, m, \ell, \tau_{\min}, \varepsilon)$:

   **Input:** evaluation oracle $f : 2^{\mathcal{U}} \to \mathbb{R}_{\geq 0}$, constraint $m$, constant $\ell$, minimum threshold value $\tau_{\min}$, error $\varepsilon$

   **Initialize :** $M \leftarrow \max_{x \in \mathcal{U}} \Delta(\{x\} \mid \emptyset), I = [\ell], m_0 \leftarrow m, A_j \leftarrow A'_j \leftarrow \emptyset, \tau_j \leftarrow M, V_j \leftarrow \mathcal{U}, \forall j \in [\ell]$

2   **while** $I \neq \emptyset$ *and* $m_0 > 0$ **do**

3     **for** $j \in I$ *in parallel* **do** /* `Update candidate sets with high-quality elements` */

4       $\{V_j, \tau_j\} \leftarrow$ UPDATE$(f_{A_j}\!\upharpoonright_{\mathcal{U}\setminus\left(\bigcup_{l\in[\ell]} A_l\right)}, V_j \setminus \left(\bigcup_{l\in[\ell]}\right) A_l, \tau_j, \varepsilon)$

5       **if** $\tau_j < \tau_{\min}$ **then** $I \leftarrow I - j$

6     **if** $\exists i \in I$ *s.t.* $|V_i| < 2\ell$ **then** /* `Add 1 element to each solution alternately` */

7       **for** $j \in I$ *in sequence* **do**

8         **if** $|V_j| = 0$ **then** $\{V_j, \tau_j\} \leftarrow$ UPDATE$(f_{A_j}\!\upharpoonright_{\mathcal{U}\setminus\left(\bigcup_{l\in[\ell]} A_l\right)}, V_j, \tau_j, \varepsilon)$

9         **if** $\tau_j < \tau_{\min}$ **then** $I \leftarrow I - j$

10         **else**

11           $x_j \leftarrow$ randomly select one element from $V_j$

12           $A_j \leftarrow A_j + x_j, A'_j \leftarrow A'_j + x_j$

13           $V_l \leftarrow V_l - x_j, \forall l \in [\ell]$

14       $m_0 \leftarrow m_0 - 1$

15     **else** /* `Add an equal number of elements to each solution` */

16       $\{\mathcal{V}_l : l \in I\} \leftarrow$ DISTRIBUTE$(\{V_l : l \in I\})$ // `Create pairwise disjoint candidate sets`

17       $s \leftarrow \min\{m_0, \min\{|\mathcal{V}_l| : l \in I\}\}$

18       **for** $j \in I$ *in parallel* **do** $i^*_j, B_j \leftarrow$ PREFIX-SELECTION$(f_{A_j}, \mathcal{V}_j, s, \tau_j, \varepsilon)$

19       $i^* \leftarrow \min\{i^*_1, \ldots, i^*_\ell\}$

20       **for** $j \leftarrow 1$ *to* $\ell$ *in parallel* **do** /* `Add i* high-quality elements to each set` */

21         $S_j \leftarrow$ select $i^*$ elements from $\mathcal{V}_j[1 : i^*_l]$ in three passes, prioritizing $B_j[i] =$ **true**, then $B_j[i] =$ **none**, and finally $B_j[i] =$ **false**

22         $S'_j \leftarrow S_j \cap \{v_i \in \mathcal{V}_j : B_j[i] \neq$ **false**$\}$

23         $A_j \leftarrow A_j \cup S_j, A'_j \leftarrow A'_j \cup S'_j$

24       $m_0 \leftarrow m_0 - i*$

25   **return** $\{A'_l : l \in [\ell]\}$

---

$B_j$).

**Second pass**: From the remaining elements in the prefix, select those with marginal gains between 0 and $\tau_j$ (marked as none in $B_j$).

**Third pass**: Fill any remaining slots with remaining elements from the prefix (marked as false in $B_j$).

This approach, combined with submodularity, ensures that any element marked as **true** in the selected subset has a marginal gain greater than $\tau_j$. By prioritizing the addition of these **true** elements, the selected subset remain high-quality while adhering to the alternating addition framework.

## 5. Empirical Evaluation

To evaluate the effectiveness of our algorithms, we conducted experiments on 2 applications of SM-GEN, comparing its performance to 5 baseline algorithms. We measured the objective value (normalized by ATG (Chen & Kuhnle, 2024)) achieved by each algorithm, the number of queries made, and the number of adaptive rounds required. The results showed that our algorithm achieved competitive objective value, number of queries and adaptive rounds compared to nearly linear time algorithms. [4]

**Applications and Datasets.** The algorithms were evaluated on 2 applications: Maximum Cut (`maxcut`) and Revenue Maximization (`revmax`), with er ($n = 99,997$), a synthetic random graph, web-Google ($n = 875,713$), musae-github ($n = 37,700$) and twitch-gamers ($n = 168,114$) datasets, the rest of which are real-world social network datasets from Stanford Large Network Dataset Collection (Leskovec & Krevl, 2014). See Appendix H.1 and H.2 for more details.

---

[4]Our code is available at https://gitlab.com/luciacyx/size-constraints-parallel-algorithms.git.

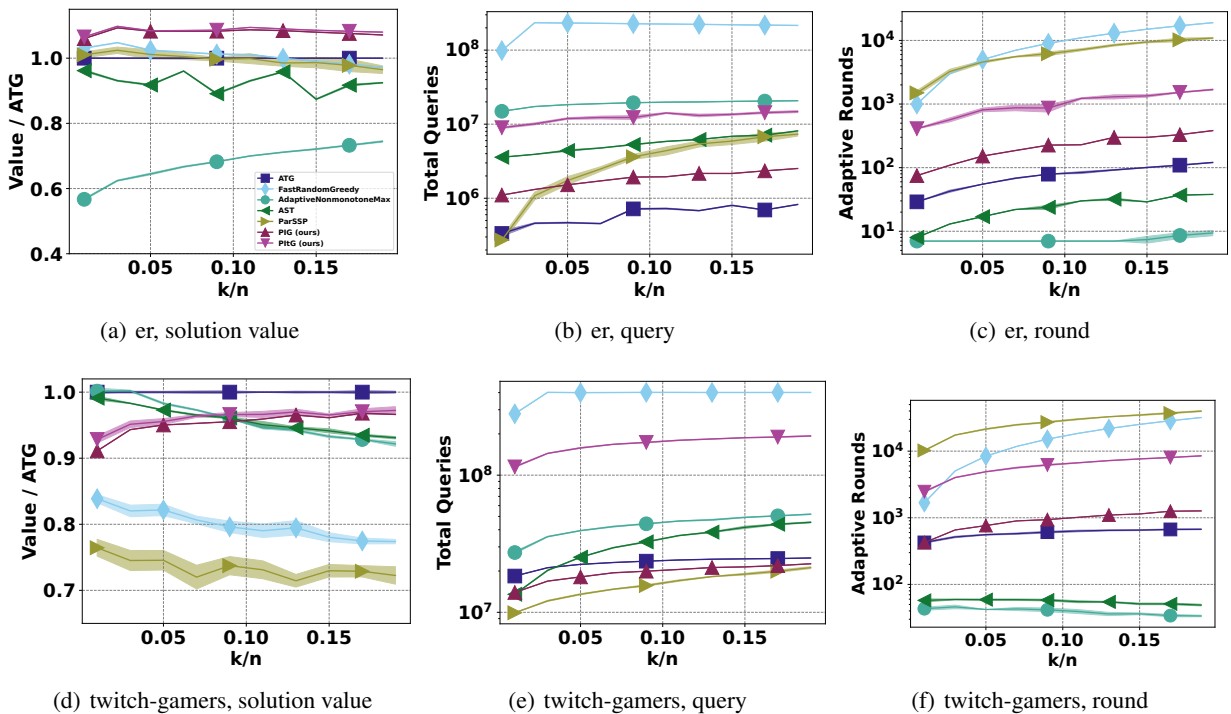

(a) er, solution value

(b) er, query

(c) er, round

(d) twitch-gamers, solution value

(e) twitch-gamers, query

(f) twitch-gamers, round

*Figure 2.* Results for `maxcut` on er with $n = 99,997$, and `revmax` on twitch-gamers with $n = 168,114$.

We provide the result of er and twitch-gamers in the main paper, while the other results can be found in Appendix H.3.

**Baselines and their Setups.** We compare our algorithms with FASTRANDOMGREEDY (Buchbinder et al., 2017), ADAPTIVENONMONOTONEMAX (Fahrbach et al., 2023), AST (Chen & Kuhnle, 2024), ATG (Chen & Kuhnle, 2024), and PARSSP (Cui et al., 2023) using binary search. For those algorithms who required an UNCONSTRAINEDMAX algorithm, a random subset was employed instead. For all algorithms, the accuracy parameter $\varepsilon$ and the failure probability parameter $\delta$ were both set to $0.1$. Whenever smaller $\varepsilon$ and $\delta$ values were specified by the algorithm, we substituted them with the input values. FASTRANDOMGREEDY samples each element in the ground set $\mathcal{U}$ with a probability $p = 8k^{-1}\varepsilon^{-2}\log(2\varepsilon^{-1})$. If $p > 1$, RANDOMGREEDY in Buchbinder et al. (2014) was implemented instead. As for THRESHOLD-SAMPLING in ADAPTIVENONMONOTONE-MAX, 100 samples were used to estimate an indicator. In the implementation of PITG, $\ell$ was set to 5. All randomized algorithms were repeated with 5 runs, and the mean is reported. The standard deviation is represented by a shaded region in the plots.

**Overview of Results.** On the er dataset (Fig. 2(a)), PIG and PITG achieve highest objective values, followed by ATG, FASTRANDOMGREEDY and PARSSP. For the twitch-gamers dataset (Fig. 2(d)), PIG and PITG demonstrate greater robustness, particularly for larger values of $k$.

In terms of query complexity, PARSSP exhibits the highest query complexity ($\mathcal{O}\left(n\log^2(n)\log(k)\right)$), followed by our proposed algorithms ($\mathcal{O}\left(n\log(n)\log(k)\right)$), compared to other algorithms ($\mathcal{O}\left(n\log(k)\right)$). Nevertheless, PIG achieves second-best performance across both datasets. The top three performing algorithms consistently include PIG, ATG and PARSSP. PITG is slightly better than ADAPTIVENONMONOTONEMAX on er, and both are more efficient than FASTRANDOMGREEDY on the two datasets. FASTRANDOMGREEDY's high query count arises from its requirement for a large number of samples (specifically, $8nk^{-1}\varepsilon^{-1}\log(2\varepsilon^{-1})$) at every iteration. When $8k^{-1}\varepsilon^{-1}\log(2\varepsilon^{-1}) \geq 1$, it defaults to executing RANDOM-GREEDY instead, which incurs $\mathcal{O}(nk)$ queries.

Regarding adaptive rounds (Fig. 2(c) and 2(e)), the results align with theoretical guarantees. ADAPTIVENONMONO-TONEMAX and AST operate with $\mathcal{O}(\log(n))$ adaptivity and achieve the best performance. They are followed by ATG, PIG, and PITG which all achieve $\mathcal{O}(\log(n)\log(k))$ adaptivity.

PARSSP exhibits an interesting trade-off: while its binary search procedure benefits query complexity, it significantly increases adaptive rounds, performing worse than even FAS-TRANDOMGREEDY (which requires $k$ adaptive rounds) on the twitch-gamers dataset.

## Impact Statement

This paper presents work whose goal is to advance the field of Machine Learning. There are many potential societal consequences of our work, none which we feel must be specifically highlighted here.

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

# A. Technical Lemmata

**Lemma A.1.**

$$1 - \frac{1}{x} \leq \log(x) \leq x - 1, \qquad\qquad \forall x > 0$$

$$1 - \frac{1}{x+1} \geq e^{-\frac{1}{x}}, \qquad\qquad \forall x \in \mathbb{R}$$

$$(1 - x)^{y-1} \geq e^{-xy}, \qquad\qquad \forall xy \leq 1$$

**Lemma A.2** (Chernoff bounds (Mitzenmacher & Upfal, 2017))**.** Suppose $X_1, \dots, X_n$ are independent binary random variables such that $\mathbb{P}[X_i = 1] = p_i$. Let $\mu = \sum_{i=1}^n p_i$, and $X = \sum_{i=1}^n X_i$. Then for any $\delta \geq 0$, we have

$$\mathbb{P}[X \geq (1+\delta)\mu] \leq e^{-\frac{\delta^2 \mu}{2+\delta}}. \tag{6}$$

Moreover, for any $0 \leq \delta \leq 1$, we have

$$\mathbb{P}[X \leq (1-\delta)\mu] \leq e^{-\frac{\delta^2 \mu}{2}}. \tag{7}$$

**Lemma A.3** (Chen et al. (2021))**.** Suppose there is a sequence of $n$ Bernoulli trials: $X_1, X_2, \ldots, X_n$, where the success probability of $X_i$ depends on the results of the preceding trials $X_1, \ldots, X_{i-1}$. Suppose it holds that

$$\mathbb{P}[X_i = 1 | X_1 = x_1, X_2 = x_2, \ldots, X_{i-1} = x_{i-1}] \geq \eta,$$

where $\eta > 0$ is a constant and $x_1, \ldots, x_{i-1}$ are arbitrary.

Then, if $Y_1, \ldots, Y_n$ are independent Bernoulli trials, each with probability $\eta$ of success, then

$$\mathbb{P}\left[\sum_{i=1}^n X_i \leq b\right] \leq \mathbb{P}\left[\sum_{i=1}^n Y_i \leq b\right],$$

where $b$ is an arbitrary integer.

Moreover, let $A$ be the first occurrence of success in sequence $X_i$. Then,

$$\mathbb{E}[A] \leq 1/\eta.$$

# B. Propositions on Submodularity

**Proposition B.1.** *Let $\{A_1, A_2, \ldots, A_m\}$ be $m$ pairwise disjoint subsets of $\mathcal{U}$, and $B \in \mathcal{U}$. For any submodular function $f : 2^{\mathcal{U}} \to \mathbb{R}_{\geq 0}$, it holds that*

$$\textit{1)}\ \sum_{i \in [m]} \Delta(A_i \mid B) \geq \Delta\left(\bigcup_{i \in [m]} A_i \,\middle|\, B\right),$$

$$\textit{2)}\ \sum_{i \in [m]} f(B \cup A_i) \geq (m-1)f(B).$$

**Proposition B.2.** *Let $A = \{a_1, \ldots, a_m\}$ and $A_i = \{a_1, \ldots, a_i\}$ for all $i \in [m]$. For any submodular function $f : 2^{\mathcal{U}} \to \mathbb{R}_{\geq 0}$, let $B = \underset{B \subseteq A, |B| = m-1}{\arg\max} \sum_{a_i \in B} \Delta(a_i \mid A_{i-1})$. It holds that $f(B) \geq \left(1 - \frac{1}{m}\right) f(A)$.*

**Proposition B.3.** *For any submodular function $f : 2^{\mathcal{U}} \to \mathbb{R}_{\geq 0}$, let $O_1 = \arg\max_{S \subseteq \mathcal{U}, |S| \leq k_1} f(S)$ and $O_2 = \arg\max_{S \subseteq \mathcal{U}, |S| \leq k_2} f(S)$. It holds that*

$$f(O_1) \geq \frac{k_1}{k_2} f(O_2).$$

## C. Pseudocode and Theoretical Guarantees of INTERPOLATEDGREEDY (Chen & Kuhnle, 2023)

In this section, we provide the original greedy version of INTERPOLATEDGREEDY (Chen & Kuhnle, 2023) with its theoretical guarantees.

---

**Algorithm 7:** INTERPOLATEDGREEDY$(f, k, \varepsilon)$: An $1/(e + \varepsilon)$-approximation algorithm for SMCC

**Input:** oracle $f : 2^{\mathcal{U}} \to \mathbb{R}_{\geq 0}$, constraint $k$, error $\varepsilon$

**Initialize:** $\ell \leftarrow \frac{2e}{\varepsilon} + 1$, $G_0 \leftarrow \emptyset$

1 **for** $m \leftarrow 1$ *to* $\ell$ **do**
2      $\{a_1, \ldots, a_\ell\} \leftarrow$ top $\ell$ elements in $\mathcal{U} \setminus G_{m-1}$ with respect to marginal gains on $G_{m-1}$
3      **for** $u \leftarrow 0$ *to* $\ell$ *in parallel* **do**
4          **if** $u = 0$ **then** $A_{u,l} \leftarrow G \cup \{a_l\}$, for all $1 \leq l \leq \ell$
5          **else** $A_{u,l} \leftarrow G \cup \{a_u\}$, for all $1 \leq l \leq \ell$
6          **for** $j \leftarrow 1$ *to* $k/\ell - 1$ **do**
7              **for** $i \leftarrow 1$ *to* $\ell$ **do**
8                  $x_{j,i} \leftarrow \arg \max_{x \in \mathcal{U} \setminus (\bigcup_{l=1}^{\ell} A_{u,l})} \Delta\left(x \mid A_{u,i}\right)$
9                  $A_{u,i} \leftarrow A_{u,i} \cup \{x_{j,i}\}$
10      $G_m \leftarrow$ a random set in $\{A_{u,i} : 1 \leq i \leq \ell, 0 \leq u \leq \ell\}$
11 **return** $G_\ell$

---

**Theorem C.1.** *Let $\varepsilon \geq 0$, and $(f, k)$ be an instance of* SMCC*, with optimal solution value OPT. Algorithm* INTERPOLATEDGREEDY *outputs a set $G_\ell$ with $\mathcal{O}\left(\varepsilon^{-2} kn\right)$ queries such that $OPT \leq (e + \varepsilon)\mathbb{E}\left[f\left(G_\ell\right)\right]$ with probability $(\ell + 1)^{-\ell}$, where $\ell = \frac{2e}{\varepsilon} + 1$.*

## D. Analysis of Simplified INTERLACEGREEDY (Alg. 8)

In this section, we present a detailed approximation analysis of the simplified INTERLACEGREEDY (Alg. 8), demonstrating that while the algorithm removes the initial guessing step from its original formulation, it maintains the same theoretical approximation guarantees.

---

**Algorithm 8:** A deterministic $1/4$-approximation algorithm with $\mathcal{O}(nk)$ queries.

**Input:** evaluation oracle $f : 2^{\mathcal{U}} \to \mathbb{R}_{\geq 0}$, constraint $k$

**Initialize:** $A \leftarrow B \leftarrow \emptyset$, add $2k$ dummy elements to the ground set

1 **for** $i \leftarrow 1$ *to* $k$ **do**
2      $a \leftarrow \arg \max_{x \in \mathcal{U} \setminus (A \cup B)} \Delta\left(x \mid A\right)$
3      $A \leftarrow A + a$
4      $b \leftarrow \arg \max_{x \in \mathcal{U} \setminus (A \cup B)} \Delta\left(x \mid B\right)$
5      $B \leftarrow B + b$
6 **return** $S \leftarrow \arg \max\{f\left(A\right), f\left(B\right)\}$

---

**Theorem D.1.** *With input instance $(f, k)$, Alg. 8 returns a set $S$ with $\mathcal{O}(kn)$ queries such that $f(S) \geq 1/4 f(O)$.*

*Proof of Theorem D.1.* **Notation.** Let $a_i$ be the $i$-th element added to $A$, and $A_i$ be the set containing the first $i$ elements of $A$. Similarly, define $b_i$ and $B_i$ for the solution $B$.

Since the two solutions $A_k$ and $B_k$ are disjoint, by submodularity and non-negativity,

$$f(O) \leq f(O \cup A_k) + f(O \cup B_k).$$

Let $i^* = \max\{i \in [k] : A_i \subseteq O\}$ and $j^* = \max\{j \in [k] : B_j \subseteq O\}$. If either $i^* = k$ or $j^* = k$, then $f(S) = f(O)$. In the following, we consider $i^* < k$ and $j^* < k$ and discuss two cases of the relationship between $i^*$ and $j^*$ (Fig. 3).

**Case 1:** $0 \leq i^* \leq j^* < k$; **Fig. 3(a).** First, we bound $f(O \cup A_k)$. Consider the set $\tilde{O} = O \setminus (A_k \cup B_{i^*})$. Obviously, it holds that $|\tilde{O}| \leq k - i^*$. Then, order $\tilde{O}$ as $\{o_1, o_2, \ldots\}$ such that $o_i \notin B_{i+i^*-1}$, for all $1 \leq i \leq |\tilde{O}|$. Thus, by the greedy

selection step in Line 2, it holds that $\Delta\left(a_{i+i^*} \mid A_{i+i^*-1}\right) \geq \Delta\left(o_i \mid A_{i+i^*-1}\right)$ for all $1 \leq i \leq |\tilde{O}|$. Then,

$$
f\left(O \cup A_k\right) - f\left(A_k\right) \leq \Delta\left(B_{i^*} \mid A_k\right) + \Delta\left(\tilde{O} \mid A_k\right)
$$

$$
\leq f\left(B_{i^*}\right) + \sum_{i=1}^{|\tilde{O}|} \Delta\left(o_i \mid A_k\right)
$$

$$
\leq f\left(B_{i^*}\right) + \sum_{i=1}^{|\tilde{O}|} \Delta\left(o_i \mid A_{i+i^*-1}\right)
$$

$$
\leq f\left(B_{i^*}\right) + \sum_{i=i^*+1}^{k} \Delta\left(a_i \mid A_{i-1}\right) = f\left(B_{i^*}\right) + f\left(A_k\right) - f\left(A_{i^*}\right),
$$

where the first three inequalities follow from submodularity; and the last inequality follows from $\Delta\left(a_{i+i^*} \mid A_{i+i^*-1}\right) \geq \Delta\left(o_i \mid A_{i+i^*-1}\right)$ for all $1 \leq i \leq |\tilde{O}|$, and $\Delta\left(a_i \mid A_{i-1}\right) \geq 0$ for all $i \in [k]$.

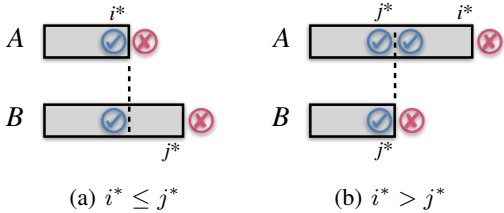

(a) $i^* \leq j^*$      (b) $i^* > j^*$

*Figure 3.* This figure depicts the components of solution sets $A$ and $B$ in Alg. 8. The black rectangle highlights a sequence of consecutive elements from $O$ that were added to the solution at the initial. Red circles with a cross mark signifies the first element in $A$ or $B$ that is outside $O$.

Next, we bound $f\left(O \cup B_k\right)$. Consider the set $\tilde{O} = O \setminus \left(A_{i^*} \cup B_k\right)$. Obviously, it holds that $|\tilde{O}| \leq k - i^*$. Since $i^* = \max\{i \in [k] : A_i \subseteq O\}$, we know that $a_{i^*+1} \notin O$. Thus, we can order $|\tilde{O}|$ as $\{o_1, o_2, \dots\}$ such that $o_i \notin A_{i+i^*}$ for all $1 \leq i \leq |\tilde{O}|$. Then, by the greedy selection step in Line 4, it holds that $\Delta\left(b_{i+i^*} \mid B_{i+i^*-1}\right) \geq \Delta\left(o_i \mid B_{i+i^*-1}\right)$ for all $1 \leq i \leq |\tilde{O}|$. Following the analysis for $f\left(O \cup A\right)$, we get

$$
f\left(O \cup B_k\right) - f\left(B_k\right) \leq \Delta\left(A_{i^*} \mid B_k\right) + \Delta\left(\tilde{O} \mid B_k\right)
$$

$$
\leq f\left(A_{i^*}\right) + \sum_{i=1}^{|\tilde{O}|} \Delta\left(o_i \mid B_k\right)
$$

$$
\leq f\left(A_{i^*}\right) + \sum_{i=1}^{|\tilde{O}|} \Delta\left(o_i \mid B_{i+i^*-1}\right)
$$

$$
\leq f\left(A_{i^*}\right) + \sum_{i=i^*+1}^{k} \Delta\left(b_i \mid B_{i-1}\right) = f\left(A_{i^*}\right) + f\left(B_k\right) - f\left(B_{i^*}\right).
$$

**Case 2:** $0 \leq j^* < i^* < k$**; Fig. 3(b).** First, we bound $f\left(O \cup A_k\right)$. Consider the set $\tilde{O} = O \setminus \left(A_k \cup B_{j^*}\right)$, where $|\tilde{O}| \leq k - j^* - 1$. By the definition of $j^*$, we know that $b_{j^*+1} \notin O$. Thus, we can order $\tilde{O}$ as $\{o_1, o_2, \dots\}$ such that $o_i \notin B_{i+j^*}$ for all $1 \leq i \leq |\tilde{O}|$. Then, by the greedy selection step in Line 2, it holds that $\Delta\left(a_{i+j^*+1} \mid A_{i+j^*}\right) \geq \Delta\left(o_i \mid A_{i+j^*}\right)$ for all $1 \leq i \leq |\tilde{O}|$. Following the above analysis, we get

$$
f\left(O \cup A_k\right) - f\left(A_k\right) \leq \Delta\left(B_{j^*} \mid A_k\right) + \Delta\left(\tilde{O} \mid A_k\right)
$$

$$
\leq f\left(B_{j^*}\right) + \sum_{i=1}^{|\tilde{O}|} \Delta\left(o_i \mid A_k\right)
$$

$$\leq f\left(B_{j^*}\right) + \sum_{i=1}^{|\tilde{O}|} \Delta\left(o_i \mid A_{i+j^*}\right)$$

$$\leq f\left(B_{j^*}\right) + \sum_{i=j^*+2}^{k} \Delta\left(a_i \mid A_{i-1}\right) = f\left(B_{j^*}\right) + f\left(A_k\right) - f\left(A_{j^*+1}\right).$$

Next, we bound $f\left(O \cup B_k\right)$. Consider the set $\tilde{O} = O \backslash \left(A_{j^*+1} \cup B_k\right)$, where $|\tilde{O}| \leq k - j^* - 1$. Then, order $\tilde{O}$ as $\{o_1, o_2, \ldots\}$ such that $o_i \notin A_{i+j^*}$ for all $1 \leq i \leq |\tilde{O}|$. By the greedy selection step in Line 4, it holds that $\Delta\left(b_{i+j^*} \mid B_{i+j^*-1}\right) \geq \Delta\left(o_i \mid B_{i+j^*-1}\right)$. Then,

$$f\left(O \cup B_k\right) - f\left(B_k\right) \leq \Delta\left(A_{j^*+1} \mid B_k\right) + \Delta\left(\tilde{O} \mid B_k\right)$$

$$\leq f\left(A_{j^*+1}\right) + \sum_{i=1}^{|\tilde{O}|} \Delta\left(o_i \mid B_k\right)$$

$$\leq f\left(A_{j^*+1}\right) + \sum_{i=1}^{|\tilde{O}|} \Delta\left(o_i \mid B_{i+j^*-1}\right)$$

$$\leq f\left(A_{j^*+1}\right) + \sum_{i=j^*+1}^{k} \Delta\left(b_i \mid B_{i-1}\right) = f\left(A_{j^*+1}\right) + f\left(B_k\right) - f\left(B_{j^*}\right).$$

Therefore, in both cases, it holds that

$$f\left(O\right) \leq f\left(O \cup A_k\right) + f\left(O \cup B_k\right) \leq 2\left(f\left(A_k\right) + f\left(B_k\right)\right) \leq 4f\left(S\right).$$

$\square$

# E. Analysis of Simplified INTERPOLATEDGREEDY (Alg. 2, Section 3)

In the section, we first provide key Lemmata and their analysis in Appendix E.1 for the case when $k \bmod \ell = 0$. Then, we conclude with an analysis of approximation ratio in Appendix E.2.

## E.1. Proofs of Lemmata for Theorem 3.1

In what follows, we address the scenario where $k \bmod \ell = 0$ and Alg. 2 returns a solution with size exactly $k$.

**Notation.** Let $G_{i-1}$ be $G$ at the start of $i$-th iteration in Alg. 2, $A_l$ be the set at the end of this iteration, and $a_{l,j}$ be the $j$-th element added to $A_l$ during this iteration.

**Lemma 3.2.** Fix on $G_{i-1}$ for an iteration $i$ of the outer for loop in Alg. 2. Following the definition in Claim 3.1, it holds that

$$1)\ \Delta\left(A_l \mid G_{i-1}\right) \geq \Delta\left(O_l \mid A_l\right), \forall 1 \leq l \leq \ell,$$

$$2)\ \left(1 + \frac{1}{m}\right)\left(\Delta\left(A_{l_1} \mid G_{i-1}\right) + \Delta\left(A_{l_2} \mid G_{i-1}\right)\right)$$

$$\geq \Delta\left(O_{l_2} \mid A_{l_1}\right) + \Delta\left(O_{l_1} \mid A_{l_2}\right), \forall 1 \leq l_1 < l_2 \leq \ell.$$

*Proof of Lemma 3.2.* Recall that $A_{l,j}$ is $A_l$ after $j$-th element is added to $A_l$ at iteration $i$ of the outer for loop, and $c_l^* = \max\{c \in [m] : A_{l,c} \backslash G_{i-1} \subseteq O_l\}$.

First, we prove that the first inequality holds. For each $l \in [\ell]$, order the elements in $O_l$ as $\{o_1, o_2, \ldots\}$ such that $o_j \notin A_{l,j-1}$ for any $1 \leq j \leq |O_l|$. Since each $o_j$ is either in $A_l$ or not in any solution set, it remains in the candidate pool when $a_{l,j}$ is considered to be added to the solution. Therefore, it holds that

$$\Delta\left(a_{l,j} \mid A_{l,j-1}\right) \geq \Delta\left(o_j \mid A_{l,j-1}\right). \tag{8}$$

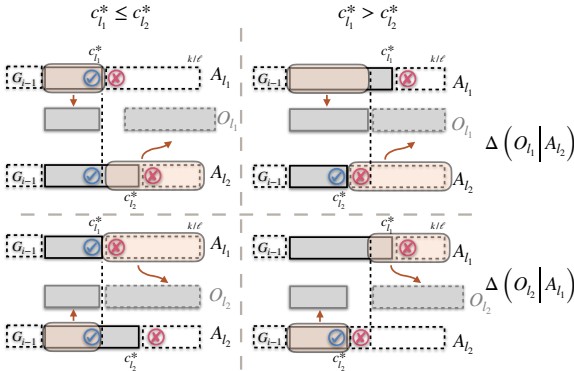

*Figure 4.* This figure depicts the components of the solution sets $A_{l_1}$ and $A_{l_2}$. A blue circle with a check mark represents an element in $O$, while a red circle with a cross mark represents an element outside of $O$. The grey rectangles indicate a sequence of consecutive elements in $O$. The pink rectangles indicate the corresponding elements used to bound $\Delta(O_{l_2} \mid A_{l_1})$ or $\Delta(O_{l_1} \mid A_{l_2})$. It is illustrated that $\Delta(O_{l_1} \mid A_{l_2}) + \Delta(O_{l_2} \mid A_{l_1}) \le \Delta(A_{l_1} \mid G_{i-1}) + \Delta(A_{l_2} \mid G_{i-1})$ under both cases.

Then,

$$\Delta(O_l \mid A_l) \le \sum_{o_j \in O_l} \Delta(o_j \mid A_l) \le \sum_{o_j \in O_l} \Delta(o_j \mid A_{l,j-1}) \le \sum_{j=1}^{m} \Delta(a_{l,j} \mid A_{l,j-1}) = \Delta(A_l \mid G_{j-1}),$$

where the first inequality follows from Proposition B.1, the second inequality follows from submodularity, and the last inequality follows from Inequality (8).

In the following, we prove that the second inequality holds. For any $1 \le l_1 \le l_2 \le \ell$, we analyze two cases of the relationship between $c_{l_1}^*$ and $c_{l_2}^*$ in the following.

**Case 1: $c_{l_1}^* \le c_{l_2}^*$; left half part in Fig. 4.**

First, we bound $\Delta(O_{l_1} \mid A_{l_2})$. Since $c_{l_1}^* \le m$, we know that the $(c_{l_1}^* + 1)$-th element in $A_{l_1} \setminus G_{i-1}$ is not in $O$. So, we can order the elements in $O_{l_1} \setminus A_{l_1,c_{l_1}^*}$ as $\{o_1, o_2, \ldots\}$ such that $o_j \notin A_{l_1,c_{l_1}^*+j+1}$. (Refer to the gray block with a dotted edge in the top left corner of Fig. 4 for $O_{l_1}$.) Since each $o_j$ is either added to $A_{l_1}$ or not in any solution set, it remains in the candidate pool when $a_{l_2,c_{l_1}^*+j}$ is considered to be added to $A_{l_2}$. Therefore, it holds that

$$\Delta\left(a_{l_2,c_{l_1}^*+j} \mid A_{l_2,c_{l_1}^*+j-1}\right) \ge \Delta\left(o_j \mid A_{l_2,c_{l_1}^*+j-1}\right), \forall 1 \le j \le m - c_{l_1}^*. \tag{9}$$

Then,

$$\Delta(O_{l_1} \mid A_{l_2}) \le \Delta\left(A_{l_1,c_{l_1}^*} \mid A_{l_2}\right) + \sum_{o_j \in O_{l_1} \setminus A_{l_1,c_{l_1}^*}} \Delta(o_j \mid A_{l_2}) \qquad \text{(Proposition B.1)}$$

$$\le \Delta\left(A_{l_1,c_{l_1}^*} \mid G_{i-1}\right) + \sum_{o_j \in O_{l_1} \setminus A_{l_1,c_{l_1}^*}} \Delta\left(o_j \mid A_{l_2,,c_{l_1}^*+j-1}\right) \qquad \text{(submodularity)}$$

$$\le f\left(A_{l_1,c_{l_1}^*}\right) - f(G_{i-1}) + \sum_{j=1}^{m-c_{l_1}^*} \Delta\left(a_{l_2,c_{l_1}^*+j} \mid A_{l_2,,c_{l_1}^*+j-1}\right) \qquad \text{(Inequality (9))}$$

$$\le f\left(A_{l_1,c_{l_1}^*}\right) - f(G_{i-1}) + f(A_{l_2}) - f\left(A_{l_2,c_{l_1}^*}\right)$$

Similarly, we bound $\Delta(O_{l_2} \mid A_{l_1})$ below. Order the elements in $O_{l_2} \setminus A_{l_2,c_{l_1}^*}$ as $\{o_1, o_2, \ldots\}$ such that $o_j \notin A_{l_2,c_{l_1}^*+j}$. (See the gray block with a dotted edge in the bottom left corner of Fig. 4 for $O_{l_2}$.) Since each $o_j$ is either added to $A_{l_2}$ or not in

any solution set, it remains in the candidate pool when $a_{l_1,c^*_{l_1}+j}$ is considered to be added to $A_{l_2}$. Therefore, it holds that

$$\Delta\left(a_{l_1,c^*_{l_1}+j} \mid A_{l_1,c^*_{l_1}+j-1}\right) \geq \Delta\left(o_j \mid A_{l_1,c^*_{l_1}+j-1}\right), \forall 1 \leq j \leq m - c^*_{l_1}. \tag{10}$$

Then,

$$\Delta\left(O_{l_2} \mid A_{l_1}\right) \leq \Delta\left(A_{l_2,c^*_{l_1}} \mid A_{l_1}\right) + \sum_{o_j \in O_{l_2} \setminus A_{l_2,c^*_{l_1}}} \Delta\left(o_j \mid A_{l_1}\right) \qquad \text{(Proposition B.1)}$$

$$\leq \Delta\left(A_{l_2,c^*_{l_1}} \mid G_{i-1}\right) + \sum_{o_j \in O_{l_2} \setminus A_{l_2,c^*_{l_1}}} \Delta\left(o_j \mid A_{l_1,,c^*_{l_1}+j-1}\right) \qquad \text{(submodularity)}$$

$$\leq f\left(A_{l_2,c^*_{l_1}}\right) - f\left(G_{i-1}\right) + \sum_{j=1}^{m-c^*_{l_1}} \Delta\left(a_{l_1,c^*_{l_1}+j} \mid A_{l_1,,c^*_{l_1}+j-1}\right) \qquad \text{(Inequality (9))}$$

$$\leq f\left(A_{l_1,c^*_{l_1}}\right) - f\left(G_{i-1}\right) + f\left(A_{l_1}\right) - f\left(A_{l_1,c^*_{l_1}}\right)$$

Thus, the lemma holds in this case.

**Case 2:** $c^*_{l_1} > c^*_{l_2}$; **right half part in Fig. 4.** First, we bound $\Delta\left(O_{l_1} \mid A_{l_2}\right)$. Order the elements in $O_{l_1} \setminus A_{l_1,c^*_{l_2}+1}$ as $\{o_1, o_2, \ldots\}$ such that $o_j \notin A_{l_1,c^*_{l_2}+j}$. (Refer to the gray block with a dotted edge in the top right corner of Fig. 4 for $O_{l_1}$.) Since each $o_j$ is either in $A_{l_1}$ or not in any solution set, it remains in the candidate pool when $a_{l_2,c^*_{l_2}+j}$ is considered to be added to $A_{l_2}$. Therefore, it holds that

$$\Delta\left(a_{l_2,c^*_{l_2}+j} \mid A_{l_2,c^*_{l_2}+j-1}\right) \geq \Delta\left(o_j \mid A_{l_2,c^*_{l_2}+j-1}\right), \forall 1 \leq j \leq m - c^*_{l_2} - 1. \tag{11}$$

Then,

$$\Delta\left(O_{l_1} \mid A_{l_2}\right) \leq \Delta\left(A_{l_1,c^*_{l_2}+1} \mid A_{l_2}\right) + \sum_{o_j \in O_{l_1} \setminus A_{l_1,c^*_{l_2}+1}} \Delta\left(o_j \mid A_{l_2}\right) \qquad \text{(Proposition B.1)}$$

$$\leq \Delta\left(A_{l_1,c^*_{l_2}+1} \mid G_{i-1}\right) + \sum_{o_j \in O_{l_1} \setminus A_{l_1,c^*_{l_2}+1}} \Delta\left(o_j \mid A_{l_2,,c^*_{l_2}+j-1}\right) \qquad \text{(submodularity)}$$

$$\leq f\left(A_{l_1,c^*_{l_2}+1}\right) - f\left(G_{i-1}\right) + \sum_{j=1}^{m-c^*_{l_2}-1} \Delta\left(a_{l_2,c^*_{l_2}+j} \mid A_{l_2,,c^*_{l_2}+j-1}\right) \qquad \text{(Inequality (11))}$$

$$\leq f\left(A_{l_1,c^*_{l_2}+1}\right) - f\left(G_{i-1}\right) + f\left(A_{l_2}\right) - f\left(A_{l_2,c^*_{l_2}}\right)$$

Similarly, we bound $\Delta\left(O_{l_2} \mid A_{l_1}\right)$ below. Since $c^*_{l_2} < c^*_{l_2}$, we know that the $(c^*_{l_2} + 1)$-th element in $A_{l_2} \setminus G_{i-1}$ is not in $O$, which implies that $|O_{l_2}| \leq m$. So, we can order the elements in $O_{l_2} \setminus A_{l_2,c^*_{l_2}}$ as $\{o_1, o_2, \ldots\}$ such that $o_j \notin A_{l_2,c^*_{l_2}+j}$ for each $1 \leq j \leq m - c^*_{l_2}$. (See the gray block with a dotted edge in the bottom right corner of Fig. 4 for $O_{l_2}$.)

When $1 \leq j < m - c^*_{l_2}$, since each $o_j$ is either in $A_{l_2}$ or not in any solution set, it remains in the candidate pool when $a_{l_1,c^*_{l_2}+i+1}$ is considered to be added to $A_{l_1}$. Therefore, it holds that

$$\Delta\left(a_{l_1,c^*_{l_2}+j+1} \mid A_{l_1,c^*_{l_2}+j}\right) \geq \Delta\left(o_j \mid A_{l_1,c^*_{l_2}+j}\right), \forall 1 \leq j < m - c^*_{l_2}. \tag{12}$$

As for the last element $o_{m-c^*_{l_2}}$ in $O_{l_2} \setminus A_{l_2,c^*_{l_2}}$, we know that $o_{m-c^*_{l_2}}$ is not added to any solution set. So,

$$\Delta\left(o_{m-c^*_{l_2}} \mid A_{l_1}\right) \leq \frac{1}{m}\sum_{j=1}^{m} \Delta\left(a_{l_1,j} \mid A_{l_1,j-1}\right) = \frac{1}{m}\Delta\left(A_{l_1} \mid G_{i-1}\right) \tag{13}$$

Then,

$$\Delta\left(O_{l_2} \mid A_{l_1}\right) \le \Delta\left(A_{l_2,c_{l_2}^*} \mid A_{l_1}\right) + \sum_{o_j \in O_{l_2} \backslash A_{l_2,c_{l_2}^*}} \Delta\left(o_j \mid A_{l_1}\right) \qquad \text{(Proposition B.1)}$$

$$\le \Delta\left(A_{l_2,c_{l_2}^*} \mid G_{i-1}\right) + \sum_{o_j \in O_{l_2} \backslash A_{l_2,c_{l_2}^*}} \Delta\left(o_j \mid A_{l_1,,c_{l_2}^*+j}\right) \qquad \text{(submodularity)}$$

$$\le f\left(A_{l_2,c_{l_2}^*}\right) - f\left(G_{i-1}\right) + \sum_{j=1}^{m-c_{l_2}^*-1} \Delta\left(a_{l_1,c_{l_2}^*+j+1} \mid A_{l_1,,c_{l_2}^*+j}\right) + \frac{1}{m}\Delta\left(A_{l_1} \mid G_{i-1}\right) \quad \text{(Inequality (11))}$$

$$\le f\left(A_{l_1,c_{l_2}^*}\right) - f\left(G_{i-1}\right) + f\left(A_{l_1}\right) - f\left(A_{l_1,c_{l_2}^*+1}\right) + \frac{1}{m}\Delta\left(A_{l_1} \mid G_{i-1}\right) \qquad (|O_{l_2}| \le m)$$

Thus, the lemma holds in this case. □

**Lemma E.1.** For any iteration $i$ of the outer for loop in Alg. 2, it holds that

$$\mathbb{E}\left[f\left(G_i\right) - f\left(G_{i-1}\right)\right] \ge \frac{1}{\ell+1}\left(1 - \frac{1}{m+1}\right)$$

$$\cdot \left(\left(1 - \frac{1}{\ell}\right)\mathbb{E}\left[f\left(O \cup G_{i-1}\right)\right] - \mathbb{E}\left[f\left(G_{i-1}\right)\right]\right)$$

*Proof of Lemma E.1.* Fix on $G_{i-1}$ for an iteration $i$ of the outer for loop in Alg. 2. Let $A_l$ be the set after for loop in Lines 3-6 ends (with $m$ iterations). Then,

$$\sum_{l \in [\ell]} \Delta\left(O \mid A_l\right) \le \sum_{l \in [\ell]} \Delta\left(O_l \mid A_l\right) + \sum_{1 \le l_1 < l_2 \le \ell} \left(\Delta\left(O_{l_1} \mid A_{l_2}\right) + \Delta\left(O_{l_2} \mid A_{l_1}\right)\right) \qquad \text{(Inequality 2)}$$

$$\le \sum_{l \in [\ell]} \Delta\left(A_l \mid G_{i-1}\right) + \sum_{1 \le l_1 < l_2 \le \ell} \left(1 + \frac{1}{m}\right)\left(\Delta\left(A_{l_1} \mid G_{i-1}\right) + \Delta\left(A_{l_2} \mid G_{i-1}\right)\right) \qquad \text{(Lemma 3.2)}$$

$$\le \ell\left(1 + \frac{1}{m}\right)\sum_{l \in [\ell]} \Delta\left(A_l \mid G_{i-1}\right)$$

$$\Rightarrow (\ell+1)\left(1 + \frac{1}{m}\right)\sum_{l \in [\ell]} \Delta\left(A_l \mid G_{i-1}\right) \ge \sum_{l \in [\ell]} f\left(O \cup A_l\right) - \ell f\left(G_{i-1}\right)$$

$$\ge (\ell-1) f\left(O \cup G_{i-1}\right) - \ell f\left(G_{i-1}\right), \qquad (14)$$

where the last inequality follows from Proposition B.1. Then, it holds that

$$\mathbb{E}\left[f\left(G_i\right) - f\left(G_{i-1}\right) \mid G_{i-1}\right] = \frac{1}{\ell}\sum_{l \in [\ell]} \Delta\left(A_l \mid G_{i-1}\right)$$

$$\ge \frac{1}{\ell+1} \cdot \frac{m}{m+1} \cdot \left(\left(1 - \frac{1}{\ell}\right) f\left(O \cup G_{i-1}\right) - f\left(G_{i-1}\right)\right) \qquad \text{(Inequality (14))}$$

By unfixing $G_{i-1}$, the lemma holds. □

**Lemma E.2.** For any iteration $i$ of the outer for loop in Alg. 2, it holds that

$$\mathbb{E}\left[f\left(O \cup G_i\right)\right] \ge \left(1 - \frac{1}{\ell}\right)\mathbb{E}\left[f\left(O \cup G_{i-1}\right)\right].$$

*Proof of Lemma E.2.* Fix on $G_{i-1}$ at the beginning of this iteration. Since $\{A_l \setminus G_{i-1}\}_{l \in [\ell]}$ are pairwise disjoint sets at the end of this iteration, by Proposition B.1, it holds that

$$\mathbb{E}\left[f\left(O \cup G_i\right) \mid G_{i-1}\right] = \frac{1}{\ell} \sum_{l \in [\ell]} f\left(O \cup A_l\right) \geq \left(1 - \frac{1}{\ell}\right) f\left(O \cup G_{i-1}\right).$$

Then, by unfixing $G_{i-1}$, the lemma holds. □

### E.2. Proof of Theorem 3.1

**Theorem 3.1.** *With input instance $(f, k, \ell, \varepsilon)$ such that $\ell = \mathcal{O}\left(\varepsilon^{-1}\right) \geq \frac{2}{e\varepsilon}$ and $k \geq \frac{2(e\ell-2)}{e\varepsilon - \frac{2}{\ell}}$, Alg. 2 returns a set $G$ with $\mathcal{O}\left(kn/\varepsilon\right)$ queries such that $\mathbb{E}\left[f\left(G\right)\right] \geq (1/e - \varepsilon) f\left(O\right)$.*

*Proof.* By Lemma E.1 and E.2, the recurrence of $\mathbb{E}\left[f\left(G_i\right)\right]$ can be expressed as follows,

$$\mathbb{E}\left[f\left(G_i\right)\right] \geq \left(1 - \frac{1}{\ell+1}\left(1 - \frac{1}{m+1}\right)\right) \mathbb{E}\left[f\left(G_{i-1}\right)\right] + \frac{1}{\ell+1}\left(1 - \frac{1}{m+1}\right)\left(1 - \frac{1}{\ell}\right)^i f\left(O\right)$$

$$\geq \left(1 - \frac{1}{\ell}\right) \mathbb{E}\left[f\left(G_{i-1}\right)\right] + \frac{1}{\ell+1}\left(1 - \frac{1}{m+1}\right)\left(1 - \frac{1}{\ell}\right)^i f\left(O\right).$$

By solving the above recurrence,

$$\mathbb{E}\left[f\left(G_\ell\right)\right] \geq \frac{\ell}{\ell+1}\left(1 - \frac{1}{m+1}\right)\left(1 - \frac{1}{\ell}\right)^\ell f\left(O\right)$$

$$\geq \frac{\ell-1}{\ell+1}\left(1 - \frac{1}{m+1}\right) e^{-1} f\left(O\right) \qquad \text{(Lemma A.1)}$$

$$\geq \left(1 - \frac{2}{\ell}\right)\left(1 - \frac{\ell}{k}\right) e^{-1} f\left(O\right) \qquad (m = \lfloor \frac{k}{\ell} \rfloor)$$

$$\geq \frac{1}{1 - \frac{\ell}{k}}\left(1 - \frac{2}{\ell} - \frac{2\ell}{k} + \frac{4}{k}\right) e^{-1} f\left(O\right)$$

$$\geq \frac{1}{1 - \frac{\ell}{k}}\left(e^{-1} - \varepsilon\right) f\left(O\right). \qquad (\ell \geq \frac{2}{e\varepsilon}, k \geq \frac{2(\ell-2)}{e\varepsilon - \frac{2}{\ell}})$$

If $k \bmod \ell = 0$, the approximation ratio holds immediately.

Otherwise, when $k \bmod \ell > 0$, the algorithm returns an approximation solution for a size constraint of $\ell \cdot \lfloor \frac{k}{\ell} \rfloor$. By Proposition B.3, it holds that

$$f\left(O'\right) \geq \ell \cdot \left\lfloor \frac{k}{\ell} \right\rfloor / k f\left(O\right) \geq \left(1 - \frac{\ell}{k}\right) f\left(O\right), O' = \underset{S \subseteq \mathcal{U}, |S| \leq \ell \cdot \lfloor \frac{k}{\ell} \rfloor}{\arg\max} f\left(S\right). \qquad (15)$$

In this case, the approximation ratio still holds. □

## F. Preliminary Warm-Up of Parallel Approaches: Nearly-Linear Time Algorithms

Kuhnle (2019) introduced a fast version of INTERLACEGREEDY, replacing the greedy procedure with a descending threshold greedy procedure (Badanidiyuru & Vondrák, 2014) to achieve a query complexity of $\mathcal{O}\left(n \log(k)\right)$. This same technique was subsequently employed in INTERPOLATEDGREEDY (Chen & Kuhnle, 2023). In this section, we present simplified versions of both algorithms, incorporating the blended marginal gain analysis strategy introduced in Section 3. These fast algorithms serve as building blocks for the parallel algorithms introduced in this work.

---

**Algorithm 9:** A nearly-linear time, $(1/4 - \varepsilon)$-approximation algorithm.

---

**Input:** evaluation oracle $f : 2^{\mathcal{U}} \to \mathbb{R}_{\geq 0}$, constraint $k$, error $\varepsilon$
**Initialize:** $A \leftarrow \emptyset,\ B \leftarrow \emptyset,\ M \leftarrow \max_{x \in \mathcal{U}} f(\{x\}),\ \tau_1 \leftarrow M,\ \tau_2 \leftarrow M$

1 **for** $i \leftarrow 1$ *to* $k$ **do**
2      **while** $\tau_1 \geq \frac{\varepsilon M}{k}$ *and* $|A| < k$ **do**
3          **if** $\exists a \in \mathcal{U} \setminus (A \cup B)$ *s.t.* $\Delta(a \mid A) \geq \tau_1$ **then**
4              $A \leftarrow A + a$
5              **break**
6          **else** $\tau_1 \leftarrow (1 - \varepsilon)\tau_1$
7      **while** $\tau_2 \geq \frac{\varepsilon M}{k}$ *and* $|B| < k$ **do**
8          **if** $\exists b \in \mathcal{U} \setminus (A \cup B)$ *s.t.* $\Delta(b \mid B) \geq \tau_2$ **then**
9              $B \leftarrow B + b$
10             **break**
11          **else** $\tau_2 \leftarrow (1 - \varepsilon)\tau_2$
12 **return** $S \leftarrow \arg\max\{f(A), f(B)\}$

---

### F.1. Simplified Fast INTERLACEGREEDY with $1/4 - \varepsilon$ Approximation Ratio (Alg. 9)

**Theorem F.1.** *With input instance* $(f, k, \varepsilon)$, *Alg. 9 returns a set* $S$ *with* $\mathcal{O}(n \log(k)/\varepsilon)$ *queries such that* $f(S) \geq \left(\frac{1}{4} - \varepsilon\right) f(O)$.

*Proof.* **Query Complexity.** Without loss of generality, we analyze the number queries related to set $A$. For each threshold value $\tau_1$, at most $n$ queries are made to the value oracle. Since $\tau_1$ is initialized with value $M$, decreases by a factor of $1 - \varepsilon$, and cannot exceed $\frac{\varepsilon M}{k}$, there are at most $\log_{1-\varepsilon}\left(\frac{\varepsilon}{k}\right) + 1$ possible values of $\tau_1$. Therefore, the total number of queries is bounded as follows,

$$\#\text{Queries} \leq 2 \cdot n \cdot \left(\log_{1-\varepsilon}\left(\frac{\varepsilon}{k}\right) + 1\right) \leq \mathcal{O}(n \log(k)/\varepsilon),$$

where the last inequality follows from the first inequality in Lemma A.1.

**Approximation Ratio.** Since $A$ and $B$ are disjoint, by submodularity and non-negativity,

$$f(O) \leq f(O \cup A) + f(O \cup B). \tag{16}$$

Let $a_i$ be the $i$-th element added to $A$, $A_i$ be the first $i$ elements added to $A$, and $\tau_1^{a_i}$ be the threshold value when $a_i$ is added to $A$. Similarly, define $b_i$, $B_i$, and $\tau_2^{b_i}$. Let $i^* = \max\{i \leq |A| : A_i \subseteq O\}$ and $j^* = \max\{i \leq |B| : B_i \subseteq O\}$. If either $i^* = k$ or $j^* = k$, then $f(S) = f(O)$. Next, we follow the analysis of Alg. 8 in Section D to analyze the approximation ratio of Alg. 9.

**Case 1:** $0 \leq i^* \leq j^* < k$**; Fig. 3(a).** First, we bound $f(O \cup A)$. Since $B_{i^*} \subseteq O$, by submodularity

$$f(O \cup A) - f(A) \leq \Delta(B_{i^*} \mid A) + \Delta(O \setminus B_{i^*} \mid A) \leq f(B_{i^*}) + \sum_{o \in O \setminus (A \cup B_{i^*})} \Delta(o \mid A). \tag{17}$$

Next, we bound $\Delta(o \mid A)$ for each $o \in O \setminus (A \cup B_{i^*})$.

Let $\tilde{O} = O \setminus (A \cup B_{i^*})$. Obviously, it holds that $|\tilde{O}| \leq k - i^*$. Then, order $\tilde{O}$ as $\{o_1, o_2, \dots\}$ such that $o_i \notin B_{i+i^*-1}$, for all $1 \leq i \leq |\tilde{O}|$. If $|A| < k$, the algorithm terminates with $\tau_1 < \frac{\varepsilon M}{k}$. Thus, it follows that

$$\Delta(o_i \mid A) < \frac{\varepsilon M}{k(1 - \varepsilon)}, \forall |A| - i^* < i \leq |\tilde{O}|. \tag{18}$$

Next, consider tuple $(o_i, a_{i+i^*}, A_{i+i^*-1})$, for any $1 \leq i \leq \min\{|\tilde{O}|, |A| - i^*\}$. Since $\tau_1^{a_{i+i^*}}$ is the threshold value when $a_{i+i^*}$ is added, it holds that

$$\Delta(a_{i+i^*} \mid A_{i+i^*-1}) \geq \tau_1^{a_{i+i^*}}, \forall 1 \leq i \leq |A| - i^*. \tag{19}$$

Then, we show that $\Delta\left(o_i \mid A_{i+i^*-1}\right) < \tau_1^{a_{i+i^*}}/(1-\varepsilon)$ always holds for any $1 \le i \le \min\{|\tilde{O}|, |A| - i^*\}$.

Since $M = \max_{x \in \mathcal{U}} f\left(\{x\}\right)$, if $\tau_1^{a_{i+i^*}} \ge M$, it always holds that $\Delta\left(o_i \mid A_{i+i^*-1}\right) < M/(1-\varepsilon) \le \tau_1^{a_{i+i^*}}/(1-\varepsilon)$. If $\tau_1^{a_{i+i^*}} < M$, since $o_i \notin B_{i+i^*-1}$, $o_i$ is not considered to be added to $A$ with threshold value $\tau_1^{a_{i+i^*}}/(1-\varepsilon)$. Then, by submodularity, $\Delta\left(o_i \mid A_{i+i^*-1}\right) < \tau_1^{a_{i+i^*}}/(1-\varepsilon)$. Therefore, by submodularity and Inequality (19), it holds that

$$\Delta\left(o_i \mid A\right) \le \Delta\left(o_i \mid A_{i+i^*-1}\right) < \Delta\left(a_{i+i^*} \mid A_{i+i^*-1}\right)/(1-\varepsilon), \forall 1 \le i \le \min\{|\tilde{O}|, |A| - i^*\}. \tag{20}$$

Then,

$$\begin{aligned}
f\left(O \cup A\right) - f\left(A\right) &\le f\left(B_{i^*}\right) + \sum_{o \in O \setminus (A \cup B_{i^*})} \Delta\left(o \mid A\right) \\
&\le f\left(B_{i^*}\right) + \sum_{i=1}^{\min\{|\tilde{O}|, |A|\} - i^*} \Delta\left(a_{i+i^*} \mid A_{i+i^*-1}\right)/(1-\varepsilon) + \frac{\varepsilon M}{1-\varepsilon} \\
&\le \frac{1}{1-\varepsilon}\left(f\left(B_{i^*}\right) + f\left(A\right) - f\left(A_{i^*}\right) + \varepsilon f\left(O\right)\right), 
\end{aligned} \tag{21}$$

where the first inequality follows from Inequality (17); the second inequality follows from Inequalities (18) and (20); and the last inequality follows from $M \le f\left(O\right)$.

Second, we bound $f\left(O \cup B\right)$. Since $A_{i^*} \subseteq O$, by submodularity

$$f\left(O \cup B\right) - f\left(B\right) \le \Delta\left(A_{i^*} \mid B\right) + \Delta\left(O \setminus A_{i^*} \mid B\right) \le f\left(A_{i^*}\right) + \sum_{o \in O \setminus (B \cup A_{i^*})} \Delta\left(o \mid B\right). \tag{22}$$

Next, we bound $\Delta\left(o \mid B\right)$ for each $o \in O \setminus (B \cup A_{i^*})$.

Let $\tilde{O} = O \setminus (B \cup A_{i^*})$. Obviously, it holds that $|\tilde{O}| \le k - i^*$. Then, since $a_{i^*+1} \notin O$, we can order $\tilde{O}$ as $\{o_1, o_2, \ldots\}$ such that $o_i \notin A_{i+i^*}$, for all $1 \le i \le |\tilde{O}|$. If $|B| < k$, the algorithm terminates with $\tau_2 < \frac{\varepsilon M}{k}$. Thus, it follows that

$$\Delta\left(o_i \mid B\right) < \frac{\varepsilon M}{k(1-\varepsilon)}, \forall |B| - i^* < i \le |\tilde{O}|. \tag{23}$$

Next, consider tuple $(o_i, b_{i+i^*}, B_{i+i^*-1})$, for any $1 \le i \le \min\{|\tilde{O}|, |B| - i^*\}$. Since $\tau_2^{b_{i+i^*}}$ is the threshold value when $b_{i+i^*}$ is added, it holds that

$$\Delta\left(b_{i+i^*} \mid B_{i+i^*-1}\right) \ge \tau_2^{b_{i+i^*}}, \forall 1 \le i \le |B| - i^*. \tag{24}$$

Then, we show that $\Delta\left(o_i \mid B_{i+i^*-1}\right) < \tau_2^{b_{i+i^*}}/(1-\varepsilon)$ always holds for any $1 \le i \le \min\{|\tilde{O}|, |B| - i^*\}$.

Since $M = \max_{x \in \mathcal{U}} f\left(\{x\}\right)$, if $\tau_2^{b_{i+i^*}} \ge M$, it always holds that $\Delta\left(o_i \mid B_{i+i^*-1}\right) < M/(1-\varepsilon) \le \tau_2^{b_{i+i^*}}/(1-\varepsilon)$. If $\tau_2^{b_{i+i^*}} < M$, since $o_i \notin A_{i+i^*}$, $o_i$ is not considered to be added to $B$ with threshold value $\tau_2^{b_{i+i^*}}/(1-\varepsilon)$. Then, by submodularity, $\Delta\left(o_i \mid B_{i+i^*-1}\right) < \tau_2^{b_{i+i^*}}/(1-\varepsilon)$. Therefore, by submodularity and Inequality (24), it holds that

$$\Delta\left(o_i \mid B\right) \le \Delta\left(o_i \mid B_{i+i^*-1}\right) < \Delta\left(b_{i+i^*} \mid B_{i+i^*-1}\right)/(1-\varepsilon), \forall 1 \le i \le \min\{|\tilde{O}|, |B| - i^*\}. \tag{25}$$

Then,

$$\begin{aligned}
f\left(O \cup B\right) - f\left(B\right) &\le f\left(A_{i^*}\right) + \sum_{o \in O \setminus (B \cup A_{i^*})} \Delta\left(o \mid B\right) \\
&\le f\left(A_{i^*}\right) + \sum_{i=1}^{\min\{|\tilde{O}|, |B|\} - i^*} \Delta\left(b_{i+i^*} \mid B_{i+i^*-1}\right)/(1-\varepsilon) + \frac{\varepsilon M}{1-\varepsilon} \\
&\le \frac{1}{1-\varepsilon}\left(f\left(A_{i^*}\right) + f\left(B\right) - f\left(B_{i^*}\right) + \varepsilon f\left(O\right)\right), 
\end{aligned} \tag{26}$$

where the first inequality follows from Inequality (22); the second inequality follows from Inequalities (23) and (25); and the last inequality follows from $M \leq f(O)$.

By Inequalities (16), (21) and (26), it holds that

$$f(O) \leq \frac{2 - \varepsilon}{1 - \varepsilon} \left(f(A) + f(B)\right) + \frac{2\varepsilon}{1 - \varepsilon} f(O)$$

$$\Rightarrow f(S) \geq \left(\frac{1}{4} - \frac{5}{2(4 - 2\varepsilon)}\varepsilon\right) f(O) \geq \left(\frac{1}{4} - \varepsilon\right) f(O) \qquad (\varepsilon < 1/2)$$

**Case 2:** $0 \leq j^* < i^* < k$; **Fig. 3(b).**

First, we bound $f(O \cup A)$. Since $B_{j^*} \subseteq O$, by submodularity

$$f(O \cup A) - f(A) \leq \Delta(B_{j^*} \mid A) + \Delta(O \setminus B_{j^*} \mid A) \leq f(B_{j^*}) + \sum_{o \in O \setminus \left(A \cup B_{j^*}\right)} \Delta(o \mid A). \qquad (27)$$

Next, we bound $\Delta(o \mid A)$ for each $o \in O \setminus (A \cup B_{j^*})$.

Let $\tilde{O} = O \setminus (A \cup B_{j^*})$. Since $i^* > j^* \geq 0$, it holds that $|\tilde{O}| \leq k - j^* - 1$. Since $b_{j^*+1} \notin O$, we can order $\tilde{O}$ as $\{o_1, o_2, \ldots\}$ such that $o_i \notin B_{i+j^*}$, for all $1 \leq i \leq |\tilde{O}|$. If $|A| < k$, the algorithm terminates with $\tau_1 < \frac{\varepsilon M}{k}$. Thus, it follows that

$$\Delta(o_i \mid A) < \frac{\varepsilon M}{k(1 - \varepsilon)}, \forall |A| - j^* - 1 < i \leq |\tilde{O}|. \qquad (28)$$

Next, consider tuple $(o_i, a_{i+j^*+1}, A_{i+j^*})$, for any $1 \leq i \leq \min\{|\tilde{O}|, |A| - j^* - 1\}$. Since $\tau_1^{a_{i+j^*+1}}$ is the threshold value when $a_{i+j^*+1}$ is added, it holds that

$$\Delta(a_{i+j^*+1} \mid A_{i+j^*}) \geq \tau_1^{a_{i+j^*+1}}, \forall 1 \leq i \leq |A| - j^* - 1. \qquad (29)$$

Then, we show that $\Delta(o_i \mid A_{i+j^*}) < \tau_1^{a_{i+j^*+1}}/(1 - \varepsilon)$ always holds for any $1 \leq i \leq \min\{|\tilde{O}|, |A| - j^* - 1\}$.

Since $M = \max_{x \in \mathcal{U}} f(\{x\})$, if $\tau_1^{a_{i+j^*+1}} \geq M$, it always holds that $\Delta(o_i \mid A_{i+j^*}) < M/(1 - \varepsilon) \leq \tau_1^{a_{i+j^*+1}}/(1 - \varepsilon)$. If $\tau_1^{a_{i+j^*+1}} < M$, since $o_i \notin B_{i+j^*}$, $o_i$ is not considered to be added to $A$ with threshold value $\tau_1^{a_{i+j^*+1}}/(1 - \varepsilon)$. Then, by submodularity, $\Delta(o_i \mid A_{i+j^*}) < \tau_1^{a_{i+j^*+1}}/(1 - \varepsilon)$. Therefore, by submodularity and Inequality (29), it holds that

$$\Delta(o_i \mid A) \leq \Delta(o_i \mid A_{i+j^*}) < \Delta(a_{i+j^*+1} \mid A_{i+j^*})/(1 - \varepsilon), \forall 1 \leq i \leq \min\{|\tilde{O}|, |A| - j^* - 1\}. \qquad (30)$$

Then,

$$f(O \cup A) - f(A) \leq f(B_{j^*}) + \sum_{o \in O \setminus \left(A \cup B_{j^*}\right)} \Delta(o \mid A)$$

$$\leq f(B_{j^*}) + \sum_{i=1}^{\min\{|\tilde{O}|, |A| - j^* - 1\}} \Delta(a_{i+j^*+1} \mid A_{i+j^*})/(1 - \varepsilon) + \frac{\varepsilon M}{1 - \varepsilon}$$

$$\leq \frac{1}{1 - \varepsilon}\left(f(B_{j^*}) + f(A) - f(A_{j^*+1}) + \varepsilon f(O)\right), \qquad (31)$$

where the first inequality follows from Inequality (27); the second inequality follows from Inequalities (28) and (30); and the last inequality follows from $M \leq f(O)$.

Second, we bound $f(O \cup B)$. Since $A_{j^*+1} \subseteq O$, by submodularity

$$f(O \cup B) - f(B) \leq \Delta(A_{j^*+1} \mid B) + \Delta(O \setminus A_{j^*+1} \mid B) \leq f(A_{j^*+1}) + \sum_{o \in O \setminus \left(B \cup A_{j^*+1}\right)} \Delta(o \mid B). \qquad (32)$$

Next, we bound $\Delta(o \mid B)$ for each $o \in O \setminus (B \cup A_{j^*+1})$.

Let $\tilde{O} = O \setminus (B \cup A_{j^*+1})$. Obviously, it holds that $|\tilde{O}| \leq k - j^* - 1$. Then, order $\tilde{O}$ as $\{o_1, o_2, \dots\}$ such that $o_i \notin A_{i+j^*}$, for all $1 \leq i \leq |\tilde{O}|$. If $|B| < k$, the algorithm terminates with $\tau_2 < \frac{\varepsilon M}{k}$. Thus, it follows that

$$\Delta\left(o_i \mid B\right) < \frac{\varepsilon M}{k(1 - \varepsilon)}, \forall |B| - j^* - 1 < i \leq |\tilde{O}|. \tag{33}$$

Next, consider tuple $(o_i, b_{i+j^*}, B_{i+j^*-1})$, for any $1 \leq i \leq \min\{|\tilde{O}|, |B| - j^* - 1\}$. Since $\tau_2^{b_{i+j^*}}$ is the threshold value when $b_{i+j^*}$ is added, it holds that

$$\Delta\left(b_{i+j^*} \mid B_{i+j^*-1}\right) \geq \tau_2^{b_{i+j^*}}, \forall 1 \leq i \leq |B| - j^* - 1. \tag{34}$$

Then, we show that $\Delta\left(o_i \mid B_{i+j^*-1}\right) < \tau_2^{b_{i+j^*}}/(1 - \varepsilon)$ always holds for any $1 \leq i \leq \min\{|\tilde{O}|, |B| - j^* - 1\}$.

Since $M = \max_{x \in \mathcal{U}} f(\{x\})$, if $\tau_2^{b_{i+j^*}} \geq M$, it always holds that $\Delta\left(o_i \mid B_{i+j^*-1}\right) < M/(1 - \varepsilon) \leq \tau_2^{b_{i+j^*}}/(1 - \varepsilon)$. If $\tau_2^{b_{i+j^*}} < M$, since $o_i \notin A_{i+j^*}$, $o_i$ is not considered to be added to $B$ with threshold value $\tau_2^{b_{i+j^*}}/(1 - \varepsilon)$. Then, by submodularity, $\Delta\left(o_i \mid B_{i+j^*-1}\right) < \tau_2^{b_{i+j^*}}/(1 - \varepsilon)$. Therefore, by submodularity and Inequality (34), it holds that

$$\Delta\left(o_i \mid B\right) \leq \Delta\left(o_i \mid B_{i+j^*-1}\right) < \Delta\left(b_{i+j^*} \mid B_{i+j^*-1}\right)/(1 - \varepsilon), \forall 1 \leq i \leq \min\{|\tilde{O}|, |B| - j^* - 1\}. \tag{35}$$

Then,

$$f\left(O \cup B\right) - f\left(B\right) \leq f\left(A_{j^*+1}\right) + \sum_{o \in O \setminus \left(B \cup A_{j^*+1}\right)} \Delta\left(o \mid B\right)$$

$$\leq f\left(A_{j^*+1}\right) + \sum_{i=1}^{\min\{|\tilde{O}|, |B| - j^* - 1\}} \Delta\left(b_{i+j^*} \mid B_{i+j^*-1}\right)/(1 - \varepsilon) + \frac{\varepsilon M}{1 - \varepsilon}$$

$$\leq \frac{1}{1 - \varepsilon}\left(f\left(A_{j^*+1}\right) + f\left(B\right) - f\left(B_{i^*}\right) + \varepsilon f\left(O\right)\right), \tag{36}$$

where the first inequality follows from Inequality (32); the second inequality follows from Inequalities (33) and (35); and the last inequality follows from $M \leq f(O)$.

By Inequalities (16), (31) and (36), it holds that

$$f\left(O\right) \leq \frac{2 - \varepsilon}{1 - \varepsilon}\left(f\left(A\right) + f\left(B\right)\right) + \frac{2\varepsilon}{1 - \varepsilon}f\left(O\right)$$

$$\Rightarrow f\left(S\right) \geq \left(\frac{1}{4} - \frac{5}{2(4 - 2\varepsilon)}\varepsilon\right)f\left(O\right) \geq \left(\frac{1}{4} - \varepsilon\right)f\left(O\right) \qquad (\varepsilon < 1/2)$$

Therefore, in both cases, it holds that

$$f\left(S\right) \geq \left(\frac{1}{4} - \varepsilon\right)f\left(O\right).$$

$\square$

## F.2. Simplified Fast INTERPOLATEDGREEDY with $1/e - \varepsilon$ Approximation Ratio (Alg. 10)

**Theorem F.2.** *With input instance $(f, k, \varepsilon)$ such that $\ell = \mathcal{O}\left(\varepsilon^{-1}\right) \geq \frac{4}{e\varepsilon}$ and $k \geq \frac{2(2-\varepsilon)\ell^2}{e\varepsilon\ell - 4}$, Alg. 2 (Alg. 10) returns a set $G_\ell$ with $\mathcal{O}\left(n\log(k)/\varepsilon^2\right)$ queries such that $f\left(G_\ell\right) \geq (1/e - \varepsilon)f\left(O\right)$.*

*Proof.* When $k \bmod \ell > 0$, the algorithm returns an approximation with a size constraint of $\ell \cdot \left\lfloor \frac{k}{\ell} \right\rfloor$, where by Proposition B.3,

$$f\left(O'\right) \geq \left(1 - \frac{\ell}{k}\right)f\left(O\right), O' = \underset{S \subseteq \mathcal{U}, |S| \leq \ell \cdot \left\lfloor \frac{k}{\ell} \right\rfloor}{\arg\max} f\left(S\right). \tag{37}$$

---

**Algorithm 10:** A nearly-linear time, $(1/e - \varepsilon)$-approximation algorithm.

---

**Input:** evaluation oracle $f : 2^{\mathcal{U}} \to \mathbb{R}_{\geq 0}$, constraint $k$, error $\varepsilon$

**Initialize:** $G_0 \leftarrow \emptyset$, $\varepsilon' \leftarrow \frac{\varepsilon}{2}$, $m \leftarrow \lfloor \frac{k}{\ell} \rfloor$, $\ell \leftarrow \lceil \frac{4}{e\varepsilon'} \rceil$, $M \leftarrow \max_{x \in \mathcal{U}} f(\{x\})$

1 **for** $i \leftarrow 1$ *to* $\ell$ **do**

2     $\tau_l \leftarrow M, \forall l \in [\ell]$

3     $A_l \leftarrow G_{i-1}, \forall l \in [\ell]$

4     **for** $j \leftarrow 1$ *to* $m$ **do**

5        **for** $l \leftarrow 1$ *to* $\ell$ **do**

6           **while** $\tau_l \geq \frac{\varepsilon' M}{k}$ *and* $|A_l \setminus G_{i-1}| < m$ **do**

7             **if** $\exists x \in \mathcal{U} \setminus \left( \bigcup_{r \in [\ell]} A_r \right)$ *s.t.* $\Delta(a \,|\, A_l) \geq \tau_l$ **then**

8                $A_l \leftarrow A_l + x$

9                **break**

10             **else** $\tau_l \leftarrow (1 - \varepsilon')\tau_l$

11     $G_i \leftarrow$ a random set in $\{A_l\}_{l \in [\ell]}$

12 **return** $G_\ell$

---

In the following, we only consider the case where $k \bmod \ell = 0$.

At every iteration of the outer for loop, $\ell$ solutions are constructed, with each solution being augmented by at most $k/\ell$ elements. To bound the marginal gain of the optimal set $O$ on each solution set $A_l$, we consider partitioning $O$ into $\ell$ subsets. We formalize this partition in the following claim, which yields a result analogous to Claim 3.1 presented in Section 3.3. Specifically, the claim states that the optimal set $O$ can be evenly divided into $\ell$ subsets, where each subset only overlaps with only one solution set.

**Claim F.1.** At an iteration $i$ of the outer for loop in Alg. 10, let $G_{i-1}$ be $G$ at the start of this iteration, and $A_l$ be the set at the end of this iteration, for each $l \in [\ell]$. The set $O \setminus G_{i-1}$ can then be split into $\ell$ pairwise disjoint sets $\{O_1, \ldots, O_\ell\}$ such that $|O_l| \leq \frac{k}{\ell}$ and $(O \setminus G_{i-1}) \cap (A_l \setminus G_{i-1}) \subseteq O_l$, for all $l \in [\ell]$.

Next, based on such partition, we introduce the following lemma, which provides a bound on the marginal gain of any subset $O_{l_1}$ with respect to any solution set $A_{l_2}$, where $1 \leq l_1, l_2 \leq \ell$.

**Lemma F.3.** Fix on $G_{i-1}$ for an iteration $i$ of the outer for loop in Alg. 10. Following the definition in Claim 3.1, it holds that

1) $\Delta(O_l \,|\, A_l) \leq \dfrac{\Delta(A_l \,|\, G_{j-1})}{1 - \varepsilon'} + \dfrac{\varepsilon' M}{(1 - \varepsilon')\ell}, \forall 1 \leq l \leq \ell,$

2) $\Delta(O_{l_2} \,|\, A_{l_1}) + \Delta(O_{l_1} \,|\, A_{l_2}) \leq \dfrac{1}{1 - \varepsilon'} \left( 1 + \dfrac{1}{m} \right) (\Delta(A_{l_1} \,|\, G_{i-1}) + \Delta(A_{l_2} \,|\, G_{i-1})) + \dfrac{2\varepsilon' M}{(1 - \varepsilon')\ell}, \forall 1 \leq l_1 < l_2 \leq \ell.$

Followed by the above lemma, we provide the recurrence of $\mathbb{E}[f(G_i)]$ and $\mathbb{E}[f(O \cup G_i)]$.

**Lemma F.4.** For any iteration $i$ of the outer for loop in Alg. 10, it holds that

1) $\mathbb{E}[f(O \cup G_i)] \geq \left( 1 - \dfrac{1}{\ell} \right) \mathbb{E}[f(O \cup G_{i-1})]$

2) $\mathbb{E}[f(G_i) - f(G_{i-1})] \geq \dfrac{1}{1 + \frac{\ell}{1 - \varepsilon'}} \left( 1 - \dfrac{1}{m+1} \right) \left( \left( 1 - \dfrac{1}{\ell} \right) \mathbb{E}[f(O \cup G_{i-1})] - \mathbb{E}[f(G_{i-1})] - \dfrac{\varepsilon'}{1 - \varepsilon'} f(O) \right).$

By solving the recurrence in Lemma F.4, we calculate the approximation ratio of the algorithm as follows,

$$\mathbb{E}[f(G_i)] \geq \left( 1 - \frac{1}{\ell} \right) \mathbb{E}[f(G_{i-1})] + \frac{1}{1 + \frac{\ell}{1-\varepsilon'}} \left( 1 - \frac{1}{m+1} \right) \left( \left( 1 - \frac{1}{\ell} \right)^i - \frac{\varepsilon'}{1 - \varepsilon'} \right) f(O)$$

$$\Rightarrow \mathbb{E}\left[f\left(G_\ell\right)\right] \geq \frac{\ell}{1+\frac{\ell}{1-\varepsilon'}}\left(1-\frac{1}{m+1}\right)\left(\left(1-\frac{1}{\ell}\right)^\ell - \frac{\varepsilon'}{1-\varepsilon'}\left(1-\left(1-\frac{1}{\ell}\right)^\ell\right)\right)f\left(O\right)$$

$$\geq \frac{\ell-1}{1+\frac{\ell}{1-\varepsilon'}}\left(1-\frac{1}{m+1}\right)\left(e^{-1}-\frac{\varepsilon'}{1-\varepsilon'}\left(1-e^{-1}\right)\right)f\left(O\right)$$

$$\geq \frac{1}{1-\frac{\ell}{k}}\left(1-\varepsilon'-\frac{2}{\ell}\right)\left(1-\frac{\ell}{k}\right)^2\left(e^{-1}-\frac{\varepsilon'}{1-\varepsilon'}\left(1-e^{-1}\right)\right)f\left(O\right)$$

$$\geq \frac{1}{1-\frac{\ell}{k}}\left(1-\varepsilon'-\frac{2}{\ell}-\frac{2(1-\varepsilon')\ell}{k}\right)\left(e^{-1}-\frac{\varepsilon'}{1-\varepsilon'}\left(1-e^{-1}\right)\right)f\left(O\right)$$

$$\geq \frac{1}{1-\frac{\ell}{k}}\left(1-(e+1)\varepsilon'\right)\left(e^{-1}-\frac{\varepsilon'}{1-\varepsilon'}\left(1-e^{-1}\right)\right)f\left(O\right) \qquad (\ell \geq \tfrac{2}{e\varepsilon'}, k \geq \tfrac{2(1-\varepsilon')\ell}{e\varepsilon'-\frac{2}{\ell}})$$

$$\geq \frac{1}{1-\frac{\ell}{k}}\left(e^{-1}-\varepsilon\right)f\left(O\right). \qquad (\varepsilon'=\tfrac{\varepsilon}{2})$$

By Inequality 37, the approximation ratio of Alg. 10 is $e^{-1}-\varepsilon$. $\qquad\square$

In the rest of this section, we provide the proofs for Lemma F.3 and F.4.

*Proof of Lemma F.3.* At iteration $i$ of the outer for loop, let $A_l$ be the set at the end of iteration $i$, $a_{l,j}$ be the $j$-th element added to $A_l$, $\tau_l^j$ be the threshold value of $\tau_l$ when $a_{l,j}$ is added to $A_l$, and $A_{l,j}$ be $A_l$ after $a_{l,j}$ is added to $A_l$. Let $c_l^* = \max\{c \in [m] : A_{l,c} \setminus G_{i-1} \subseteq O_l\}$.

First, we prove that the first inequality holds. For each $l \in [\ell]$, order the elements in $O_l$ as $\{o_1, o_2, \ldots\}$ such that $o_j \notin A_{l,j-1}$ for any $1 \leq j \leq |A_l \setminus G_{i-1}|$, and $o_j \notin A_l$ for any $|A_l \setminus G_{i-1}| < j \leq m$.

When $1 \leq j \leq |A_l \setminus G_{i-1}|$, by Claim 3.1, each $o_j$ is either added to $A_l$ or not in any solution set. Since $\tau_l$ is initialized with the maximum marginal gain $M$, $o_j$ is not considered to be added to $A_l$ with threshold value $\tau_l^j/(1-\varepsilon')$. Therefore, by submodularity it holds that

$$\Delta\left(o_j \mid A_{l,j-1}\right) < \tau_l^j/(1-\varepsilon') \leq \Delta\left(a_{l,j} \mid A_{l,j-1}\right)/(1-\varepsilon'), \forall 1 \leq j \leq |A_l \setminus G_{i-1}|. \qquad (38)$$

When $|A_l \setminus G_{i-1}| < m$, the minimum value of $\tau_l$ is less than $\frac{\varepsilon'M}{k}$. Then, for any $|A_l \setminus G_{i-1}| < j \leq m$, $o_j$ is not considered to be added to $A_l$ with threshold value less than $\frac{\varepsilon'M}{(1-\varepsilon')k}$. It follows that

$$\Delta\left(o_j \mid A_l\right) \leq \frac{\varepsilon'M}{(1-\varepsilon')k}, \forall |A_l \setminus G_{i-1}| < j \leq m. \qquad (39)$$

Then,

$$\Delta\left(O_l \mid A_l\right) \leq \sum_{o_j \in O_l} \Delta\left(o_j \mid A_l\right) \qquad \text{(Proposition B.1)}$$

$$\leq \sum_{j=1}^{|A_l \setminus G_{i-1}|} \Delta\left(o_j \mid A_{l,j-1}\right) + \sum_{j=|A_l \setminus G_{i-1}|+1}^{m} \Delta\left(o_j \mid A_l\right) \qquad \text{(Submodularity)}$$

$$\leq \sum_{j=1}^{|A_l \setminus G_{i-1}|} \frac{\Delta\left(a_{l,j} \mid A_{l,j-1}\right)}{1-\varepsilon'} + \frac{\varepsilon'M}{(1-\varepsilon')\ell} \qquad \text{(Inequalities (38) and (39))}$$

$$= \frac{\Delta\left(A_l \mid G_{j-1}\right)}{1-\varepsilon'} + \frac{\varepsilon'M}{(1-\varepsilon')\ell}.$$

The first inequality holds.

In the following, we prove that the second inequality holds. For any $1 \leq l_1 \leq l_2 \leq \ell$, we analyze two cases of the relationship between $c_{l_1}^*$ and $c_{l_2}^*$ in the following.

**Case 1: $c_{l_1}^* \leq c_{l_2}^*$; left half part in Fig. 4.**

First, we bound $\Delta\left(O_{l_1} \mid A_{l_2}\right)$. Order the elements in $O_{l_1} \setminus A_{l_1,c_{l_1}^*}$ as $\{o_1, o_2, \ldots\}$ such that $o_j \notin A_{l_1,c_{l_1}^*+j}$. (Refer to the gray block with a dotted edge in the top left corner of Fig. 4 for $O_{l_1}$. If $c_{l_1}^* + j$ is greater than the number of elements added to $A_{l_1}$, $A_{l_1,c_{l_1}^*+j}$ refers to $A_{l_1}$.) Note that, since $A_{l_1,c_{l_1}^*} \subseteq O_{l_1}$, it follows that $|O_{l_1} \setminus A_{l_1,c_{l_1}^*}| \leq m - c_{l_1}^*$.

When $1 \leq j \leq |A_{l_2} \setminus G_{i-1}| - c_{l_1}^*$, since each $o_j$ is either added to $A_{l_1}$ or not in any solution set by Claim 3.1 and $\tau_{l_2}$ is initialized with the maximum marginal gain $M$, $o_j$ is not considered to be added to $A_{l_2}$ with threshold value $\tau_{l_2}^{c_{l_1}^*+j}/(1-\varepsilon')$. Therefore, it holds that

$$\Delta\left(o_j \,\Big|\, A_{l_2,c_{l_1}^*+j-1}\right) < \frac{\tau_{l_2}^{c_{l_1}^*+j}}{1-\varepsilon'} \leq \frac{\Delta\left(a_{l_2,c_{l_1}^*+j} \,\Big|\, A_{l_2,c_{l_1}^*+j-1}\right)}{1-\varepsilon'}, \forall 1 \leq j \leq |A_{l_2} \setminus G_{i-1}| - c_{l_1}^*. \tag{40}$$

When $|A_{l_2} \setminus G_{i-1}| < m$ and $|A_{l_2} \setminus G_{i-1}| - c_{l_1}^* < j \leq m - c_{l_1}^*$, this iteration ends with $\tau_{l_2} < \frac{\varepsilon' M}{k}$ and $o_j$ is never considered to be added to $A_{l_2}$. Thus, it holds that

$$\Delta\left(o_j \mid A_{l_2}\right) < \frac{\varepsilon' M}{(1-\varepsilon')k}, \forall |A_{l_2} \setminus G_{i-1}| - c_{l_1}^* < j \leq m - c_{l_1}^*. \tag{41}$$

Then,

$$\Delta\left(O_{l_1} \mid A_{l_2}\right) \leq \Delta\left(A_{l_1,c_{l_1}^*} \,\Big|\, A_{l_2}\right) + \sum_{o_j \in O_{l_1} \setminus A_{l_1,c_{l_1}^*}} \Delta\left(o_j \mid A_{l_2}\right) \qquad \text{(Proposition B.1)}$$

$$\leq \Delta\left(A_{l_1,c_{l_1}^*} \,\Big|\, G_{i-1}\right) + \sum_{j=1}^{|A_{l_2} \setminus G_{i-1}|-c_{l_1}^*} \Delta\left(o_j \,\Big|\, A_{l_2,,c_{l_1}^*+j-1}\right) + \sum_{j=|A_{l_2} \setminus G_{i-1}|-c_{l_1}^*+1}^{m-c_{l_1}^*} \Delta\left(o_j \mid A_{l_2}\right)$$

$$\text{(submodularity)}$$

$$\leq f\left(A_{l_1,c_{l_1}^*}\right) - f\left(G_{i-1}\right) + \sum_{j=1}^{|A_{l_2} \setminus G_{i-1}|-c_{l_1}^*} \frac{\Delta\left(a_{l_2,c_{l_1}^*+j} \,\Big|\, A_{l_2,,c_{l_1}^*+j-1}\right)}{1-\varepsilon'} + \frac{\varepsilon' M}{(1-\varepsilon')\ell}$$

$$\text{(Inequality (40) and (41))}$$

$$\leq f\left(A_{l_1,c_{l_1}^*}\right) - f\left(G_{i-1}\right) + \frac{f\left(A_{l_2}\right) - f\left(A_{l_2,c_{l_1}^*}\right)}{1-\varepsilon'} + \frac{\varepsilon' M}{(1-\varepsilon')\ell} \tag{42}$$

Similarly, we bound $\Delta\left(O_{l_2} \mid A_{l_1}\right)$ below. Order the elements in $O_{l_2} \setminus A_{l_2,c_{l_1}^*}$ as $\{o_1, o_2, \ldots\}$ such that $o_j \notin A_{l_2,c_{l_1}^*+j-1}$. (See the gray block with a dotted edge in the bottom left corner of Fig. 4 for $O_{l_2}$. If $c_{l_1}^* + j - 1$ is greater than the number of elements added to $A_{l_2}$, $A_{l_2,c_{l_1}^*+j-1}$ refers to $A_{l_2}$.) Note that, since $A_{l_2,c_{l_1}^*} \subseteq O_{l_2}$, it follows that $|O_{l_2} \setminus A_{l_2,c_{l_1}^*}| \leq m - c_{l_1}^*$.

When $1 \leq j \leq |A_{l_1} \setminus G_{i-1}| - c_{l_1}^*$, since each $o_j$ is either added to $A_{l_2}$ or not in any solution set by Claim 3.1 and $\tau_{l_1}$ is initialized with the maximum marginal gain $M$, $o_j$ is not considered to be added to $A_{l_1}$ with threshold value $\tau_{l_1}^{c_{l_1}^*+j}/(1-\varepsilon')$. Therefore, it holds that

$$\Delta\left(o_j \,\Big|\, A_{l_1,c_{l_1}^*+j-1}\right) < \frac{\tau_{l_1}^{c_{l_1}^*+j}}{1-\varepsilon'} \leq \frac{\Delta\left(a_{l_1,c_{l_1}^*+j} \,\Big|\, A_{l_1,c_{l_1}^*+j-1}\right)}{1-\varepsilon'}, \forall 1 \leq j \leq |A_{l_2} \setminus G_{i-1}| - c_{l_1}^*. \tag{43}$$

When $|A_{l_1} \setminus G_{i-1}| < m$ and $|A_{l_1} \setminus G_{i-1}| - c_{l_1}^* < j \leq m - c_{l_1}^*$, this iteration ends with $\tau_{l_1} < \frac{\varepsilon' M}{k}$ and $o_j$ is never considered to be added to $A_{l_1}$. Thus, it holds that

$$\Delta\left(o_j \mid A_{l_1}\right) < \frac{\varepsilon' M}{(1-\varepsilon')k}, \forall |A_{l_1} \setminus G_{i-1}| - c_{l_1}^* < j \leq m - c_{l_1}^*. \tag{44}$$

Then,

$$\Delta\left(O_{l_2} \mid A_{l_1}\right) \le \Delta\left(A_{l_2,c_{l_1}^*} \mid A_{l_1}\right) + \sum_{o_j \in O_{l_2} \setminus A_{l_2,c_{l_1}^*}} \Delta\left(o_j \mid A_{l_1}\right) \qquad \text{(Proposition B.1)}$$

$$\le \Delta\left(A_{l_2,c_{l_1}^*} \mid G_{i-1}\right) + \sum_{j=1}^{|A_{l_1} \setminus G_{i-1}| - c_{l_1}^*} \Delta\left(o_j \mid A_{l_1,c_{l_1}^*+j-1}\right) + \sum_{j=|A_{l_1} \setminus G_{i-1}| - c_{l_1}^*+1}^{m-c_{l_1}^*} \Delta\left(o_j \mid A_{l_1}\right)$$

$$\text{(submodularity)}$$

$$\le f\left(A_{l_2,c_{l_1}^*}\right) - f\left(G_{i-1}\right) + \sum_{j=1}^{|A_{l_1} \setminus G_{i-1}| - c_{l_1}^*} \frac{\Delta\left(a_{l_1,c_{l_1}^*+j} \mid A_{l_1,,c_{l_1}^*+j-1}\right)}{1-\varepsilon'} + \frac{\varepsilon' M}{(1-\varepsilon')\ell}$$

$$\text{(Inequality (43) and (44))}$$

$$\le f\left(A_{l_2,c_{l_1}^*}\right) - f\left(G_{i-1}\right) + \frac{f\left(A_{l_1}\right) - f\left(A_{l_1,c_{l_1}^*}\right)}{1-\varepsilon'} + \frac{\varepsilon' M}{(1-\varepsilon')\ell} \qquad (45)$$

By Inequalities (42) and (45),

$$\Delta\left(O_{l_1} \mid A_{l_2}\right) + \Delta\left(O_{l_2} \mid A_{l_1}\right) \le \frac{1}{1-\varepsilon'}\left(\Delta\left(A_{l_1} \mid G_{i-1}\right) + \Delta\left(A_{l_2} \mid G_{i-1}\right)\right) + \frac{2\varepsilon' M}{(1-\varepsilon')\ell}$$

Thus, the lemma holds in this case.

**Case 2: $c_{l_1}^* > c_{l_2}^*$; right half part in Fig. 4.**

First, we bound $\Delta\left(O_{l_1} \mid A_{l_2}\right)$. Order the elements in $O_{l_1} \setminus A_{l_1,c_{l_2}^*+1}$ as $\{o_1, o_2, \ldots\}$ such that $o_j \notin A_{l_1,c_{l_1}^*+j}$. (Refer to the gray block with a dotted edge in the top right corner of Fig. 4 for $O_{l_1}$. If $c_{l_1}^* + j$ is greater than the number of elements added to $A_{l_1}$, $A_{l_1,c_{l_1}^*+j}$ refers to $A_{l_1}$.) Note that, since $A_{l_1,c_{l_2}^*+1} \subseteq O_{l_1}$, it follows that $|O_{l_1} \setminus A_{l_1,c_{l_2}^*+1}| \le m - c_{l_2}^* - 1$.

When $1 \le j \le |A_{l_2} \setminus G_{i-1}| - c_{l_2}^* - 1$, since each $o_j$ is either added to $A_{l_1}$ or not in any solution set by Claim 3.1 and $\tau_{l_2}$ is initialized with the maximum marginal gain $M$, $o_j$ is not considered to be added to $A_{l_2}$ with threshold value $\tau_{l_2}^{c_{l_2}^*+j}/(1-\varepsilon')$. Therefore, it holds that

$$\Delta\left(o_j \mid A_{l_2,c_{l_2}^*+j-1}\right) < \frac{\tau_{l_2}^{c_{l_2}^*+j}}{1-\varepsilon'} \le \frac{\Delta\left(a_{l_2,c_{l_2}^*+j} \mid A_{l_2,c_{l_2}^*+j-1}\right)}{1-\varepsilon'}, \forall 1 \le j \le |A_{l_2} \setminus G_{i-1}| - c_{l_2}^* - 1. \qquad (46)$$

When $|A_{l_2} \setminus G_{i-1}| < m$ and $|A_{l_2} \setminus G_{i-1}| - c_{l_2}^* - 1 < j \le m - c_{l_2}^* - 1$, this iteration ends with $\tau_{l_2} < \frac{\varepsilon' M}{k}$ and $o_j$ is never considered to be added to $A_{l_2}$. Thus, it holds that

$$\Delta\left(o_j \mid A_{l_2}\right) < \frac{\varepsilon' M}{(1-\varepsilon')k}, \forall |A_{l_2} \setminus G_{i-1}| - c_{l_2}^* - 1 < j \le m - c_{l_2}^* - 1. \qquad (47)$$

Then,

$$\Delta\left(O_{l_1} \mid A_{l_2}\right) \le \Delta\left(A_{l_1,c_{l_2}^*} \mid A_{l_2}\right) + \sum_{o_j \in O_{l_1} \setminus A_{l_1,c_{l_2}^*+1}} \Delta\left(o_j \mid A_{l_2}\right) \qquad \text{(Proposition B.1)}$$

$$\le \Delta\left(A_{l_1,c_{l_2}^*} \mid G_{i-1}\right) + \sum_{j=1}^{|A_{l_2} \setminus G_{i-1}| - c_{l_2}^*-1} \Delta\left(o_j \mid A_{l_2,,c_{l_2}^*+j-1}\right) + \sum_{j=|A_{l_2} \setminus G_{i-1}| - c_{l_2}^*}^{m-c_{l_2}^*-1} \Delta\left(o_j \mid A_{l_2}\right)$$

$$\text{(submodularity)}$$

$$\le f\left(A_{l_1,c_{l_2}^*}\right) - f\left(G_{i-1}\right) + \sum_{j=1}^{|A_{l_2} \setminus G_{i-1}| - c_{l_2}^*-1} \frac{\Delta\left(a_{l_2,c_{l_2}^*+j} \mid A_{l_2,,c_{l_2}^*+j-1}\right)}{1-\varepsilon'} + \frac{\varepsilon' M}{(1-\varepsilon')\ell}$$

$$\text{(Inequality (46) and (47))}$$

$$\leq f\left(A_{l_1,c_{l_2}^*}\right) - f\left(G_{i-1}\right) + \frac{f\left(A_{l_2}\right) - f\left(A_{l_2,c_{l_2}^*}\right)}{1-\varepsilon'} + \frac{\varepsilon' M}{(1-\varepsilon')\ell} \tag{48}$$

Similarly, we bound $\Delta\left(O_{l_2} \mid A_{l_1}\right)$ below. Order the elements in $O_{l_2} \setminus A_{l_2,c_{l_2}^*}$ as $\{o_1, o_2, \ldots\}$ such that $o_j \notin A_{l_2,c_{l_2}^*+j}$. (See the gray block with a dotted edge in the bottom right corner of Fig. 4 for $O_{l_2}$. If $c_{l_2}^* + j$ is greater than the number of elements added to $A_{l_2}$, $A_{l_2,c_{l_2}^*+j}$ refers to $A_{l_2}$.) Note that, since $A_{l_2,c_{l_2}^*} \subseteq O_{l_2}$, it follows that $|O_{l_2} \setminus A_{l_2,c_{l_2}^*}| \leq m - c_{l_2}^*$.

When $1 \leq j \leq |A_{l_1} \setminus G_{i-1}| - c_{l_2}^* - 1$, since each $o_j$ is either added to $A_{l_2}$ or not in any solution set by Claim 3.1 and $\tau_{l_1}$ is initialized with the maximum marginal gain $M$, $o_j$ is not considered to be added to $A_{l_1}$ with threshold value $\tau_{l_1}^{c_{l_2}^*+j+1}/(1-\varepsilon')$. Therefore, it holds that

$$\Delta\left(o_j \,\Big|\, A_{l_1,c_{l_2}^*+j}\right) < \frac{\tau_{l_1}^{c_{l_2}^*+j+1}}{1-\varepsilon'} \leq \frac{\Delta\left(a_{l_1,c_{l_2}^*+j+1} \,\Big|\, A_{l_1,c_{l_2}^*+j}\right)}{1-\varepsilon'}, \forall 1 \leq j \leq |A_{l_2} \setminus G_{i-1}| - c_{l_2}^* - 1. \tag{49}$$

If $|A_{l_1} \setminus G_{i-1}| = m$, consider the last element $o_{m-c_{l_2}^*}$ in $O_{l_2} \setminus A_{l_2,c_{l_2}^*}$. Since $o_{m-c_{l_2}^*} \notin A_{l_2}$ and $o_{m-c_{l_2}^*} \notin A_{l_1}$, $o_{m-c_{l_2}^*}$ is not considered to be added to $A_{l_1}$ with threshold value $\tau_{l_1}^j/(1-\varepsilon')$ for any $j \in [m]$. Then,

$$\Delta\left(o_{m-c_{l_2}^*} \,\Big|\, A_{l_1}\right) < \frac{\sum_{j=1}^m \tau_{l_1}^j}{(1-\varepsilon')m} \leq \frac{\sum_{j=1}^m \Delta\left(a_{l_1,j} \mid A_{l_1,j-1}\right)}{(1-\varepsilon')m} = \frac{\Delta\left(A_{l_1} \mid G_{i-1}\right)}{(1-\varepsilon')m}. \tag{50}$$

When $|A_{l_1} \setminus G_{i-1}| < m$ and $|A_{l_1} \setminus G_{i-1}| - c_{l_2}^* - 1 < j \leq m - c_{l_2}^*$, this iteration ends with $\tau_{l_1} < \frac{\varepsilon' M}{k}$ and $o_j$ is never considered to be added to $A_{l_1}$. Thus, it holds that

$$\Delta\left(o_j \mid A_{l_1}\right) < \frac{\varepsilon' M}{(1-\varepsilon')k}, \forall |A_{l_1} \setminus G_{i-1}| - c_{l_2}^* - 1 < j \leq m - c_{l_2}^*. \tag{51}$$

Then,

$$\Delta\left(O_{l_2} \mid A_{l_1}\right) \leq \Delta\left(A_{l_2,c_{l_2}^*} \,\Big|\, A_{l_1}\right) + \sum_{o_j \in O_{l_2} \setminus A_{l_2,c_{l_2}^*}} \Delta\left(o_j \mid A_{l_1}\right) \tag{Proposition B.1}$$

$$\leq \Delta\left(A_{l_2,c_{l_2}^*} \,\Big|\, G_{i-1}\right) + \sum_{j=1}^{|A_{l_1}\setminus G_{i-1}|-c_{l_2}^*-1} \Delta\left(o_j \,\Big|\, A_{l_1,c_{l_2}^*+j-1}\right) + \sum_{j=|A_{l_1}\setminus G_{i-1}|-c_{l_2}^*}^{m} \Delta\left(o_j \mid A_{l_1}\right)$$
$$\text{(submodularity)}$$

$$\leq f\left(A_{l_2,c_{l_2}^*}\right) - f\left(G_{i-1}\right) + \sum_{j=1}^{|A_{l_1}\setminus G_{i-1}|-c_{l_2}^*-1} \frac{\Delta\left(a_{l_1,c_{l_2}^*+j} \,\Big|\, A_{l_1,,c_{l_2}^*+j-1}\right)}{1-\varepsilon'} + \frac{\Delta\left(A_{l_1} \mid G_{i-1}\right)}{(1-\varepsilon')m} + \frac{\varepsilon' M}{(1-\varepsilon')\ell}$$
$$\text{(Inequalities (49)-(51))}$$

$$\leq f\left(A_{l_2,c_{l_2}^*}\right) - f\left(G_{i-1}\right) + \frac{f\left(A_{l_1}\right) - f\left(A_{l_1,c_{l_2}^*}\right)}{1-\varepsilon'} + \frac{\Delta\left(A_{l_1} \mid G_{i-1}\right)}{(1-\varepsilon')m} + \frac{\varepsilon' M}{(1-\varepsilon')\ell} \tag{52}$$

By Inequalities (48) and (52),

$$\Delta\left(O_{l_1} \mid A_{l_2}\right) + \Delta\left(O_{l_2} \mid A_{l_1}\right) \leq \frac{1}{1-\varepsilon'}\left(1 + \frac{1}{m}\right)\left(\Delta\left(A_{l_1} \mid G_{i-1}\right) + \Delta\left(A_{l_2} \mid G_{i-1}\right)\right) + \frac{2\varepsilon' M}{(1-\varepsilon')\ell}$$

Thus, the lemma holds in this case. $\square$

*Proof of Lemma F.4.* Fix on $G_{i-1}$ at the beginning of this iteration, Since $\{A_l \setminus G_{i-1} : l \in [\ell]\}$ are pairwise disjoint sets, by Proposition B.1, it holds that

$$\mathbb{E}\left[f\left(O \cup G_i\right) \mid G_{i-1}\right] = \frac{1}{\ell} \sum_{l \in [\ell]} f\left(O \cup A_l\right) \geq \left(1 - \frac{1}{\ell}\right) f\left(O \cup G_{i-1}\right).$$

Then, by unfixing $G_{i-1}$, the first inequality holds.

To prove the second inequality, also consider fix on $G_{i-1}$ at the beginning of iteration $i$. Then,

$$\sum_{l\in[\ell]} \Delta\left(O\mid A_l\right) \leq \sum_{l_1\in[\ell]}\sum_{l_2\in[\ell]} \Delta\left(O_{l_1}\mid A_{l_2}\right) \tag{Proposition B.1}$$

$$= \sum_{l\in[\ell]} \Delta\left(O_l\mid A_l\right) + \sum_{1\leq l_1<l_2\leq\ell}\left(\Delta\left(O_{l_1}\mid A_{l_2}\right) + \Delta\left(O_{l_2}\mid A_{l_1}\right)\right) \tag{Lemma F.3}$$

$$\leq \sum_{l\in[\ell]}\left(\frac{\Delta\left(A_l\mid G_{i-1}\right)}{1-\varepsilon'} + \frac{\varepsilon' M}{(1-\varepsilon')\ell}\right)$$

$$+ \sum_{1\leq l_1<l_2\leq\ell}\left(\frac{1}{1-\varepsilon'}\left(1+\frac{1}{m}\right)\left(\Delta\left(A_{l_1}\mid G_{i-1}\right)+\Delta\left(A_{l_2}\mid G_{i-1}\right)\right) + \frac{2\varepsilon' M}{(1-\varepsilon')\ell}\right) \tag{Lemma F.3}$$

$$\leq \frac{\ell}{1-\varepsilon'}\left(1+\frac{1}{m}\right)\sum_{l\in[\ell]}\Delta\left(A_l\mid G_{i-1}\right) + \frac{\varepsilon'\ell}{1-\varepsilon'}f(O) \tag{$M\leq f(O)$}$$

$$\Rightarrow \left(1+\frac{\ell}{1-\varepsilon'}\right)\left(1+\frac{1}{m}\right)\sum_{l\in[\ell]}\Delta\left(A_l\mid G_{i-1}\right) \geq \sum_{l\in[\ell]} f\left(O\cup A_l\right) - \ell f\left(G_{i-1}\right) - \frac{\varepsilon'\ell}{1-\varepsilon'}f(O)$$

$$\geq (\ell-1) f\left(O\cup G_{i-1}\right) - \ell f\left(G_{i-1}\right) - \frac{\varepsilon'\ell}{1-\varepsilon'}f(O)$$

Thus,

$$\mathbb{E}\left[f\left(G_i\right) - f\left(G_{i-1}\right)\mid G_{i-1}\right] = \frac{1}{\ell}\sum_{l\in[\ell]}\Delta\left(A_l\mid G_{i-1}\right)$$

$$\geq \frac{1}{1+\frac{\ell}{1-\varepsilon'}}\frac{m}{m+1}\left(\left(1-\frac{1}{\ell}\right) f\left(O\cup G_{i-1}\right) - f\left(G_{i-1}\right) - \frac{\varepsilon'}{1-\varepsilon'}f(O)\right) \tag{Proposition B.1}$$

By unfixing $G_{i-1}$, the second inequality holds. $\qquad\square$

# G. Analysis of Parallel Algorithms Introduced in Section 4

This section presents the formal analysis of our parallel algorithms introduced in Section 4. We first prove fundamental lemmata for PIG in Appendix G.1, then establish its approximation guarantees in Appendix G.2, and finally analyze PITG in Appendix G.3.

## G.1. Key Lemmata for PIG (Alg. 6, Section 4)

We provide the key lemmata achieved by PIG as follows,

**Lemma G.1.** With input $(f, m, \ell, \tau_{\min}, \varepsilon)$, PARALLELINTERLACEGREEDY (Alg. 6) runs in $\mathcal{O}\left(\ell^2\varepsilon^{-2}\log(n)\log\left(\frac{M}{\tau_{\min}}\right)\right)$ adaptive rounds and $\mathcal{O}\left(\ell^3\varepsilon^{-2}n\log(n)\log\left(\frac{M}{\tau_{\min}}\right)\right)$ queries with a probability of $1-1/n$, and terminates with $\{(A_l, A'_l): l\in[\ell]\}$ s.t.

1. $A'_l\subseteq A_l$, $\Delta\left(A'_l\mid\emptyset\right)\geq\Delta\left(A_l\mid\emptyset\right)$, $\forall 1\leq l\leq\ell$, and $\{A_l: l\in[\ell]\}$ are pairwise disjoint sets,

2. $\Delta\left(O_l\mid A_l\right)\leq\frac{\Delta\left(A'_l\mid\emptyset\right)}{(1-\varepsilon)^2} + \frac{m\cdot\tau_{\min}}{1-\varepsilon}$, $\forall 1\leq l\leq\ell$,

3. $\Delta\left(O_{l_2}\mid A_{l_1}\right) + \Delta\left(O_{l_1}\mid A_{l_2}\right)\leq\frac{1+\frac{1}{m}}{(1-\varepsilon)^2}\left(\Delta\left(A'_{l_1}\mid\emptyset\right)+\Delta\left(A'_{l_2}\mid\emptyset\right)\right) + \frac{2m\cdot\tau_{\min}}{1-\varepsilon}$, if $O_{l_1} = O_{l_2}$, $\forall 1\leq l_1<l_2\leq\ell$,

where $O_l\subseteq\mathcal{U}$, $|O_l|\leq m$, and $O_l\cap A_j=\emptyset$ for each $j\neq l$.

Especially, when $\ell=2$,

4. $\Delta\left(S \mid A_1\right) + \Delta\left(S \mid A_2\right) \leq \frac{1}{(1-\varepsilon)^2}\left(\Delta\left(A'_{l_1} \mid \emptyset\right) + \Delta\left(A'_{l_2} \mid \emptyset\right)\right) + \frac{2m \cdot \tau_{\min}}{1-\varepsilon}, \forall S \subseteq \mathcal{U}, |S| \leq m.$

Before proving Lemma G.1, we provide the following lemma regarding each iteration of PARALLELINTERLACEGREEDY.

**Lemma G.2.** For any iteration of the while loop in PARALLELINTERLACEGREEDY (Alg. 6), let $A_{l,0}, A'_{l,0}, V_{l,0}, \tau_{l,0}$ be the set and threshold value at the beginning, and $A_l, A'_l, V_l, \tau_l$ be those at the end. The following properties hold.

1. With a probability of at least $1/2$, there exists $l \in [\ell]$ s.t. $\tau_l < \tau_{l,0}$ or $m_0 = 0$ or $|V_l| \leq \left(1 - \frac{\varepsilon}{4\ell}\right)|V_{l,0}|$.

2. $\{A_l : l \in [\ell]\}$ have the same size and are pairwise disjoint.

3. For each $x \in A_l \setminus A_{l,0}$, let $\tau_l^{(x)}$ be the threshold value when $x$ is added to the solution, $A_{l,(x)}$ be the largest prefix of $A_l$ that do not include $x$, and for any $j \in [\ell]$ and $j \neq l$, $A_{j,(x)}$ be the prefix of $A_j$ with $|A_{l,(x)}|$ elements if $j < l$, or with $|A_{l,(x)}| - 1$ elements if $j > l$. Then, for any $l \in [\ell]$, $x \in A_l \setminus A_{l,0}$, and $y \in \mathcal{U} \setminus \left(\bigcup_{j \in [\ell]} A_{j,(x)}\right)$, it holds that
$$\Delta\left(y \mid A_{l,(x)}\right) < \frac{\tau_l^{(x)}}{1-\varepsilon}.$$

4. $A'_l \subseteq A_l$, $\Delta\left(A'_l \mid A'_{l,0}\right) \geq \Delta\left(A_l \mid A_{l,0}\right)$, and $\Delta\left(A'_l \mid A'_{l,0}\right) \geq (1-\varepsilon)\sum_{x \in A_l \setminus A_{l,0}} \tau_l^{(x)}$ for all $l \in [\ell]$.

*Proof of Lemma G.2.* **Proof of Property 1.** At the beginning of the iteration, if there exists $l \in I$ s.t. $|V_{l,0}| < 2\ell$, then either $\tau_{l,0}$ is decreased to $\tau_l$ and $V_l$ is updated accordingly, or an element $x_l$ from $V_{l,0}$ is added to $A_j$ and $A'_j$ and subsequently removed from $V_{l,0}$. This implies that
$$|V_l| \leq |V_{l,0}| - 1 < \left(1 - \frac{1}{2\ell}\right)|V_{l,0}|.$$

Property 1 holds in this case.

Otherwise, for all $l \in I$, it holds that $|V_{l,0}| \geq 2\ell$, and the algorithm proceeds to execute Lines 16-24. By Lemma 4.3, in Line 16, $|\mathcal{V}_l| \geq \frac{|V_{l,0}|}{2\ell}$ for each $l \in I$. Consider the index $j \in I$ where $i_j^* = i^*$. Then, $O_j$ consists of the first $i^*$ elements in $\mathcal{V}_j$ by Line 21. By Lemma 4.5, with probability greater than $1/2$, either $i^* = m_0$ or at least an $\frac{\varepsilon}{2}$-fraction of elements $x \in \mathcal{V}_j$ satisfy $\Delta\left(x \mid A_j\right) < \tau_{j,0}$. Consequently, either $m_0 = 0$ after Line 24, or, after the UPDATE procedure in Line 4, one of the following holds: $|V_l| \leq \left(1 - \frac{\varepsilon}{4\ell}\right)|V_{l,0}|$, or $\tau_j < \tau_{j,0}$. Therefore, Property 1 holds in this case.

**Proof of Property 2.** At any iteration of the while, either $|I|$ different elements or $|I|$ pairwise disjoint sets with same size $i^*$ are added to solution sets $\{A_l : l \in I\}$. Therefore, Property 2 holds.

**Proof of Property 3.** At any iteration, if $\tau_l$ is not updated on Line 8, then prior to this iteration, all the elements outside of the solutions have marginal gain less than $\frac{\tau_l^{(x)}}{1-\varepsilon}$. Thus, for any $x \in A_l \setminus A_{l,0}$, $y \in \mathcal{U} \setminus \left(\bigcup_{j \in [\ell]} A_{j,0}\right)$, it holds that $\Delta\left(y \mid A_{l,(x)}\right) < \frac{\tau_l^{(x)}}{1-\varepsilon}$ by submodularity. Property 3 holds in this case.

Otherwise, if $\tau_l$ is updated on Line 8, only one element is added to each solution set during this iteration. Let $x = A_l \setminus A_{l,0}$. For any $j \in [\ell]$ and $j \neq l$, it holds that $A_{j,(x)} = A_j$ if $j < l$, or $A_{j,(x)} = A_{j,0}$ if $j > l$. Since elements are added to each pair of solutions in sequence within the for loop in Lines 7-13, by the UPDATE procedure, for any $y \in \mathcal{U} \setminus \left(\bigcup_{j \in [\ell]} A_{j,(x)}\right)$, it holds that $\Delta\left(y \mid A_{l,(x)}\right) < \frac{\tau_l^{(x)}}{1-\varepsilon}$. Therefore, Property 3 also holds in this case.

**Proof of Property 4.** First, we prove $A'_l \subseteq A_l$ by induction. At the beginning of the algorithm, $A'_l$ and $A_l$ are initialized as empty sets. Clearly, the property holds in the base case. Then, suppose that $A'_{l,0} \subseteq A_{l,0}$. There are three possible cases of updating $A'_{l,0}$ and $A_{l,0}$ at any iteration: 1) $A'_l = A'_{l,0}$ and $A_l = A_{l,0}$, 2) $A'_l = A'_{l,0} + x_l$ and $A_l = A_{l,0} + x_l$ in Line 12, or 3) $A'_l = A'_{l,0} \cup S'_l$ and $A_l = A_{l,0} \cup S_l$ in Line 23. Clearly, $A'_l \subseteq A_l$ holds in all cases.

Next, we prove the rest of Property 4.

If $A'_l = A'_{l,0}$ and $A_l = A_{l,0}$, then $\Delta\left(A'_l \mid A'_{l,0}\right) = \Delta\left(A_l \mid A_{l,0}\right) = 0$. Property 4 holds.

If $A'_{l,0}$ and $A_{l,0}$ are updated in Line 12, by submodularity, $\Delta\left(A'_l \mid A'_{l,0}\right) = \Delta\left(x_l \mid A'_{l,0}\right) \geq \Delta\left(x_l \mid A_{l,0}\right) = \Delta\left(A_l \mid A_{l,0}\right) \geq \tau_l^{(x_l)}$. Therefore, Property 4 also holds.

If $A'_{l,0}$ and $A_{l,0}$ are updated in Line 23, we know that $A'_l = A'_{l,0} \cup S'_l$ and $A_l = A_{l,0} \cup S_l$. Suppose the elements in $S_l$ and $S'_l$ retain their original order within $\mathcal{V}_l$. For each $x \in S_l$, let $S_{l,(x)}$, $\mathcal{V}_{l,(x)}$ and $A_{l,(x)}$ be the largest prefixes of $S_l$, $\mathcal{V}_l$ and $A_l$ that do not include $x$, respectively. Moreover, let $S'_{l,(x)} = S_{l,(x)} \cap S'_l$ and $A'_{l,(x)} = A_{l,(x)} \cap A'_l$. Say an element $x \in S_l$ **true** if $B_l[(x)] = $ **true**, where $B_l[(x)]$ is the $i$-th element in $B_l$ if $x$ is the $i$-th element in $\mathcal{V}_l$. Similarly, say an element $x \in S_l$ **false** if $B_l[(x)] = $ **false**, and **none** otherwise.

Following the above definitions, for any **true** or **none** element $x \in S_l$, by Line 21, it holds that $S_{l,(x)} \subseteq \mathcal{V}_{l,(x)}$. Then, by Line 5 and submodularity,

$$\Delta\left(x \mid A_{l,(x)}\right) = \Delta\left(x \mid A_{l,0} \cup S_{l,(x)}\right) \geq \Delta\left(x \mid A_{l,0} \cup \mathcal{V}_{l,(x)}\right) \geq \begin{cases} \tau_l^{(x)}, & \text{if } x \text{ is \textbf{true} element} \\ 0, & \text{if } x \text{ is \textbf{none} element} \end{cases}$$

Since **true** elements are selected at first and $i_j^* \geq i^*$, there are more than $(1-\varepsilon)i^*$ **true** elements in $S_l$. Therefore,

$$\begin{aligned}
\Delta\left(A'_l \mid A'_{l,0}\right) &= \sum_{x \in A'_l \setminus A'_{l,0}, x \text{ is \textbf{true} element}} \Delta\left(x \mid A'_{l,(x)}\right) + \sum_{x \in A'_l \setminus A'_{l,0}, x \text{ is \textbf{none} element}} \Delta\left(x \mid A'_{l,(x)}\right) \\
&\geq \sum_{x \in A'_l \setminus A'_{l,0}, x \text{ is \textbf{true} element}} \Delta\left(x \mid A_{l,(x)}\right) + \sum_{x \in A'_l \setminus A'_{l,0}, x \text{ is \textbf{none} element}} \Delta\left(x \mid A_{l,(x)}\right) \\
&\geq (1-\varepsilon)|A_l \setminus A_{l,0}|\tau_l^{(x)}, \text{for any } x \in A_l \setminus A_{l,0} \\
&= (1-\varepsilon) \sum_{x \in A_l \setminus A_{l,0}} \tau_l^{(x)}.
\end{aligned}$$

The third part of Property 4 holds.

To prove the second part of Property 4, consider any **false** element $x \in S_l$. By Line 21, it holds that $\mathcal{V}_{l,(x)} = S_{l,(x)}$. Then, by Line 6

$$\Delta\left(x \mid A_{l,(x)}\right) = \Delta\left(x \mid A_{l,0} \cup S_{l,(x)}\right) = \Delta\left(x \mid A_{l,0} \cup \mathcal{V}_{l,(x)}\right) < 0. \tag{53}$$

By Line 22, all the elements in $S_l \setminus S'_l$ are **false** elements. Then,

$$\begin{aligned}
f\left(A_l\right) - f\left(A_{l,0}\right) &= \sum_{x \in S'_l} \Delta\left(x \mid A_{l,(x)}\right) + \sum_{x \in S_l \setminus S'_l} \Delta\left(x \mid A_{l,(x)}\right) \\
&< \sum_{x \in S'_l} \Delta\left(x \mid A_{l,(x)}\right) && \text{(Inequality 53)} \\
&\leq \sum_{x \in S'_l} \Delta\left(x \mid A'_{l,(x)}\right) && \text{(Submodularity)} \\
&= f\left(A'_l\right) - f\left(A'_{l,0}\right).
\end{aligned}$$

$\square$

By Lemma G.2, we are ready to prove Lemma G.1.

*Proof of Lemma G.1.* **Proof of Property 1.** By Property 2 and 3 in Lemma G.2, this property holds immediately.

**Proof of Property 2.** For any $l \in [\ell]$, since $O_l \cap A_j = \emptyset$ for each $j \neq l$, $O_l \setminus A_l$ is outside of any solution set. If $|A_l| = m$, by Property 4 of Lemma G.2,

$$\begin{aligned}
\Delta\left(O_l \mid A_l\right) &\leq \sum_{y \in O_l \setminus A_l} \Delta\left(y \mid A_l\right) \\
&\leq \sum_{x \in A_l} \tau_l^{(x)}/(1-\varepsilon) && \text{(Property 3 in Lemma G.2)} \\
&\leq \frac{\Delta\left(A'_l \mid \emptyset\right)}{(1-\varepsilon)^2}. && \text{(Property 5 in Lemma G.2)}
\end{aligned}$$

If $|A_l| < m$, then the threshold value for solution $A_l$ has been updated to be less than $\tau_{\min}$. Therefore, for any $y \in O_l \setminus A_l$, it holds that $\Delta(y \mid A_l) < \frac{\tau_{\min}}{1-\varepsilon}$. Then,

$$\Delta(O_l \mid A_l) \leq \sum_{y \in O_l \setminus A_l} \Delta(y \mid A_l) \leq \frac{m\tau_{\min}}{1-\varepsilon}.$$

Therefore, Property 2 holds by summing the above two inequalities.

**Proof of Property 3 and 4.** Let $a_{l,j}$ be the $j$-th element added to $A_l$, $\tau_l^j$ be the threshold value of $\tau_l$ when $a_{l,j}$ is added to $A_l$, and $A_{l,j}$ be $A_l$ after $a_{l,j}$ is added to $A_l$. Let $c_l^* = \max\{c \in [m] : A_{l,c} \subseteq O_l\}$.

In the following, we analyze these properties together under two cases, similar to the analysis of Alg. 11. For the case where $\ell = 2$, let $O_1 = S \setminus A_2$, and $O_2 = S \setminus A_1$, unifying the notations used in Property 3 and 4. Note that, the only difference between the two analyses is that, a small portion (no more than $\varepsilon$ fraction) of elements in the solution returned by Alg. 6 do not have marginal gain greater than the threshold value.

**Case 1:** $c_{l_1}^* \leq c_{l_2}^*$; left half part in Fig. 4.

First, we bound $\Delta(O_{l_1} \mid A_{l_2})$. Consider elements in $A_{l_1,c_{l_1}^*} \subseteq O_{l_1}$. Let $A_{l_1,c_{l_1}^*} = \{o_1, \ldots, o_{c_{l_1}^*}\}$. For each $1 \leq j \leq c_{l_1}^*$, since $o_j$ is added to $A_{l_1}$ with threshold value $\tau_{l_1}^j$ and the threshold value starts from the maximum marginal gain $M$, clearly, $o_j$ has been filtered out with threshold value $\tau_{l_1}^j/(1-\varepsilon)$. Then, by submodularity,

$$\Delta\left(A_{l_1,c_{l_1}^*} \mid A_{l_2}\right) \leq \Delta\left(A_{l_1,c_{l_1}^*} \mid \emptyset\right) = \sum_{j=1}^{c_{l_1}^*} \Delta(o_j \mid A_{l_1,j-1}) \leq \sum_{j=1}^{c_{l_1}^*} \tau_{l_1}^j/(1-\varepsilon). \tag{54}$$

Next, consider the elements in $O_{l_1} \setminus A_{l_1,c_{l_1}^*}$. Order the elements in $O_{l_1} \setminus A_{l_1,c_{l_1}^*}$ as $\{o_1, o_2, \ldots\}$ such that $o_j \notin A_{l_1,c_{l_1}^*+j}$. (Refer to the gray block with a dotted edge in the top left corner of Fig. 4 for $O_{l_1}$. If $c_{l_1}^* + j$ is greater than $|A_{l_1}|$, $A_{l_1,c_{l_1}^*+j}$ refers to $A_{l_1}$.) Note that, since $A_{l_1,c_{l_1}^*} \subseteq O_{l_1}$, it follows that $|O_{l_1} \setminus A_{l_1,c_{l_1}^*}| \leq m - c_{l_1}^*$.

When $1 \leq j \leq |A_{l_2}| - c_{l_1}^*$, since each $o_j$ is either added to $A_{l_1}$ or not in any solution set and $\tau_{l_2}$ is initialized with the maximum marginal gain $M$, $o_j$ is not considered to be added to $A_{l_2}$ with threshold value $\tau_{l_2}^{c_{l_1}^*+j}/(1-\varepsilon)$ by Property 3 of Lemma G.2. Therefore, it holds that

$$\Delta\left(o_j \mid A_{l_2,c_{l_1}^*+j-1}\right) < \frac{\tau_{l_2}^{c_{l_1}^*+j}}{1-\varepsilon}, \forall 1 \leq j \leq |A_{l_2}| - c_{l_1}^*. \tag{55}$$

When $|A_{l_2}| < m$ and $|A_{l_2}| - c_{l_1}^* < j \leq m - c_{l_1}^*$, the algorithm ends with $\tau_{l_2} < \tau_{\min}$ and $o_j$ is never considered to be added to $A_{l_2}$. Thus, it holds that

$$\Delta(o_j \mid A_{l_2}) < \frac{\tau_{\min}}{1-\varepsilon}, \forall |A_{l_2}| - c_{l_1}^* < j \leq m - c_{l_1}^*. \tag{56}$$

Then,

$$\Delta(O_{l_1} \mid A_{l_2}) \leq \Delta\left(A_{l_1,c_{l_1}^*} \mid A_{l_2}\right) + \sum_{o_j \in O_{l_1} \setminus A_{l_1,c_{l_1}^*}} \Delta(o_j \mid A_{l_2}) \qquad \text{(Proposition B.1)}$$

$$\leq \Delta\left(A_{l_1,c_{l_1}^*} \mid \emptyset\right) + \sum_{j=1}^{|A_{l_2}|-c_{l_1}^*} \Delta\left(o_j \mid A_{l_2,,c_{l_1}^*+j-1}\right) + \sum_{j=|A_{l_2}|-c_{l_1}^*+1}^{m-c_{l_1}^*} \Delta(o_j \mid A_{l_2}) \qquad \text{(submodularity)}$$

$$\leq \sum_{j=1}^{c_{l_1}^*} \frac{\tau_{l_1}^j}{1-\varepsilon} + \sum_{j=c_{l_1}^*+1}^{|A_{l_2}|} \frac{\tau_{l_2}^j}{1-\varepsilon} + \frac{m \cdot \tau_{\min}}{1-\varepsilon} \tag{57}$$

where the last inequality follows from Inequalities (54)-(56).

Similarly, we bound $\Delta\left(O_{l_2} \mid A_{l_1}\right)$ below. Consider elements in $A_{l_2,c_{l_1}^*} \subseteq O_{l_2}$. Let $A_{l_2,c_{l_1}^*} = \{o_1, \ldots, o_{c_{l_1}^*}\}$. For each $1 \leq j \leq c_{l_1}^*$, since $o_j$ is added to $A_{l_2}$ with threshold value $\tau_{l_2}^j$ and the threshold value starts from the maximum marginal gain $M$, clearly, $o_j$ has been filtered out with threshold value $\tau_{l_2}^j/(1-\varepsilon)$. Then, by submodularity,

$$\Delta\left(A_{l_2,c_{l_1}^*} \mid A_{l_1}\right) \leq \Delta\left(A_{l_2,c_{l_1}^*} \mid \emptyset\right) = \sum_{j=1}^{c_{l_1}^*} \Delta\left(o_j \mid A_{l_2,j-1}\right) \leq \sum_{j=1}^{c_{l_1}^*} \tau_{l_2}^j/(1-\varepsilon). \tag{58}$$

Next, consider the elements in $O_{l_2} \setminus A_{l_2,c_{l_1}^*}$. Order the elements in $O_{l_2} \setminus A_{l_2,c_{l_1}^*}$ as $\{o_1, o_2, \ldots\}$ such that $o_j \notin A_{l_2,c_{l_1}^*+j-1}$. (See the gray block with a dotted edge in the bottom left corner of Fig. 4 for $O_{l_2}$. If $c_{l_1}^* + j - 1$ is greater than $|A_{l_2}|$, $A_{l_2,c_{l_1}^*+j-1}$ refers to $A_{l_2}$.) Note that, since $A_{l_2,c_{l_1}^*} \subseteq O_{l_2}$, it follows that $|O_{l_2} \setminus A_{l_2,c_{l_1}^*}| \leq m - c_{l_1}^*$.

When $1 \leq j \leq |A_{l_1}| - c_{l_1}^*$, since each $o_j$ is either added to $A_{l_2}$ or not in any solution set, and $\tau_{l_1}$ is initialized with the maximum marginal gain $M$, $o_j$ is not considered to be added to $A_{l_1}$ with threshold value $\tau_{l_1}^{c_{l_1}^*+j}/(1-\varepsilon)$ by Property 3 of Lemma G.2. Therefore, it holds that

$$\Delta\left(o_j \mid A_{l_1,c_{l_1}^*+j-1}\right) < \frac{\tau_{l_1}^{c_{l_1}^*+j}}{1-\varepsilon}, \forall 1 \leq j \leq |A_{l_2}| - c_{l_1}^*. \tag{59}$$

When $|A_{l_1}| < m$ and $|A_{l_1}| - c_{l_1}^* < j \leq m - c_{l_1}^*$, this iteration ends with $\tau_{l_1} < \tau_{\min}$ and $o_j$ is never considered to be added to $A_{l_1}$. Thus, it holds that

$$\Delta\left(o_j \mid A_{l_1}\right) < \frac{\tau_{\min}}{1-\varepsilon}, \forall |A_{l_1}| - c_{l_1}^* < j \leq m - c_{l_1}^*. \tag{60}$$

Then,

$$\Delta\left(O_{l_2} \mid A_{l_1}\right) \leq \Delta\left(A_{l_2,c_{l_1}^*} \mid A_{l_1}\right) + \sum_{o_j \in O_{l_2} \setminus A_{l_2,c_{l_1}^*}} \Delta\left(o_j \mid A_{l_1}\right) \tag{Proposition B.1}$$

$$\leq \Delta\left(A_{l_2,c_{l_1}^*} \mid \emptyset\right) + \sum_{j=1}^{|A_{l_1}|-c_{l_1}^*} \Delta\left(o_j \mid A_{l_1,c_{l_1}^*+j-1}\right) + \sum_{j=|A_{l_1}|-c_{l_1}^*+1}^{m-c_{l_1}^*} \Delta\left(o_j \mid A_{l_1}\right) \tag{submodularity}$$

$$\leq \sum_{j=1}^{c_{l_1}^*} \frac{\tau_{l_2}^j}{1-\varepsilon} + \sum_{j=c_{l_1}^*+1}^{|A_{l_1}|} \frac{\tau_{l_1}^j}{1-\varepsilon} + \frac{m \cdot \tau_{\min}}{1-\varepsilon} \tag{61}$$

where the last inequality follows from Inequalities 58-60.

By Inequalities (57) and (61),

$$\Delta\left(O_{l_1} \mid A_{l_2}\right) + \Delta\left(O_{l_2} \mid A_{l_1}\right) \leq \sum_{j=1}^{|A_{l_1}|} \frac{\tau_{l_1}^j}{1-\varepsilon} + \sum_{j=1}^{|A_{l_2}|} \frac{\tau_{l_2}^j}{1-\varepsilon} + \frac{2m \cdot \tau_{\min}}{1-\varepsilon} \tag{62}$$

**Case 2: $c_{l_1}^* > c_{l_2}^*$; right half part in Fig. 4.**

First, we bound $\Delta\left(O_{l_1} \mid A_{l_2}\right)$. Consider elements in $A_{l_1,c_{l_2}^*+1} \subseteq O_{l_1}$. Let $A_{l_1,c_{l_2}^*+1} = \{o_1, \ldots, o_{c_{l_2}^*+1}\}$. For each $1 \leq j \leq c_{l_2}^* + 1$, since $o_j$ is added to $A_{l_1}$ with threshold value $\tau_{l_1}^j$ and the threshold value starts from the maximum marginal gain $M$, clearly, $o_j$ has been filtered out with threshold value $\tau_{l_1}^j/(1-\varepsilon)$. Then, by submodularity,

$$\Delta\left(A_{l_1,c_{l_2}^*+1} \mid A_{l_2}\right) \leq \Delta\left(A_{l_1,c_{l_2}^*+1} \mid \emptyset\right) = \sum_{j=1}^{c_{l_2}^*+1} \Delta\left(o_j \mid A_{l_1,j-1}\right) \leq \sum_{j=1}^{c_{l_2}^*+1} \tau_{l_1}^j/(1-\varepsilon). \tag{63}$$

Next, consider the elements in $O_{l_1} \setminus A_{l_1, c_{l_2}^* + 1}$. Order the elements in $O_{l_1} \setminus A_{l_1, c_{l_2}^* + 1}$ as $\{o_1, o_2, \ldots\}$ such that $o_j \notin A_{l_1, c_{l_1}^* + j}$. (Refer to the gray block with a dotted edge in the top right corner of Fig. 4 for $O_{l_1}$. If $c_{l_1}^* + j$ is greater than $|A_{l_1}|$, $A_{l_1, c_{l_1}^* + j}$ refers to $A_{l_1}$.) Note that, since $A_{l_1, c_{l_2}^* + 1} \subseteq O_{l_1}$, it follows that $|O_{l_1} \setminus A_{l_1, c_{l_2}^* + 1}| \le m - c_{l_2}^* - 1$.

When $1 \le j \le |A_{l_2}| - c_{l_2}^* - 1$, since each $o_j$ is either added to $A_{l_1}$ or not in any solution set and $\tau_{l_2}$ is initialized with the maximum marginal gain $M$, $o_j$ is not considered to be added to $A_{l_2}$ with threshold value $\tau_{l_2}^{c_{l_2}^* + j}/(1 - \varepsilon)$. Therefore, it holds that

$$\Delta\left(o_j \,\middle|\, A_{l_2, c_{l_2}^* + j - 1}\right) < \frac{\tau_{l_2}^{c_{l_2}^* + j}}{1 - \varepsilon}, \forall 1 \le j \le |A_{l_2} \setminus G_{i-1}| - c_{l_2}^* - 1. \tag{64}$$

When $|A_{l_2}| < m$ and $|A_{l_2}| - c_{l_2}^* - 1 < j \le m - c_{l_2}^* - 1$, this iteration ends with $\tau_{l_2} < \tau_{\min}$ and $o_j$ is never considered to be added to $A_{l_2}$. Thus, it holds that

$$\Delta\left(o_j \,|\, A_{l_2}\right) < \frac{\tau_{\min}}{1 - \varepsilon}, \forall |A_{l_2}| - c_{l_2}^* - 1 < j \le m - c_{l_2}^* - 1. \tag{65}$$

Then,

$$\Delta\left(O_{l_1} \,|\, A_{l_2}\right) \le \Delta\left(A_{l_1, c_{l_2}^*} \,\middle|\, A_{l_2}\right) + \sum_{o_j \in O_{l_1} \setminus A_{l_1, c_{l_2}^* + 1}} \Delta\left(o_j \,|\, A_{l_2}\right) \tag{Proposition B.1}$$

$$\le \Delta\left(A_{l_1, c_{l_2}^*} \,\middle|\, \emptyset\right) + \sum_{j=1}^{|A_{l_2}| - c_{l_2}^* - 1} \Delta\left(o_j \,\middle|\, A_{l_2, c_{l_2}^* + j - 1}\right) + \sum_{j = |A_{l_2}| - c_{l_2}^*}^{m - c_{l_2}^* - 1} \Delta\left(o_j \,|\, A_{l_2}\right) \tag{submodularity}$$

$$\le \sum_{j=1}^{c_{l_2}^* + 1} \tau_{l_1}^j / (1 - \varepsilon) + \sum_{j = c_{l_2}^* + 1}^{|A_{l_2}|} \tau_{l_2}^j / (1 - \varepsilon) + \frac{m \cdot \tau_{\min}}{1 - \varepsilon} \tag{66}$$

where the last inequality follows from Inequalities (63)-(65).

Similarly, we bound $\Delta\left(O_{l_2} \,|\, A_{l_1}\right)$ below. Consider elements in $A_{l_1, c_{l_2}^*} \subseteq O_{l_2}$. Let $A_{l_2, c_{l_2}^*} = \{o_1, \ldots, o_{c_{l_2}^*}\}$. For each $1 \le j \le c_{l_2}^*$, since $o_j$ is added to $A_{l_2}$ with threshold value $\tau_{l_2}^j$ and the threshold value starts from the maximum marginal gain $M$, clearly, $o_j$ has been filtered out with threshold value $\tau_{l_2}^j / (1 - \varepsilon)$. Then, by submodularity,

$$\Delta\left(A_{l_2, c_{l_2}^*} \,\middle|\, A_{l_1}\right) \le \Delta\left(A_{l_2, c_{l_2}^*} \,\middle|\, \emptyset\right) = \sum_{j=1}^{c_{l_2}^*} \Delta\left(o_j \,|\, A_{l_2, j-1}\right) \le \sum_{j=1}^{c_{l_2}^*} \tau_{l_2}^j / (1 - \varepsilon). \tag{67}$$

Next, consider the elements in $O_{l_2} \setminus A_{l_2, c_{l_2}^*}$. Order these elements as $\{o_1, o_2, \ldots\}$ such that $o_j \notin A_{l_2, c_{l_2}^* + j}$. (See the gray block with a dotted edge in the bottom right corner of Fig. 4 for $O_{l_2}$. If $c_{l_2}^* + j$ is greater than the number of elements added to $A_{l_2}$, $A_{l_2, c_{l_2}^* + j}$ refers to $A_{l_2}$.) Note that, since $A_{l_2, c_{l_2}^*} \subseteq O_{l_2}$, it follows that $|O_{l_2} \setminus A_{l_2, c_{l_2}^*}| \le m - c_{l_2}^*$.

Furthermore, for the case where $\ell = 2$, as considered in Property 4, we have $O_1 = S \setminus A_2$ and $O_2 = S \setminus A_1$ for a given $S \subseteq \mathcal{U}$ where $|S| \le m$. Since $c_{l_1}^* > c_{l_2}^* \ge 0$, it follows that $c_{l_1}^* \ge 1$, which implies $|O_2| = |S \setminus A_1| \le m - 1$. In this case, it holds that $|O_{l_2} \setminus A_{l_2, c_{l_2}^*}| \le m - c_{l_2}^* - 1$.

When $1 \le j \le |A_{l_1}| - c_{l_2}^* - 1$, since each $o_j$ is either added to $A_{l_2}$ or not in any solution set by Claim 3.1 and $\tau_{l_1}$ is initialized with the maximum marginal gain $M$, $o_j$ is not considered to be added to $A_{l_1}$ with threshold value $\tau_{l_1}^{c_{l_2}^* + j + 1}/(1 - \varepsilon)$. Therefore, it holds that

$$\Delta\left(o_j \,\middle|\, A_{l_1, c_{l_2}^* + j}\right) < \frac{\tau_{l_1}^{c_{l_2}^* + j + 1}}{1 - \varepsilon}, \forall 1 \le j \le |A_{l_2}| - c_{l_2}^* - 1. \tag{68}$$

If $|A_{l_1}| = m$, consider the last element $o_{m - c_{l_2}^*}$ in $O_{l_2} \setminus A_{l_2, c_{l_2}^*}$. Since $o_{m - c_{l_2}^*} \notin A_{l_2}$ and $o_{m - c_{l_2}^*} \notin A_{l_1}$, $o_{m - c_{l_2}^*}$ is not

considered to be added to $A_{l_1}$ with threshold value $\tau_{l_1}^j/(1-\varepsilon)$ for any $j \in [m]$. Then,

$$\Delta\left(o_{m-c_{l_2}^*} \,\middle|\, A_{l_1}\right) < \frac{\sum_{j=1}^m \tau_{l_1}^j}{(1-\varepsilon)m}. \tag{69}$$

Else, $|A_{l_1}| < m$ and this iteration ends with $\tau_{l_1} < \frac{\varepsilon M}{k}$. For any $|A_{l_1}| - c_{l_2}^* - 1 < j \le m - c_{l_2}^*$, $o_j$ is never considered to be added to $A_{l_1}$. Thus, it holds that

$$\Delta\left(o_j \mid A_{l_1}\right) < \frac{\tau_{\min}}{1-\varepsilon}, \forall |A_{l_1}| - c_{l_2}^* - 1 < j \le m - c_{l_2}^*. \tag{70}$$

Then,

$$\Delta\left(O_{l_2} \mid A_{l_1}\right) \le \Delta\left(A_{l_2, c_{l_2}^*} \,\middle|\, A_{l_1}\right) + \sum_{o_j \in O_{l_2} \setminus A_{l_2, c_{l_2}^*}} \Delta\left(o_j \mid A_{l_1}\right) \qquad \text{(Proposition B.1)}$$

$$\le \begin{cases} \Delta\left(A_{l_2, c_{l_2}^*} \,\middle|\, \emptyset\right) + \displaystyle\sum_{j=1}^{|A_{l_1} \setminus G_{i-1}| - c_{l_2}^* - 1} \Delta\left(o_j \,\middle|\, A_{l_1, c_{l_2}^* + j - 1}\right) + \displaystyle\sum_{j=|A_{l_1} \setminus G_{i-1}| - c_{l_2}^*}^{m - c_{l_2}^*} \Delta\left(o_j \mid A_{l_1}\right), & \text{if } |O_{l_2}| = m \\[3em] \Delta\left(A_{l_2, c_{l_2}^*} \,\middle|\, \emptyset\right) + \displaystyle\sum_{j=1}^{|A_{l_1} \setminus G_{i-1}| - c_{l_2}^* - 1} \Delta\left(o_j \,\middle|\, A_{l_1, c_{l_2}^* + j - 1}\right) + \displaystyle\sum_{j=|A_{l_1} \setminus G_{i-1}| - c_{l_2}^*}^{m - c_{l_2}^* - 1} \Delta\left(o_j \mid A_{l_1}\right), & \text{otherwise} \end{cases}$$

$$\text{(submodularity)}$$

$$\le \begin{cases} \displaystyle\sum_{j=1}^{c_{l_2}^*} \frac{\tau_{l_2}^j}{1-\varepsilon} + \displaystyle\sum_{j=c_{l_2}^*+2}^{|A_{l_2}|} \left(1 + \frac{1}{m}\right) \frac{\tau_{l_1}^j}{1-\varepsilon} + \frac{m \cdot \tau_{\min}}{1-\varepsilon}, & \text{if } |O_{l_2}| = m \\[3em] \displaystyle\sum_{j=1}^{c_{l_2}^*} \frac{\tau_{l_2}^j}{1-\varepsilon} + \displaystyle\sum_{j=c_{l_2}^*+2}^{|A_{l_2}|} \frac{\tau_{l_1}^j}{1-\varepsilon} + \frac{m \cdot \tau_{\min}}{1-\varepsilon}, & \text{otherwise} \end{cases} \tag{71}$$

where the last inequality follows from Inequalities (67)-(70).

By Inequalities (66) and (71),

$$\Delta\left(O_{l_1} \mid A_{l_2}\right) + \Delta\left(O_{l_2} \mid A_{l_1}\right) \le \begin{cases} \left(1 + \dfrac{1}{m}\right) \dfrac{1}{1-\varepsilon} \left(\displaystyle\sum_{j=1}^{|A_{l_1}|} \tau_{l_1}^j + \sum_{j=1}^{|A_{l_2}|} \tau_{l_2}^j\right) + \dfrac{2m \cdot \tau_{\min}}{1-\varepsilon}, & \text{if } |O_{l_2}| = m \\[3em] \dfrac{1}{1-\varepsilon} \left(\displaystyle\sum_{j=1}^{|A_{l_1}|} \tau_{l_1}^j + \sum_{j=1}^{|A_{l_2}|} \tau_{l_2}^j\right) + \dfrac{2m \cdot \tau_{\min}}{1-\varepsilon}, & \text{otherwise} \end{cases} \tag{72}$$

Overall, in both cases, if $|O_{l_2}| = m$,

$$\Delta\left(O_{l_1} \mid A_{l_2}\right) + \Delta\left(O_{l_2} \mid A_{l_1}\right) \le \left(1 + \frac{1}{m}\right) \frac{1}{1-\varepsilon} \left(\sum_{j=1}^{|A_{l_1}|} \tau_{l_1}^j + \sum_{j=1}^{|A_{l_2}|} \tau_{l_2}^j\right) + \frac{2m \cdot \tau_{\min}}{1-\varepsilon} \qquad \text{(Inequalities (62) and (72))}$$

$$\le \left(1 + \frac{1}{m}\right) \frac{1}{(1-\varepsilon)^2} \left(\Delta\left(A_{l_1}' \mid \emptyset\right) + \Delta\left(A_{l_2}' \mid \emptyset\right)\right) + \frac{2m \cdot \tau_{\min}}{1-\varepsilon} \qquad \text{(Property 4 of Lemma G.2)}$$

Otherwise, if $|O_{l_2}| < m$,

$$\Delta\left(O_{l_1} \mid A_{l_2}\right) + \Delta\left(O_{l_2} \mid A_{l_1}\right) \le \frac{1}{1-\varepsilon} \left(\sum_{j=1}^{|A_{l_1}|} \tau_{l_1}^j + \sum_{j=1}^{|A_{l_2}|} \tau_{l_2}^j\right) + \frac{2m \cdot \tau_{\min}}{1-\varepsilon} \qquad \text{(Inequalities (62) and (72))}$$

$$\leq \frac{1}{(1-\varepsilon)^2} \left( \Delta \left( A'_{l_1} \mid \emptyset \right) + \Delta \left( A'_{l_2} \mid \emptyset \right) \right) + \frac{2m \cdot \tau_{\min}}{1-\varepsilon} \qquad \text{(Property 4 of Lemma G.2)}$$

Property (3) and (4) hold.

**Proof of Adaptivity and Query Complexity.** Note that, at the beginning of every iteration, for any $j \in I$, $V_j$ contains all the elements outside of all solutions that has marginal gain greater than $\tau_j$ with respect to solution $A_j$. Say an iteration *successful* if either 1) algorithm terminates after this iteration because of $m_0 = 0$, 2) all the elements in $V_j$ can be filtered out at the end of this iteration and the value of $\tau_j$ decreases, or 3) the size of $V_j$ decreases by a factor of $1 - \frac{\varepsilon}{4\ell}$. Then, by Property 1 of Lemma G.2, with a probability of at least $1/2$, the iteration is successful. Furthermore, if $\tau_j$ is less than $\tau_{\min}$, $j$ will be removed from $I$ and solutions $A_j$ and $A'_j$ won't be updated anymore.

For each $j \in [\ell]$, there are at most $\log_{1-\varepsilon} \left( \frac{\tau_{\min}}{M} \right) \leq \varepsilon^{-1} \log \left( \frac{M}{\tau_{\min}} \right)$ possible threshold values. And, for each threshold value, with at most $\log_{1-\frac{\varepsilon}{4\ell}} \left( \frac{1}{n} \right) \leq 4\ell\varepsilon^{-1} \log(n)$ successful iterations regarding solution $A_j$, the threshold value $\tau_j$ will decrease or the algorithm terminates because of $m_0 = 0$. Overall, with at most $4\ell^2\varepsilon^{-2} \log(n) \log \left( \frac{M}{\tau_{\min}} \right)$ successful iterations, the algorithm terminates because of $m_0 = 0$ or $I = \emptyset$.

Next, we prove that, after $N = 4 \left( \log(n) + 4\ell^2\varepsilon^{-2} \log(n) \log \left( \frac{M}{\tau_{\min}} \right) \right)$ iterations, with a probability of $1 - \frac{1}{n}$, there exists at least $4\ell^2\varepsilon^{-2} \log(n) \log \left( \frac{M}{\tau_{\min}} \right)$ successful iterations, or equivalently, the algorithm terminates. Let $X$ be the number of successful iterations. Then, $X$ can be regarded as a sum of $N$ dependent Bernoulli trails, where the success probability is larger than $1/2$. Let $Y$ be a sum of $N$ independent Bernoulli trials, where the success probability is equal to $1/2$. Then, the probability that the algorithm terminates with at most $N$ iterations can be bounded as follows,

$$\mathbb{P}\left[\#\text{iterations} > N\right] \leq \mathbb{P}\left[ X \leq 4\ell^2\varepsilon^{-2} \log(n) \log \left( \frac{M}{\tau_{\min}} \right) \right]$$

$$\overset{(a)}{\leq} \mathbb{P}\left[ Y \leq 4\ell^2\varepsilon^{-2} \log(n) \log \left( \frac{M}{\tau_{\min}} \right) \right] \qquad \text{(Lemma A.3)}$$

$$\leq e^{-\frac{N}{4} \left( 1 - \frac{8\ell^2\varepsilon^{-2} \log(n) \log\left( \frac{M}{\tau_{\min}} \right)}{N} \right)^2} \qquad \text{(Lemma A.2)}$$

$$= e^{-\frac{\left( 4\log(n) + 8\ell^2\varepsilon^{-2} \log(n) \log\left( \frac{M}{\tau_{\min}} \right) \right)^2}{16\left( \log(n) + 4\ell^2\varepsilon^{-2} \log(n) \log\left( \frac{M}{\tau_{\min}} \right) \right)}} \leq \frac{1}{n}.$$

Therefore, with a probability of $1 - \frac{1}{n}$, the algorithm terminates with $\mathcal{O}\left( \ell^2\varepsilon^{-2} \log(n) \log \left( \frac{M}{\tau_{\min}} \right) \right)$ iterations of the while loop.

In Alg. 6, oracle queries occur during calls to UPDATE and PREFIX-SELECTION on Line 8, 18 and 4. The PREFIX-SELECTION algorithm, with input $(f, \mathcal{V}, s, \tau, \varepsilon)$, operates with 1 adaptive rounds and at most $|\mathcal{V}|$ queries. The UPDATE algorithm, with input $(f, V_0, \tau_0, \varepsilon)$, outputs $(V, \tau)$ with $1 + \log_{1-\varepsilon} \left( \frac{\tau}{\tau_0} \right)$ adaptive rounds and at most $|V| + n \log_{1-\varepsilon} \left( \frac{\tau}{\tau_0} \right)$ queries. Here, $\log_{1-\varepsilon} \left( \frac{\tau}{\tau_0} \right)$ equals the number of iterations in the while loop within UPDATE. Notably, every iteration is successful, as the threshold value is updated. Consequently, we can regard an iteration of the while loop in UPDATE as a separate iteration of the while loop in Alg. 6, where such iteration only update one threshold value $\tau_j$ and its corresponding candidate set $V_j$. So, each redefined iteration has no more than 2 adaptive rounds, and then the adaptivity of the algorithm should be no more than the number of successful iterations, which is $\mathcal{O}\left( \ell^2\varepsilon^{-2} \log(n) \log \left( \frac{M}{\tau_{\min}} \right) \right)$. Since there are at most $\ell n$ queries at each adaptive rounds, the query complexity is bounded by $\mathcal{O}\left( \ell^3\varepsilon^{-2} n \log(n) \log \left( \frac{M}{\tau_{\min}} \right) \right)$.

$\square$

## G.2. Analysis of Guarantees achieved by PIG (Theorem 4.1, Section 4)

In this section, we provide the analysis of the parallel $(1/4 - \varepsilon)$-approximation algorithm.

**Theorem 4.1.** *With input $(f, k, 2, \frac{\varepsilon M}{k}, \varepsilon)$, where $M = \max_{x \in \mathcal{U}} f(x)$, PIG (Alg. 6) returns $\{A'_1, A'_2\}$ with $\mathcal{O}\left(\varepsilon^{-4} \log(n) \log(k)\right)$ adaptive rounds and $\mathcal{O}\left(\varepsilon^{-5} n \log(n) \log(k)\right)$ queries with a probability of $1 - 1/n$. It satisfies that $\max\{f(A'_1), f(A'_2)\} \geq (1/4 - \varepsilon) f(O)$.*

*Proof of Theorem 4.1.* The adaptivity and query complextiy are quite straightforward. In the following, we will analyze the approximation ratio.

Let $S = O$ in Lemma G.1, it holds that

$$f(A'_l) \geq f(A_l), \forall l = 1, 2 \tag{73}$$
$$A_1 \cap A_2 = \emptyset \tag{74}$$

$$\Delta(O \mid A_1) + \Delta(O \mid A_2) \leq \frac{1}{(1-\varepsilon)^2}(f(A'_1) + f(A'_2)) + \frac{2\varepsilon M}{1-\varepsilon} \tag{75}$$

Then,

$$f(O) \leq f(O \cup A_1) + f(O \cup A_2) \qquad \text{(Submodularity, Nonnegativity, Inequality (74))}$$

$$\leq f(A_1) + f(A_2) + \frac{1}{(1-\varepsilon)^2}(f(A'_1) + f(A'_2)) + \frac{2\varepsilon M}{1-\varepsilon} \qquad \text{(Inequality (75))}$$

$$\leq 2\left(1 + \frac{1}{(1-\varepsilon)^2}\right) f(G) + \frac{2\varepsilon}{1-\varepsilon} f(O) \qquad \text{(Inequality (73) and } G = \arg\max\{f(A'_1), f(A'_2)\})$$

$$\Rightarrow f(G) \geq \frac{(1-3\varepsilon)(1-\varepsilon)}{2\left((1-\varepsilon)^2 + 1 + \frac{1}{k}\right)} f(O) \geq \left(\frac{1}{4} - \varepsilon\right) f(O)$$

$$\square$$

### G.3. Pseudocode and Analysis of PITG (Theorem 4.2, Section 4)

---

**Algorithm 11:** A randomized $(1/e - \varepsilon)$-approximation algorithm with $\mathcal{O}\left(\ell^3 \varepsilon^{-2} \log(n) \log(k)\right)$ adaptivity and $\mathcal{O}\left(\ell^4 \varepsilon^{-2} n \log(n) \log(k)\right)$ query complexity

---

1 **Procedure** PARALLELINTERPOLATEDGREEDY $(f, k, \varepsilon)$**:**

    **Input:** evaluation oracle $f : 2^{\mathcal{U}} \to \mathbb{R}_{\geq 0}$, constraint $k$, constant $\ell$, error $\varepsilon$

    **Initialize :** $G \leftarrow \emptyset, \varepsilon' \leftarrow \frac{\varepsilon}{2}, m \leftarrow \lfloor \frac{k}{\ell} \rfloor, M \leftarrow \max_{x \in \mathcal{U}} f(\{x\}), \tau_{\min} \leftarrow \frac{\varepsilon' M}{k}$

2     **for** $i \leftarrow 1$ *to* $\ell$ **do**

3         $\{A'_l : l \in [\ell]\} \leftarrow$ PARALLELINTERLACEGREEDY $(f_G, m, \ell, \tau_{\min}, \varepsilon')$

4         $G \leftarrow$ a random set in $\{G \cup A'_l : l \in [\ell]\}$

5     **return** $G$

---

This subsection presents the pseudocode and theoretical analysis of our parallel $(1/e - \varepsilon)$-approximation algorithm.

First, we provide the following lemma which provides a lower bound on the gains achieved after every iteration in Alg. 11.

**Lemma G.3.** For any iteration $i$ of the outer for loop in Alg. 11, it holds that

1) $\mathbb{E}[f(O \cup G_i)] \geq \left(1 - \frac{1}{\ell}\right) \mathbb{E}[f(O \cup G_{i-1})]$

2) $\mathbb{E}[f(G_i) - f(G_{i-1})] \geq \frac{1}{1 + \frac{\ell}{(1-\varepsilon')^2}}\left(1 - \frac{1}{m+1}\right)\left(\left(1 - \frac{1}{\ell}\right)\mathbb{E}[f(O \cup G_{i-1})] - \mathbb{E}[f(G_{i-1})] - \frac{\varepsilon'}{1-\varepsilon'}f(O)\right).$

*Proof of Lemma G.3.* Fix on $G_{i-1}$ at the beginning of this iteration, Since $\{A_l : l \in [\ell]\}$ are pairwise disjoint sets, by Proposition B.1, it holds that

$$\mathbb{E}[f(O \cup G_i) \mid G_{i-1}] = \frac{1}{\ell} \sum_{l \in [\ell]} f(O \cup G_{i-1} \cup A_l) \geq \left(1 - \frac{1}{\ell}\right) f(O \cup G_{i-1}).$$

Then, by unfixing $G_{i-1}$, the first inequality holds.

To prove the second inequality, also consider fix on $G_{i-1}$ at the beginning of iteration $i$. By Lemma G.1, $\{A_l : l \in [\ell]\}$ are paiewise disjoint sets, and the following inequalities hold,

$$A_l' \subseteq A_l, \Delta\left(A_l' \mid \emptyset\right) \geq \Delta\left(A_l \mid \emptyset\right), \forall 1 \leq l \leq \ell \tag{76}$$

$$\Delta\left(O_l \mid A_l\right) \leq \frac{\Delta\left(A_l' \mid \emptyset\right)}{(1-\varepsilon')^2} + \frac{\varepsilon' M}{(1-\varepsilon')\ell}, \forall 1 \leq l \leq \ell \tag{77}$$

$$\Delta\left(O_{l_2} \mid A_{l_1}\right) + \Delta\left(O_{l_1} \mid A_{l_2}\right) \leq \frac{1+\frac{1}{m}}{(1-\varepsilon')^2}\left(\Delta\left(A_{l_1}' \mid \emptyset\right) + \Delta\left(A_{l_2}' \mid \emptyset\right)\right) + \frac{2\varepsilon' M}{(1-\varepsilon')\ell}, \forall 1 \leq l_1 < l_2 \leq \ell \tag{78}$$

Then,

$$\sum_{l \in [\ell]} \Delta\left(O \mid A_l \cup G_{i-1}\right) \leq \sum_{l_1 \in [\ell]} \sum_{l_2 \in [\ell]} \Delta\left(O_{l_1} \mid A_{l_2} \cup G_{i-1}\right) \qquad \text{(Proposition B.1)}$$

$$= \sum_{l \in [\ell]} \Delta\left(O_l \mid A_l \cup G_{i-1}\right) + \sum_{1 \leq l_1 < l_2 \leq \ell} \left(\Delta\left(O_{l_1} \mid A_{l_2} \cup G_{i-1}\right) + \Delta\left(O_{l_2} \mid A_{l_1} \cup G_{i-1}\right)\right)$$

$$\text{(Lemma F.3)}$$

$$\leq \sum_{l \in [\ell]} \left(\frac{\Delta\left(A_l' \mid G_{i-1}\right)}{(1-\varepsilon')^2} + \frac{\varepsilon' M}{(1-\varepsilon')\ell}\right)$$

$$+ \sum_{1 \leq l_1 < l_2 \leq \ell} \left(\frac{\left(1+\frac{1}{m}\right)}{(1-\varepsilon')^2}\left(\Delta\left(A_{l_1}' \mid G_{i-1}\right) + \Delta\left(A_{l_2}' \mid G_{i-1}\right)\right) + \frac{2\varepsilon' M}{(1-\varepsilon')\ell}\right)$$

$$\text{(Inequalities (77) and (78))}$$

$$\leq \frac{\ell}{(1-\varepsilon')^2}\left(1+\frac{1}{m}\right) \sum_{l \in [\ell]} \Delta\left(A_l' \mid G_{i-1}\right) + \frac{\varepsilon'\ell}{1-\varepsilon'} f\left(O\right) \qquad (M \leq f\left(O\right))$$

$$\Rightarrow \left(1+\frac{\ell}{(1-\varepsilon')^2}\right)\left(1+\frac{1}{m}\right) \sum_{l \in [\ell]} \Delta\left(A_l' \mid G_{i-1}\right) \geq \sum_{l \in [\ell]} f\left(O \cup A_l \cup G_{i-1}\right) - \ell f\left(G_{i-1}\right) - \frac{\varepsilon'\ell}{1-\varepsilon'} f\left(O\right)$$

$$\text{(Inequality (76))}$$

$$\geq (\ell-1) f\left(O \cup G_{i-1}\right) - \ell f\left(G_{i-1}\right) - \frac{\varepsilon'\ell}{1-\varepsilon'} f\left(O\right)$$

Thus,

$$\mathbb{E}\left[f\left(G_i\right) - f\left(G_{i-1}\right) \mid G_{i-1}\right] = \frac{1}{\ell} \sum_{l \in [\ell]} \Delta\left(A_l' \mid G_{i-1}\right)$$

$$\geq \frac{1}{1+\frac{\ell}{(1-\varepsilon')^2}} \frac{m}{m+1} \left(\left(1-\frac{1}{\ell}\right) f\left(O \cup G_{i-1}\right) - f\left(G_{i-1}\right) - \frac{\varepsilon'}{1-\varepsilon'} f\left(O\right)\right) \qquad \text{(Proposition B.1)}$$

By unfixing $G_{i-1}$, the second inequality holds. $\qquad \square$

**Theorem 4.2.** *With input* $(f, k, \varepsilon)$ *such that* $\ell = \mathcal{O}\left(\varepsilon^{-1}\right) \geq \frac{4}{e\varepsilon}$ *and* $k \geq \frac{(2-\varepsilon)^2 \ell}{e\varepsilon\ell-4}$*,* PITG *(Alg. 11) returns $G$ such that* $\mathbb{E}\left[f\left(G\right)\right] \geq (1/e - \varepsilon)f\left(O\right)$ *with* $\mathcal{O}\left(\varepsilon^{-5} \log(n) \log(k)\right)$ *adaptive rounds and* $\mathcal{O}\left(\varepsilon^{-6} n \log(n) \log(k)\right)$ *queries with a probability of* $1 - \mathcal{O}\left(1/(\varepsilon n)\right)$*.*

*Proof of Theorem 4.2.* Since the algorithm contains a for loop which runs PARALLELINTERLACEGREEDY $\ell = \mathcal{O}\left(1/\varepsilon\right)$ times, by Lemma G.1, the adaptivity, query complexity and success probability holds immediately.

Next, we provide the analysis of approximation ratio. By solving the recurrence in Lemma G.3, we calculate the approximation ratio of the algorithm as follows,

$$\mathbb{E}\left[f\left(G_i\right)\right] \geq \left(1-\frac{1}{\ell}\right) \mathbb{E}\left[f\left(G_{i-1}\right)\right] + \frac{1}{1+\frac{\ell}{(1-\varepsilon')^2}} \left(1-\frac{1}{m+1}\right)\left(\left(1-\frac{1}{\ell}\right)^i - \frac{\varepsilon'}{1-\varepsilon'}\right) f\left(O\right)$$

$$\Rightarrow \mathbb{E}\left[f\left(G_{\ell}\right)\right] \geq \frac{\ell}{1+\frac{\ell}{(1-\varepsilon')^{2}}}\left(1-\frac{1}{m+1}\right)\left(\left(1-\frac{1}{\ell}\right)^{\ell}-\frac{\varepsilon'}{1-\varepsilon'}\left(1-\left(1-\frac{1}{\ell}\right)^{\ell}\right)\right) f\left(O\right)$$

$$\geq \frac{\ell-1}{1+\frac{\ell}{(1-\varepsilon')^{2}}}\left(1-\frac{1}{m+1}\right)\left(e^{-1}-\frac{\varepsilon'}{1-\varepsilon'}\left(1-e^{-1}\right)\right) f\left(O\right)$$

$$\geq \frac{1}{1-\frac{\ell}{k}}\left((1-\varepsilon')^{2}-\frac{2}{\ell}\right)\left(1-\frac{\ell}{k}\right)^{2}\left(e^{-1}-\frac{\varepsilon'}{1-\varepsilon'}\left(1-e^{-1}\right)\right) f\left(O\right)$$

$$\geq \frac{1}{1-\frac{\ell}{k}}\left((1-\varepsilon')^{2}-\frac{2}{\ell}-\frac{2(1-\varepsilon')^{2}\ell}{k}\right)\left(e^{-1}-\frac{\varepsilon'}{1-\varepsilon'}\left(1-e^{-1}\right)\right) f\left(O\right)$$

$$\geq \frac{1}{1-\frac{\ell}{k}}\left(1-(e+1)\varepsilon'\right)\left(e^{-1}-\frac{\varepsilon'}{1-\varepsilon'}\left(1-e^{-1}\right)\right) f\left(O\right) \qquad (\ell \geq \frac{2}{e\varepsilon'}, k \geq \frac{2(1-\varepsilon')^{2}\ell}{e\varepsilon'-\frac{2}{\ell}})$$

$$\geq \frac{1}{1-\frac{\ell}{k}}\left(e^{-1}-\varepsilon\right) f\left(O\right). \qquad (\varepsilon' = \frac{\varepsilon}{2})$$

By Inequality 37, the approximation ratio of Alg. 10 is $e^{-1}-\varepsilon$. $\qquad \square$

# H. Experimental Setups and Additional Empirical Results

In the section, we introduce the settings in Section 5 further, and discuss more experimental results on SM-GEN and SM-MON.

## H.1. Applications

**Maxcut.** In the context of the maxcut application, we start with a graph $G = (V, E)$ where each edge $ij \in E$ has a weight $w_{ij}$. The objective is to find a cut that maximizes the total weight of edges crossing the cut. The cut function $f : 2^{V} \to \mathbb{R}_{\geq 0}$ is defined as follows,

$$f(S) = \sum_{i \in S} \sum_{j \in V \setminus S} w_{ij}, \forall S \subseteq V.$$

This is a non-monotone submodular function. In our implementation, for simplicity, all edges have a weight of 1.

**Revmax.** In our revenue maximization application, we adopt the revenue maximization model introduced in (Hartline et al., 2008), which we will briefly outline here. Consider a social network $G = (V, E)$, where $V$ denotes the buyers. Each buyer $i$'s value for a good depends on the set of buyers $S$ that already own it, which is formulated by

$$v_i(S) = f_i\left(\sum_{j \in S} w_{ij}\right),$$

where $f_i : \mathbb{R}_{\geq 0} \to \mathbb{R}_{\geq 0}$ is a non-negative, monotone, concave function, and $w_{ij}$ is drawn independently from a distribution. The total revenue generated from selling goods to the buyers $S$ is

$$f(S) = \sum_{i \in V \setminus S} f_i\left(\sum_{j \in S} w_{ij}\right).$$

This is a non-monotone submodular function. In our implementation, we randomly choose each $w_{ij} \in (0, 1)$, and $f_i(x) = x^{\alpha_i}$, where $\alpha_i \in (0, 1)$ is chosen uniformly randomly.

## H.2. Datasets

**er** is a synthetic random graph generated by Erdös-Rényi model (ERDdS & R&wi, 1959) by setting number of nodes $n = 100,000$ and edge probability $p = \frac{5}{n}$.

**web-Google** (Leskovec et al., 2009) is a web graph of $n = 875,713$ web pages as nodes and $5,105,039$ hyperlinks as edges.

**musae-github** (Rozemberczki et al., 2019) is a social network of GitHub developers with $n = 37,700$ developers and $289,003$ edges, where edges are mutual follower relationships between them.

**twitch-gamers** (Rozemberczki & Sarkar, 2021) is a social network of $n = 168,114$ Twitch users with $6,797,557$ edges, where edges are mutual follower relationships between them.

### H.3. Additional Results

Fig. 5 provides additional results on musae-github dataset with $n = 37,700$ and web-Google dataset with $n = 875,713$. It shows that as $n$ and $k$ increase, our algorithms achieve superior on objective values. The results of query complexity and adaptivity align closely with those discussed in Section 5. Notably, the number of adaptive round of PITG exceeds $k$ on musae-github, which may be attributed to the dataset's relatively small size.

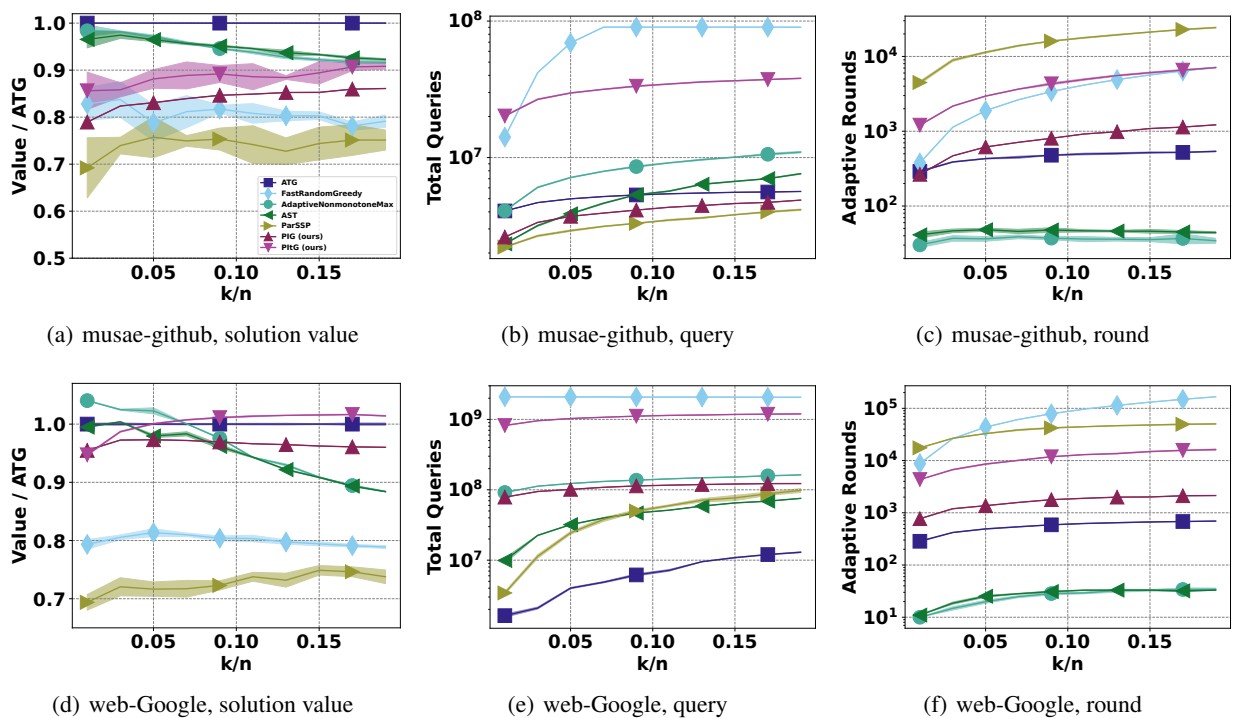

(a) musae-github, solution value      (b) musae-github, query      (c) musae-github, round

(d) web-Google, solution value      (e) web-Google, query      (f) web-Google, round

*Figure 5.* Results for `revmax` on musae-github with $n = 37,700$, and `maxcut` on web-Google with $n = 875,713$.

