# OpenReview forum: "Breaking Barriers: Combinatorial Algorithms for Non-Monotone Submodular Maximization with Sublinear Adaptivity and $1/e$ Approximation"
_ICML.cc/2025/Conference — ICML 2025 poster_

### Official Review · Reviewer_AdQT · 2025-02-20

**Overall Recommendation:** 2

**Summary:**

This paper studies the problem of maximizing a non-monotone submodular function subject to a size constraint. In the problem, we are given a set $\mathcal{U}$ of $n$ elements, (a value-oracle of) a non-monotone submodular function $f$, and a positive integer $k$, and the goal is to find a subset $S$ of $\mathcal{U}$ that maximizes $f(S)$ subject to $|S|\leq k$. A well-known randomized approximation algorithm for the problem achieves the $1/e$-approximation (in expectation). The query complexity of this algorithm is $O(kn)$, which was later improved to $O(n)$ (for some regime of $k$). Additionally, to further improve the scalability, much work has studied parallelizable algorithms for the problem. The parallelizability of an algorithm is measured by the adaptive complexity, i.e., the number of buckets needed to divide the queries to $f$ into adaptive rounds such that within each round the sets to be queried only depend on the results of queries in previous rounds. The lower the adaptive complexity, the higher the parallelizability because the queries in each round can be arbitrarily parallelizable. The current best approximation ratio among those achieved by the algorithms with a sublinear adaptive complexity is $1/e-\epsilon$; however, the algorithm achieving that ratio needs to estimate the multilinear extension of $f$, requiring a substantial number of queries to the original $f$ and making it impractical. The main contribution of this paper is a combinatorial $(1/e-\epsilon)$-approximation algorithm with a sublinear adaptive complexity and a nearly-linear query complexity (to the original $f$).

## update after rebuttal

I maintain my overall recommendation (weak reject) based on the discussion with the authors.

**Claims And Evidence:**

Although a lot of evidences (and even the proposed algorithm itself) are presented in the appendix, the claims seem to be supported.

**Essential References Not Discussed:**

N/A

**Ethical Review Concerns:**

This paper doesn't include impact statements.

**Ethical Review Flag:**

Flag this paper for an ethics review.

**Ethics Expertise Needed:**

["Inappropriate Potential Applications & Impact  (e.g., human rights concerns)"]

**Experimental Designs Or Analyses:**

The proposed algorithms are evaluated using two applications, i.e., Maximum Cut and Revenue Maximization, with large-scale real-world networks. The authors state that they only provide the results for er in the main body, but the results for twitch-gamers are also given there. The selection criterion of baseline methods is unclear. There are many existing parallel algorithms for the problem that are listed in Table 1 but not tested in the experiments. Also, regarding the implementation of the baseline methods, it seems unfair to employ a random subset for the algorithms that require to solve the unconstrained problem.

**Methods And Evaluation Criteria:**

The proposed $(1/e-\epsilon)$-approximation algorithm is called ParallelInterpolatedGreedy (PItG). This algorithm repeats the subroutine called ParallelInterlaceGreedy (PIG), which is also a novel component and achieves the $(1/4-\epsilon)$-approximation. PIG can be seen as a parallel counterpart of the existing non-parallel InterlaceGreedy and InterpolatedGreedy. To obtain PIG and its approximation ratio analysis, the authors first introduce the blended marginal gains strategy, which provides multiple upper bounds on the marginal gain of an element with respect to sets kept in the algorithm. This strategy simplifies InterlaceGreedy and InterpolatedGreedy and enables to incorporate thresholding techniques to reduce their query complexity.

**Other Comments Or Suggestions:**

Typos and minor things:
- Many equations are presented using smaller fonts.
- Sec 1: $\mathcal{U}$ and $\epsilon$ are undefined.
- Tab 1: The notation || is undefined.
- Sec 3.1: the maximum singleton $a_0$ is undefined.
- Alg 1: dummy elements should be defined.
- Alg 2: "size of solution" is misleading.

**Other Strengths And Weaknesses:**

I acknowledge the technical contribution of the paper but also believes that the presentation could have been done better in both non-technical and technical parts. Some suggestions are given below:
- The presentation of the technical contents should be improved. The most significant issue is that the authors directly dive into the technical details and do not explain the overall algorithm design and analysis before that. This makes the technical contents quite puzzled. It is understandable that Algorithms 1 and 2 are improved versions of the existing InterlaceGreedy and InterpolatedGreedy, respectively, but how those algorithms are utilized in the proposed algorithms is quite unclear. Do the authors really need Algorithm 1 to design the proposed algorithms? Also, the behavior of the thresholding techniques used in the proposed algorithms are not explained in the main text. What the property of the alternating additions means should be explained.
- The problem of maximizing non-monotone submodular functions is not motivated accordingly. Indeed, the application examples listed by the authors are mostly with monotone submodular functions (e.g., influence maximization).
- There are some descriptions that do not make sense. For instance, in the abstract, the authors state that they propose a $(1/4-\epsilon)$-approximation algorithm with high probability for the problem, but the approximation ratio itself is nothing new, so mentioning it as one of the proposals would be weird for the readers. Later it turns out that the algorithm is a preliminary result, which should have been explicitly mentioned. The same thing can be pointed out in Section 1.1. In Section 3.1, the authors state that intuitively adding an element from $O$ to the solution should not negatively impact it and the question "can the second interlaced greedy step be eliminated?" naturally arises. However, the question is not reasonable. If the addition does not negatively impact the solution, the second interlaced greedy step should work well, just meaning that the first step can be eliminated.

**Questions For Authors:**

See Other Strengths and Weaknesses.

**Relation To Broader Scientific Literature:**

Submodular maximization has actively been studied in machine learning. Dealing with non-monotone submodular functions is still important, and the parallelization is a reasonable way to accelerate the algorithms. The proposed algorithm is the first combinatorial $(1/e-\epsilon)$-approximation algorithm with a sublinear adaptive complexity and a nearly-linear query complexity, which could be a good addition to the literature.

**Theoretical Claims:**

I checked the correctness of the contents in the main body. Other than some undefined notation and typos, I did not find any technical issue.

---

> ### Author Rebuttal · Authors · 2025-03-31
>
> We thank the reviewer for the thoughtful feedback. Regarding the suggestions related to paper presentation and organization, we have provided detailed responses to each point in our reply to Reviewer KJuz. For other questions, we address each point below.
>
> ```
> The authors state that they only provide the results for er in the main body, but the results for twitch-gamers are also given there.
> ```
> Thank you for pointing this out. We will correct our statement in the next version.
>
> ```
> The selection criterion of baseline methods is unclear.
> ```
> Thank you for the suggestion. In our current experiments, we focused on parallel algorithms designed specifically for size-constrained problems. While algorithms like ParKnapsack, ParSKP, and ParSSP can handle more complex constraints, such as knapsack constraints, they currently suffer from high query complexity, which limits their practicality. For completeness, we plan to include experimental results for these algorithms in our next version.
>
> ```
> ... it seems unfair to employ a random subset for the algorithms that require to solve the unconstrained problem.
> ```
> We appreciate the reviewer’s observation regarding the use of a random subset for baseline methods that require solving the unconstrained problem. We would like to clarify the rationale behind our experimental design:
> - The $1/2$-approximation constant-adaptive algorithm proposed by [1] is used for ANM, AST, and ATG. However, this algorithm relies on querying the multilinear extension $F$ of the objective function, whose estimation requires an impractically large number of queries to the value oracle $f$. This makes the algorithm prohibitively expensive to run in experiments, despite its theoretical guarantees.
> - Based on our experimental evaluation, the solutions returned by unconstrained algorithms (e.g., [1]) do not dominate other solutions. Thus, their inclusion would not meaningfully improve objective values in practice.
> - In summary, while implementing [1] would align with theoretical comparisons, it would incur significant query complexity without commensurate gains in solution quality. Our goal was to focus on practical and scalable baselines, which better reflect real-world constraints.
>
> [1] Chen, L., Feldman, M., & Karbasi, A. (2019). Unconstrained submodular maximization with constant adaptive complexity. In Charikar, M., & Cohen, E. (Eds.), Proceedings of the 51st Annual ACM SIGACT Symposium on Theory of Computing, STOC 2019, Phoenix, AZ, USA, June 23-26, 2019, pp. 102–113. ACM.
> ```
> The problem of maximizing non-monotone submodular functions is not motivated accordingly. Indeed, the application examples listed by the authors are mostly with monotone submodular functions (e.g., inﬂuence maximization).
> ```
> We appreciate the reviewer’s feedback. For the same application, different submodular functions can be defined as either monotone or non-monotone. For example, influence maximization can have non-monotone variants when considering competition or oversaturation effects.
>
> We also acknowledge that some of our cited papers focus on monotone functions. To clarify this distinction, we will differentiate between monotone and non-monotone papers and add the following paper as a non-monotone variant for the revenue maximization application.
> - Amanatidis, Georgios, et al. "Fast adaptive non-monotone submodular maximization subject to a knapsack constraint." Advances in neural information processing systems 33 (2020).
>
> ```
> ... but the approximation ratio itself is nothing new, so mentioning it as one of the proposals would be weird for the readers ...
> ```
> We appreciate this insightful observation. While the 1/4-approximation ratio itself is not novel, our algorithm achieve this ratio with high probability ($\ge 1-1/n$). In Cui et al. (2023), ParSSP achieves $1/4$ approximation ratio in expectation. We will revise our abstract as follows which leads with our primary 1/e-approximation result and present the 1/4-result as a byproduct.
>
> "... In this work, we focus on size constraints and present the first combinatorial algorithm matching this bound - a randomized parallel approach achieving $1/e − \varepsilon$ approximation ratio. This result bridges the gap between continuous and combinatorial approaches for this problem. As a byproduct, we also develop a simpler $(1/4 − \varepsilon)$-approximation algorithm with high probability $( \ge 1 − 1/n)$..."
>
>
> ```
> In Section 3.1, the authors state that intuitively adding an element from $O$... just meaning that the ﬁrst step can be eliminated.
> ```
> We sincerely appreciate this insightful observation. We agree that the explanation here might be misleading. We will revise the text to remove ambiguous descriptions as follows.
>
> "This complex architecture naturally raises a fundamental question: Can we develop an alternative analysis framework that eliminates the need to explicitly guess the position of $o_{\max}$?"

---

> > ### Comment · Reviewer_AdQT · 2025-04-02
> >
> > Thank you for taking my comments into account.
> >
> > As mentioned in my original feedback, my main concern is about the quality of presentation in both the technical and non-technical sections. I appreciate the detailed comments on the planned revisions. However, given the extent of the necessary changes across the entire paper, I can’t guarantee that the paper will be ready for publication after these revisions.

---

### Official Review · Reviewer_9FCa · 2025-03-07

**Overall Recommendation:** 3

**Summary:**

This work introduces enhanced solutions for maximizing a non-monotone submodular function under a cardinality constraint. The authors concentrate on solution quality, query count, and adaptivity, which are the key performance indicators in this area of research.

**Claims And Evidence:**

Yes.

**Essential References Not Discussed:**

This paper provides a deep comparison with the publications in this field.

**Experimental Designs Or Analyses:**

Yes.

**Methods And Evaluation Criteria:**

Yes.

**Other Comments Or Suggestions:**

The paper would benefit from a clearer explanation of its core and novel ideas, which will improve its impact. The algorithm's complexity made it a bit hard to follow; providing more intuition would improve understanding.

Although the paper does not present an impact statement, I do not have any ethical concern.

**Other Strengths And Weaknesses:**

"+" The paper is well-written.

"+" Thorough and clearly stated comparisons with previous work are provided.

"+" The results are interesting, and the techniques might be applicable to other problems in this field.

"+" Bridging the gap between continuous and combinatorial algorithms is of high interest.

"+" The experimental section uses well-known applications in the field.

"+" The strengths and weaknesses of the approach are clearly stated.

"-" The results provide marginal improvement compared to the state of the art.

"-" The experimental section shows that the proposed algorithms have higher quality but also higher queries and adaptivity in some cases.

**Questions For Authors:**

Could you please provide a highlight of the new ideas in this work.

**Relation To Broader Scientific Literature:**

This paper presents results that are relevant to researchers in submodular maximization

**Theoretical Claims:**

Yes.

---

> ### Author Rebuttal · Authors · 2025-03-31
>
> We thank the reviewer for the constructive comments.
> Regarding the suggestions related to paper presentation and organization, we have provided detailed responses to each point in our reply to Reviewer KJuz.
>
> ```
> The results provide marginal improvement compared to the state of the art.
> ```
> We sincerely appreciate the reviewer’s feedback. While the improvement in approximation ratio may appear incremental at first glance, we would like to clarify the significance of our contribution.
>
> Our work represents the first advancement in five years since the breakthrough $1/e$-approximation achieved by a parallel continuous algorithm (Ene & Nguyen, 2020). Specifically, we improve the approximation ratio for parallel combinatorial algorithms from $1/4$ to $1/e$, bridging a key gap between combinatorial and continuous approaches. Given that the current best-known approximation ratio for non-monotone submodular maximization is 0.401 [1], an improvement exceeding 0.11 is, in fact, a meaningful step forward in this challenging problem space.
>
> [1] Buchbinder, N. and Feldman, M. Constrained submodular maximization via new bounds for dr-submodular functions. In Proceedings of the 56th Annual ACM Symposium on Theory of Computing, STOC, 2024.

---

### Official Review · Reviewer_KJuz · 2025-03-12

**Overall Recommendation:** 2

**Summary:**

The problem of submodular maximization under a cardinality constraint is a fundamental topic with numerous applications. The submodular function, defined on a ground set $U$, can be either monotone or non-monotone, with the non-monotone case being notably more challenging.

There is a wealth of results on non-monotone submodular maximization under a cardinality constraint. The current state-of-the-art algorithm is a $1/e$ approximation algorithm based on continuous greedy, which requires $O(k^2 n \log^2(n))$ oracle queries. This paper introduces a sublinear algorithm that achieves the same $1/e$ approximation guarantee but uses only $O(n \log(n) \log(k))$ queries. This makes the algorithm sublinear, assuming that $O(nk)$ is considered the linear query complexity regime. Furthermore, the paper presents a parallel version of this algorithm, which requires $O(\log(n) \log(k))$ adaptive rounds.

Overall, the paper is well-written, and I appreciate the contributions. However, there are a few aspects that need improvement:

1. Clarity and Readability:
   The paper requires significant polishing. The current version is difficult to follow, even for an expert in the area. Frequent references to the appendix make the paper hard to read, as it requires constant switching between the main text and the appendix. I recommend minimizing these references and integrating more details directly into the paper to improve its flow.

2. Algorithm Overview:
   The paper lacks a clear, intuitive overview of how the sublinear and parallel algorithms function. Section 5 describes the algorithmic steps but does not provide sufficient intuition for why these steps are taken and how they contribute to the solution. While Sections 5.1 and 5.2 attempt to offer an overview, they are cluttered with references to the appendix, which makes them hard to follow. I suggest adding a more detailed, intuitive explanation of the algorithm's workings, especially in the early sections.

3. Unusual Notations:
   In Algorithm 3 (the main algorithm), the notations used in Lines 4 and 8 are unconventional. It is unclear whether these notations are intentional or typographical errors. There is no explanation of these symbols in the paper, which makes it difficult to understand their meaning. I recommend clarifying or correcting these notations.

4. Preliminaries Section:
   The "Preliminaries" section is overly brief and lacks a proper explanation of key concepts, such as the parallel model and adaptivity. These concepts should be clarified earlier in the paper. For example, when Algorithm 3 and the "UPDATE" procedure refer to parallelism, it is not clear whether this refers to $\ell$ parallel machines or some other form of parallelism. A more detailed explanation is needed.

5. Table 1:
   Table 1 is difficult to follow. It includes many results, some of which are not essential for understanding the key contributions of the paper. I recommend simplifying the table to focus on the most important results.

6. Algorithm Structure:
   I suggest reintroducing the procedures "UPDATE", "DISTRIBUTE", and "Pre-fix selection" into the main body of the paper, or alternatively, dedicating a separate section to explain them. Additionally, Algorithm 3 could be split into two or three sub-algorithms to improve readability—one for Lines 6-14 and another for Lines 16-24.


General Suggestions:

- Overview and Intuition: It would be helpful to include a concise, intuitive overview of the algorithm in the first few pages, which explains its core idea and key steps. This will make it easier for the reader to understand the subsequent details.

- Sublinear Query Complexity: While I understand the concept of sublinear query complexity, it would be useful for the paper to explicitly define and explain it. This would ensure that the reader has a clear understanding of what is meant by sublinear complexity in the context of this paper.

- Parallel Model and Adaptive Rounds: A more detailed explanation of the parallel model and the concept of adaptive rounds is necessary. This is a crucial aspect of the algorithm, and it would benefit from a clearer definition and description.


Overall, the paper presents an interesting and valuable contribution. With some revisions to improve clarity, provide more intuition, and ensure consistency in notation and terminology, the paper would be much more accessible and impactful.

**Claims And Evidence:**

Nothing

**Essential References Not Discussed:**

no

**Experimental Designs Or Analyses:**

For me, the most important part was the theory part.

**Methods And Evaluation Criteria:**

Nothing

**Other Comments Or Suggestions:**

It's given in the review.

**Other Strengths And Weaknesses:**

It's given in the review.

**Questions For Authors:**

It's given in the review.

**Relation To Broader Scientific Literature:**

The paper contributes to the well-studied problem of submodular maximization under a cardinality constraint, which has numerous applications. The broader scientific literature on this topic includes both monotone and non-monotone submodular functions, with the non-monotone case being particularly challenging.

A key reference point in the literature is the state-of-the-art $1/e$ approximation algorithm based on continuous greedy, which requires $O(k^2 n \log^2(n))$ oracle queries. The primary contribution of this paper is introducing a sublinear algorithm that achieves the same $1/e$ approximation guarantee but significantly reduces the query complexity to $O(n \log(n) \log(k))$. This advancement aligns with recent trends in submodular optimization that seek more efficient algorithms with lower query complexity.

Additionally, the paper extends its contribution by presenting a parallel version of the algorithm, which requires only $O(\log(n) \log(k))$ adaptive rounds. This connects to the broader literature on parallel submodular optimization, an area of increasing importance for large-scale applications. However, the review suggests that the paper could benefit from a clearer discussion of the parallel model and adaptivity, ensuring that the contribution is well-situated in the context of existing work.

In summary, the paper builds on and improves key results in submodular maximization, particularly by introducing a sublinear and parallelizable approach while maintaining the best-known approximation guarantee. To strengthen its connection to prior literature, the paper should explicitly compare its results to previous work in terms of efficiency, adaptivity, and practical implications.

**Theoretical Claims:**

The correctness of the theoretical claims was not explicitly questioned in the review. However, the review does highlight issues related to clarity, notation, and organization, which may indirectly affect the readability and verification of the proofs. Specifically:

Notation Issues: The review notes that some notations in Algorithm 3 (Lines 4 and 8) are unconventional and unclear. If these notations are crucial for the proofs, their ambiguity could make it difficult to verify correctness. It is recommended to clarify or correct these notations.

Appendix References: Frequent references to the appendix make it hard to follow the logical flow of the paper. If key proofs rely on these sections, integrating more details into the main text would help in assessing their correctness.

Preliminaries and Definitions: The review points out that the "Preliminaries" section is overly brief, lacking proper explanations of key concepts like the parallel model and adaptivity. If these concepts play a role in the proofs, their insufficient explanation could hinder proper verification.

While no specific errors in the theoretical results were identified, improving the clarity, notation, and structure of the paper would facilitate a more thorough and accessible verification of the proofs.

---

> ### Author Rebuttal · Authors · 2025-03-31
>
> We thank the reviewers for their constructive suggestions. We will revise the manuscript to improve clarity and presentation. In Section 1.1, we clarify the technical contributions, focusing on how the simplified InterpolatedGreedy enables parallelization and enhancing the explanation of our parallel algorithm (PIG). In Section 2, we refine definitions of adaptivity, parallel algorithms, and the thresholding technique. We restructure Section 3, remove Section 4 for conciseness, and improve Section 5 with an overview, pseudocode, and key subroutine introductions.
>
> Below, we detail these changes.
>
> 1. Section 1.1. Contributions.
>     - Our first contribution is a technical contribution that simplifies both InterpolatedGreedy and InterlaceGreedy. Importantly, by eliminating the low-success-probability requirement, the simplified InterpolatedGreedy serves as an efficient framework for parallel algorithm design. We will clarify this in the revised manuscript as follows.
>
>     "... Most importantly, these simplified variants establish the theoretical foundation for parallel algorithms. By removing branching dependencies and probabilistic guesswork, our framework enables the first efficient parallelization of these interlaced greedy approaches while preserving their approximation guarantees."
>     - Our second contribution is the proposed parallel algorithms, where we explain how integrating the interlaced greedy approach with threshold sampling achieves both theoretical guarantees and practical efficiency. The revised version is below.
>
>     "... The core innovation of PIG lies in its novel threshold sampling procedure, which simultaneously preserves the crucial alternating selection property of interlaced greedy methods while enabling efficient parallel implementation.
>
>     Like prior parallel algorithms, PIG maintains descending thresholds for each solution, adding elements whose average marginal gain exceeds the current threshold. However, PIG introduces two critical modifications to maintain the interlaced greedy structure: 1) strict synchronization of batch sizes across all $\ell$ parallel solutions, and 2) coordinated element selection to maintain sufficient marginal gain for each solution.
>
>     This design achieves three fundamental properties. First, it preserves the essential alternating selection property of the interlaced greedy methods. Second, through threshold sampling, it geometrically reduces the size of candidate sets - crucial for achieving sublinear adaptivity. Third, its efficient batch selection ensures each added batch provides sufficient marginal contribution to the solution. Together, these properties allow PIG to match the approximation guarantees of the vanilla interlaced greedy method while achieving parallel efficiency..."
> 2. Section 2. Preliminary
>     - We will include formal definitions of adaptivity and parallel algorithms in the revised manuscript. Additionally, we will clarify the concept of sublinear adaptive algorithm.
>     - We will provide a concise introduction to the thresholding technique, explaining its role in the parallel algorithms.
> 3. Section 3. The Blending Technique for Greedy
>     - We will retitle this section as "A Parallel-Friendly Greedy Variant via Blending Technique" to better reflect its focus on developing an efficient parallelizable algorithm.
>     - We will focus on the simplified InterpolatedGreedy (achieving a $1/e$-approximation) and leave the simplified InterlaceGreedy (achieving a $1/4$-approximation) in appendix.
>     - The structure of the section will be as follows:
>         - Section 3.1: An Overview of Prior Work and Its Limitation.
>             - Summarizes how InterpolatedGreedy works and identifies key bottlenecks for parallelization.
>         - Section 3.2: Motivation for Simplification.
>             - Explains the rationale behind our approach and its advantages.
>         - Section 3.3: Technical Overview of Blended Marginal Gains Strategy for InterpolatedGreedy.
>             - Provides a high-level explanation of how the blended marginal gains strategy enhances InterpolatedGreedy's efficiency.
> 4. Section 4. Preliminary Warm-Up of Parallel Approaches: Nearly-Linear Time Algorithms (Removal)
>     - To streamline the presentation and avoid redundancy, we will remove this section. This adjustment allows us to dedicate more space to Sections 3 and 5, where we will elaborate on the high-level ideas and technical details of the proposed algorithms.
> 5. Section 5. Sublinear Adaptive Algorithms
>     - Section 5.1: Subroutines for PIG.
>         - This subsection will include the pseudocodes for key subroutines (Distribute, Update, Prefix-Selection) and provide a high-level description of their functionality.
>     - Section 5.2: Algorithm Overview
>         - Here, we will introduce the core intuition behind the algorithm, emphasizing its novel components and how they collectively achieve sublinear adaptivity.

---

### Official Review · Reviewer_GBRx · 2025-03-17

**Overall Recommendation:** 4

**Summary:**

The paper studies the non-monotone submodular maximization in the parallel model. They provide two algorithms with the best one having a $1/e$ approximation factor, $O(log(n)log(k))$ adaptivity and almost linear query complexity.Previously, the only algorithm with a $1/e$ approximation factor and logarithmic adaptivity relied on continuous approaches, which required many queries.

**Claims And Evidence:**

They provide proof for their claims.

**Essential References Not Discussed:**

No

**Experimental Designs Or Analyses:**

No.

**Methods And Evaluation Criteria:**

Their experiments seem reasonable, but the graph on solution value is missing the data for the ATG algorithm, which appears to have better adaptivity and query complexity. Comparing with PARSSP would also be interesting, as it achieves better approximation factors than other logarithmic adaptivity algorithms.

**Other Comments Or Suggestions:**

Please add the ATG data in Figures 2a and 2d.

**Other Strengths And Weaknesses:**

Their result is interesting because it achieves the best-known approximation factor with logarithmic adaptivity and near-linear query complexity. However, a limitation of their approach is that it does not achieve $O(\log n)$ adaptivity, unlike the continuous algorithm.

**Questions For Authors:**

N/A

**Relation To Broader Scientific Literature:**

There are many works on this problem, and among the works with logarithmic adaptivity (which is usually the most important factor for parallel algorithms), most works have worse approximation factors compared to them and the only work that matches their approximation factor uses continuous methods which are not practical.

**Theoretical Claims:**

I reviewed the proofs in the main part of the paper, but not the appendix.

---

> ### Author Rebuttal · Authors · 2025-03-31
>
> ```
> Their experiments seem reasonable, but the graph on solution value is missing the data for the ATG algorithm, which appears to have better adaptivity and query complexity.
> ```
> Thank you for pointing this out. Since the objective value is normalized by ATG, the ATG data would appear as a horizontal line with value 1 in the graph. To improve clarity, we will add this line to the graph.
>
>
> ```
> Comparing with PARSSP would also be interesting, as it achieves better approximation factors than other logarithmic adaptivity algorithms.
> ```
> Thank you for the suggestion, we will add the comparison to PARSSP in the next version.

---

### Decision · Program_Chairs · 2025-05-01

**Decision:**

Accept (poster)

**Comment:**

This work studies low-adaptivity algorithms for non-monotone submodular maximization subject to a cardinality constraint. Specifically, it gives the first $(1/e-\varepsilon)$-approximation algorithm with nearly optimal query and adaptivity complexity. Note that [Ene-Nguyen, ICML 2020] gave a *continuous* algorithm for this problem that achieves a $(1/e-\varepsilon)$-approximation ratio with near-linear adaptivity but $O(n k^2 \log^2 n)$ calls to an oracle for multilinear extension queries. This work also presents a simpler algorithm called `ParallelInterlaceGreedy` (Algorithm 3) that achieves a $(0.25 - \varepsilon)$-approximation guarantee w.h.p. in $O(n \log(n) \log(k))$ queries and $O(\log(n) \log(k))$ adaptive rounds, improving on the algorithm and analysis in [Kuhnle, NeurIPS 2019]. The authors also include an empirical study comparing their algorithms to related work on standard benchmarks.

Overall, this is a borderline submission with mixed reviews (scores: [4, 2, 3, 2]). The improvement to the approximation ratio for low-adaptivity combinatorial algorithms for SM-GEN is clearly a very nice improvement. However, most reviewers commented that the presentation/organization (for both technical and non-technical parts) noticeably affects the readability and verification of proofs.